



# A comparison between Envisat and ICESat sea ice thickness in the Antarctic

Jinfei Wang[1], Chao Min[1], Robert Ricker[2], Qian Shi[1], Bo Han[1], Stefan Hendricks[2], Renhao Wu[1], Qinghua Yang[1]

[1]School of Atmospheric Sciences, Sun Yat-sen University, and Southern Marine Science and Engineering Guangdong Laboratory (Zhuhai), Zhuhai 519082, China
[2]Alfred Wegener Institute Helmholtz Centre for Polar and Marine Research, Bremerhaven 27570, Germany

*Correspondence to*: Qian Shi (shiq9@mail.sysu.edu.cn)

**Abstract.** The crucial role that Antarctic sea ice plays in the global climate system is strongly linked to its thickness. While field observations are too sparse in the Antarctic to determine long-term trends of the Antarctic sea ice thickness (SIT) on a hemispheric scale, satellite radar altimetry data can be applied with a promising prospect. European Space Agency Climate Change Initiative – Sea Ice Project (ESA SICCI) includes sea ice freeboard and sea ice thickness derived from Envisat, covering the entire Antarctic year-round from 2002 to 2012. In this study, the SICCI Envisat SIT in the Antarctic is first compared with a conceptually new ICESat SIT product retrieved from an algorithm employing modified ice density. Both data sets are compared to SIT estimates from upward−looking sonar (ULS) in the Weddell Sea, showing mean differences (MD) and standard deviations (SD) of 1.29 (0.65) m for Envisat-ULS, while we find 1.11 (0.81) m for ICESat-ULS, respectively. The inter-comparisons are conducted for three seasons except winter, based on the ICESat operating periods. According to the results, the differences between Envisat and ICESat SIT reveal significant temporal and spatial variations. More specifically, the smallest seasonal SIT MD (with SD shown in brackets) of 0.00 m (0.39 m) for Envisat-ICESat for the entire Antarctic is found in spring (October-November) while larger MD of 0.52 m (0.68 m) and 0.57 m (0.45 m) exist in summer (February-March) and autumn (May-June), respectively. It is also shown that from autumn to spring, mean Envisat SIT decreases while mean ICESat SIT increases. Our findings suggest that overestimation of Envisat sea ice freeboard, potentially caused by radar backscatter originating from inside the snow layer, primarily accounts for the differences between Envisat and ICESat SIT in summer and autumn, while the uncertainties of snow depth product are not the dominant cause of the differences.

## 1 Introduction

Antarctic sea ice plays an important role in the global climate system by reflecting the solar energy and modulating the surface water salinity (Goosse and Zunz, 2014; Massom et al., 2018; Maksym, 2019). In the context of global warming and the significant declines of Arctic sea ice cover, Antarctic sea ice extent has unexpectedly increased over recent decades (Zhang, 2007; Parkinson and Cavalieri, 2012; Comiso et al., 2017), but dropped to a historic low in 2017 (Turner and Comiso, 2017).



However, it is still unclear if the recent increase in Antarctic sea ice extent is also associated with a similar change in sea ice thickness. Sea ice thickness combined with sea ice extent is necessary to quantify the sea ice volume and sea ice mass (e.g., Kurtz and Markus, 2012; Massonnet et al., 2013). Changes in sea ice volume can influence the fresh water input into the Southern Ocean. Moreover, sea ice thickness is also necessary for assessing sea ice mass balance, the surface energy budget, and predicting changes in the polar climate system. Compared to the Arctic, knowledge about Antarctic sea ice thickness is still sparse. More accurate estimations are needed to monitor and quantify global sea ice volume more precisely (Connor et al., 2009), and improve sea ice components in model simulations (e.g., McLaren et al., 2006).

However, Antarctic sea ice thickness information is difficult to obtain. In situ measurements like drilling data (e.g., Meiners et al., 2012) are accurate but extremely limited in temporal and spatial coverage, and hence they cannot be used to obtain an understanding of large-scale Antarctic sea ice thickness processes. Ship-based observations collected from the Antarctic Sea Ice Processes and Climate (ASPeCt) (Worby et al., 2008a) can provide more spatial information than drilling but they tend to underestimate the actual thickness because of visual interpretation limitations and biases due to ship routing preferably through thinner ice (Giles et al., 2008; Williams et al., 2015). In addition, airborne electromagnetic (AEM) data which measure total freeboard (sea ice freeboard plus snow depth) were collected during expeditions like ISPOL (2004/05) (El Naggar et al., 2007), WWOS (2006) (Lemke, 2009) and AWECS (2013) (Lemke, 2014). Yet, the Antarctic AEM data is still sparse and has mostly been obtained in the Weddell Sea. The airborne remote sensing program NASA Operation IceBridge provides along-track data of total freeboard and snow depth estimations in the Weddell Sea and Amundsen-Bellingshausen Sea (Koenig et al., 2010), which have been investigated in some valuable studies previously (e.g., Kwok and Maksym, 2014; Kwok and Kacimi, 2018; Wang et al., 2020). Upward-looking sonars (ULS), located at 13 different sites in the Weddell Sea, provide valuable temporal evolution of sea ice draft (Harms et al., 2001; Behrendt et al., 2013a; Behrendt et al., 2013b), but a basin-wide spatial distribution cannot be derived. More recently, satellite remote sensing has been widely applied to investigate the spatial coverage and long-term trend of sea ice thickness in the whole Antarctic (e.g., Kurtz and Markus, 2012; Bernstein et al., 2015; Li et al. 2018). Passive microwave sensors are used to obtain thin ice thickness (basically below 20 cm) by retrieving the brightness temperature, and are effectively applied in coastal polynyas (Nihashi and Ohshima, 2015). Satellite altimetry, including radar and laser altimetry, have been used in the Antarctic to retrieve sea ice thickness (e.g., Giles et al., 2008; Zwally et al., 2008) and have proven to currently be the best source for Antarctic-wide sea ice thickness retrieval over the full thickness range.

Within the framework of the Sea Ice Climate Change Initiative (SICCI) project, radar altimeter data collected by European Space Agency (ESA) satellites over the past two decades have been reprocessed and assessed. Based on these data, a new SICCI sea ice thickness data set was released in 2018, including the two radar altimetry satellites, Envisat and CyroSat-2 (Hendricks et al., 2018a; Hendricks et al., 2018b). The SICCI product covers the entire Antarctic sea ice for the complete annual cycle from 2002 to 2017, and it is finally a combined data set of Envisat and CyroSat-2. Thickness retrieval from radar





altimetry is based on the assumption that the dominant source of radar backscatter is the snow/ice interface (Beaven et al., 1995) and sea ice freeboard is measured by differential ranging over sea ice and ocean surfaces, illustrated in Fig. 1. Snow affects the radar altimetry SIT retrievals in two ways. On the one hand, over Antarctic sea ice, the complex snow stratigraphy

and frequent snow flooding associated with the formation of snow ice and superimposed ice affect radar altimetry measurements (Willatt et al., 2010). On the other hand, the snow depth climatology used in the retrieval of Envisat and CryoSat-2 SIT can cause additional uncertainties due to neglecting inter-annual variability in snow depth (Bunzel et al., 2018). The SICCI Antarctic SIT data record has therefore been categorized as experimental data by the data producers compared to a more mature climate data record in the Arctic. Additional uncertainties of the radar altimeter range retrieval arise from the

surface type mixing (Schwegmann et al., 2016; Paul et al., 2018; Tilling et al., 2019) and surface roughness (Hendricks et al., 2010; Ricker et al., 2014; Landy et al., 2020). Due to the larger footprint compared to laser altimeters, radar altimeter measurements can be more affected by surface type mixing and surface roughness, which potentially introduces additional range biases.

The Geoscience Laser Altimeter System (GLAS) aboard the Ice, Cloud and land Elevation Satellite (ICESat-1, hereinafter

called 'ICESat') allows estimating the total freeboard via determination of the surface elevation from 2003 to 2009, illustrated in Fig. 1. This data set has been investigated for many years (e.g., Markus et al., 2011; Yi et al., 2011; Kurtz and Markus, 2012; Xie et al., 2013; Kern and Spreen, 2015). Several freeboard-to-thickness retrieval algorithms have been compared (Kern et al., 2016). In contrast to radar altimetry, laser altimetry has the advantage of a well-defined reflective horizon, which is the air/snow interface. The main deficiencies of ICESat data are data gaps due to cloud coverage, and more generally the discontinuous and

short observation periods. Therefore, ICESat data cannot reflect the current characteristics of the fast-changing Antarctic sea ice. However, ICESat-2, which has been in orbit since 2018, provides a new source of year-round observations of total freeboard (Kwok et al., 2019).

Both the Envisat and CryoSat-2 SIT in the Antarctic have already been evaluated with the drilling data, AEM, ULS and ship-based data (Kern et al., 2018). These evaluations are comprehensive but still have their limitations due to small spatial coverage

or short temporal coverage. Thus, we cannot achieve an overall understanding of the data quality in the Antarctic.

To get a better understanding of the characteristics of the SICCI product version 2.0, we aim to investigate how the SICCI Envisat retrieval compares to the ICESat sea ice thickness data record, also how the different retrieval methods are represented in the ice thickness distribution. Based on the former inter-comparison study, we choose the ICESat sea ice thickness data derived from the modified ice density approach suggested by Kern et al. (2016) for comparison, which seems to agree with

independent observations and has a reasonable winter-to-spring growth (Kern et al., 2016). Furthermore, in order to evaluate the ICESat and Envisat data records in the Weddell Sea, we also compare both sea ice thickness records with the Weddell Sea ULS data first.





The study is organized as follows. In section 2, we describe the data used in this study in detail. Section 3 presents the results of both the Weddell Sea ULS data validations and the inter-comparisons between the two data sets. Potential reasons for the spatial and temporal deviations are discussed in section 4. The main conclusions are summarized in section 5 with further discussions.

## 2 Data and methods

### 2.1 Sea ice thickness from Envisat / RA-2

The ESA CCI Sea Ice Climate Change Initiative (SICCI) provides a set of Antarctic sea ice freeboard and thickness data (Hendricks et al., 2018b) obtained from the satellite missions Envisat (2002–2012) and CryoSat-2 (2010–2017). With 50-km grid resolution and monthly temporal resolution, there is a successive year-round record for Antarctic sea ice freeboard and thickness on Equal-Area Scalable Earth (EASE) grid. Since only Envisat shares overlapping periods with ICESat, we focus on the measurement characteristics of the Envisat radar altimeter. Envisat was launched on 01 March 2002 and the mission ended on 08 April 2012. The Radar Altimeter 2 (RA-2) aboard on Envisat is a nadir-looking pulse limited sensor operating at the main frequency of 13.575 GHz (Ku-Band), with a secondary frequency of 3.2 GHz (S-Band) compensating the ionospheric error (Zelli and Aerospazio, 1999). It has an orbit inclination of 98.55° (Paul et al., 2017), covering the full Southern Ocean and nominal circular footprints of 2–10 km in diameter (Connor et al., 2009). Because the RA-2 is the only altimeter carried by Envisat, we refer to it as Envisat hereafter.

The Envisat radar freeboard is retrieved based on the radar range obtained from RA-2 Level-1 waveform data over ice surface and leads between ice floes. Ideally, the signal will return at the interface between snow and ice based on the experience from laboratory work (Beaven et al., 1995). Then, snow-depth dependent radar signal delay is applied to convert the radar freeboard into the sea-ice freeboard. The delay correction is based on a conventional assumption that has been revised (Mallett et al. 2019) since the generation of the SICCI data. The illustration of sea ice freeboard is shown in Fig. 1, which is the sea ice surface elevation relative to the sea surface elevation. Sea ice thickness is retrieved from ice freeboard based on the hydrostatic equilibrium approach as first used by Laxon et al. (2003):

$$I = \frac{F\rho_{water} + S\rho_{snow}}{\rho_{water} - \rho_{ice}} \qquad (1)$$

where $F$ represents Envisat sea ice freeboard, $S$ represents snow depth, $I$ represent ice thickness, $\rho_{water}$, $\rho_{snow}$, $\rho_{ice}$ refer to the density of the sea water, snow cover and sea ice, respectively. A snow depth climatology is employed to retrieve sea ice thickness from sea ice freeboard here. This snow-depth climatology is derived from Advanced Microwave Scanning Radiometer-EOS (AMSR-E) and AMSR-2 data for the Antarctic and is based on a revised version of the approach described by Cavalieri et al. (2014) and provided by the Integrated Climate Data Center (ICDC, http://icdc.cen.uni-hamburg.de).



In addition, it is noted that Envisat sea ice thickness represents the actual SIT (i.e., mean thickness of the ice-covered fraction of the grid cell area) and values with sea ice concentration less than 70 % have been removed during Envisat SIT retrieval.

**2.2 Sea ice thickness from ICESat / GLAS**

ICESat, operated as part of NASA's Earth Observing System, provides a set of Antarctic total freeboard data from 2003 to 2009. Different from the sea ice freeboard measured by radar altimeters, laser altimeters allow to detect the distance between the snow surface and sea surface, which is called total freeboard, as shown in Fig. 1. ICESat measurements are conducted with laser footprints of ~70 m and sampling distances of 170 m (Kwok et al., 2004). However, the measurements are not continuous due to cloud coverage and each measurement campaign lasts for about 35 days (see Fig. 3 in Kern and Spreen, 2015). There

are several ICESat sea ice thickness data sets derived from different retrieval algorithms. Qualitative inter-comparisons have been done among several ICESat freeboard-to-thickness retrieval approaches (Kern et al., 2016). According to the conclusions of Kern et al. (2016), we choose the product derived with the modified ice density approach in this study, because of its reasonable winter-to-spring increase and better agreements with independent data. The data set is provided by ICDC on the polar-stereographic grid. The approach considers the snow–ice layer as one system with a modified density, in order to avoid

using a potentially biased snow depth product. According to Kern et al. (2016), the modified density can be derived as follows:

$$\rho_{ice}^* = \frac{R\rho_{ice} + \rho_{snow}}{R + 1} \tag{2}$$

where $R$ is the ratio of sea ice thickness over snow depth, which is a seasonally dependent factor and calculated from ASPeCt observations (Worby et al., 2008a). And sea ice thickness can be determined from it:

$$I = F\frac{\rho_{water}}{\rho_{water} - \rho_{ice}^*} \tag{3}$$

where $F$ represents ICESat total freeboard.

The Antarctic mean gridded sea ice freeboard and effective sea ice thickness (i.e., mean thickness per grid cell including open water areas) with a grid resolution of 100 km from 2004 to 2008 are provided in this product. Table 1 presents the available time periods of the data. It is noted that grid cells with sea ice concentration less than 60 % have been removed for the ICESat SIT retrieval.

**2.3 Sea ice thickness from Weddell Sea ULS**

The upward-looking sonars (ULS) located in the Weddell Sea provide long-term and high-frequency sea ice draft at each site (Behrendt et al., 2013a; Behrendt et al., 2013b). The moorings are deployed at more than 900 m underwater, transmitting sound pulses upwards with a footprint of 6–8 m in diameter. The signals are reflected either by the sea ice bottom or the sea surface, yielding two distances based on the travel time. The sea ice draft, which is the depth of the sea ice underwater, can consequently





be derived from the difference of the two distances, shown in Fig.1. The measurements of sea ice draft were collected once several minutes from November 1990 to March 2008. In this study, we use the monthly average sea ice draft at three sites, corresponding to Envisat and ICESat operating time. According to Behrendt (2013), the uncertainty of sea ice draft varies from 5 cm to 12 cm, depending on the seasons. The uncertainty in summer is smaller than in other seasons because the frequent leads or open water in summer provides the benchmark for sea surface height calibration. The mooring locations used in this

study are shown in Fig. 2. We choose the sea ice draft measured from 2004 to 2008 to cover the ICESat periods in this study. Sea ice thickness (z) is converted from the sea ice draft (d) through an empirical formula established from drilling data in the Weddell Sea (Harms et al., 2001):

$$z \text{ (m)} = 0.028 + 1.012d \text{ (m)}. \tag{4}$$

This empirical equation is based on the assumption that the snow depth values from drillings and ULS are comparable. But it
still bears the uncertainties from the production of slush and snow ice caused by flooding (Harms et al., 2001). All the SIT data used in this study have been summarized in Table 2.

## 2.4 Meteorological data

In this study, we use 2-m air temperature data derived from ERA-5 reanalysis (Hersbach et al., 2019) to generate the accumulative freezing-degree-days (FDD). FDD is calculated by daily degrees below freezing summed over the total number
of days when the temperature was below freezing point. Here the freezing point is set to -1.8 degrees Celsius for ocean water.

We compare FDD with the SIT variations during FM-MJ and MJ-ON represented by Envisat and ICESat SIT. Note that FDD only accounts for the thermodynamic thickening and neglects ice growth from snowfall, freezing rain or ridging.

## 2.5 Spatial and seasonal divisions

The comparisons are realized by considering the differences between the two sea ice thickness data sets in different seasons
and different sectors. The seasonal classification is based on the ICESat operating periods presented in Table 1 following Kurtz and Markus (2012). For each period, we choose the corresponding time period during which Envisat monthly data are used, also given in Table 1. We employ a time-weighted average of the monthly Envisat data to match the ICESat period. For example, consider the ON04 period from Oct 3 to Nov 8 in 2004, which is 37 days long – 29 days in October and 8 days in November. So we calculate the corresponding Envisat SIT as: $SIT_{ON04} = (29/37)*(SIT_{October}) + (8/37)*(SIT_{November})$. The
weighting has taken into account periods where only Envisat SIT of one month are present, i.e., we use this equation for grid cells where we have valid SIT data from both months, while we only use the Envisat SIT of the respective month without weighing for those grid cells where we only have valid data from either month.



Antarctic sea ice characteristics differ remarkably in different areas in the Southern Ocean. Therefore, we divide it into six sectors (Fig. 2) following Worby et al. (2008a) and discuss the differences for each of them.

## 3 Results and discussion

### 3.1 Comparisons with Weddell Sea ULS

Before the inter-comparison between Envisat and ICESat SIT, both of them are compared with ULS observations first. The ULS sea ice draft has been converted into monthly sea ice thickness data with Eq. (4) in Sect. 2.3. Both Envisat and ICESat SIT have been interpolated onto each ULS location in the nearest neighbour way. We here compare Envisat and ICESat actual SIT with ULS observations. Therefore, we exclude zero thickness measured by ULS in the statistical calculations and divide ICESat SIT by the sea ice concentration contained in the data for each grid. The sea ice concentration data are derived from Special Sensor Microwave/Imager (SSM/I) and Special Sensor Microwave/Imager Sounder (SSM/IS) provided by ICDC (Kaleschke et al., 2001), interpolated to 100 km grid NSIDC polar-stereographic grid and averaged over respective ICESat measurement periods.

During ICESat operating periods, there are only three sites with valid data for the comparison: 207, 229 and 231 (see also Fig. 2). The sites can be divided into two regions. Site 207 is near the coast of the Antarctic Peninsula, mostly characterized by perennial ice, while the others belong to the eastern Weddell Sea, predominantly characterized by first-year ice. The corresponding time periods of each SIT product during the comparison are listed in Table 3.

Figure 3 presents the time series of sea ice thickness for Weddell Sea ULS, Envisat and ICESat at each site. Due to the operating period gaps and lack of valid data along the coast, ICESat only provides a limited number of measurements for comparison. The Envisat gaps originate from grid cells with sea ice concentration below 70%, or missing data caused by failure of retrieval or instrument. We provide error bars of uncertainty information contained in both SIT products. The Envisat SIT uncertainty is computed as the error propagation of all input uncertainties with the assumption that the sea water density is negligible (see Section 2.9.8 in Paul et al., 2017). ICESat SIT uncertainties are calculated based on the uncertainties of sea ice density and snow density, also neglecting the uncertainty of water density (see Eq. (6) in Li et al., 2018). It is noted that the corresponding time of each ICESat SIT point is placed between the two months that each period covers in Fig. 3. We find that both Envisat and ICESat SIT are not consistent with the sea ice thickness observed from ULS. In the western Weddell Sea, along the coast of the Antarctic Peninsula (at site 207), the ULS thickness ranges between 0 and 1.5 m, without a clear seasonal cycle. Envisat thickness exceeds ULS, with a maximum value larger than 5 m. The relatively large error bars only cover part of the observations. In comparison, ICESat thickness also exceeds ULS and the smaller error bars of ICESat also do not cover the observations. The large differences in the error bars between Envisat and ICESat mainly result from the inclusion of uncertainties of sea ice freeboard and snow depth in Envisat SIT, while these are not considered for ICESat SIT uncertainties.



In the eastern Weddell Sea (at sites 229 and 231), ICESat does not capture grid cells where the ULS ice thickness is smaller than 0.5 m, while having a few overestimations on thicker ice. In comparison, Envisat has larger overestimations, but the error
bars can cover almost all the observations. However, since many contributions are not well characterized and quantified, it is difficult to estimate realistic uncertainties. Table 4 shows the MD, SD and root mean square deviations (RMSD) for Envisat-ULS and ICESat-ULS. The Envisat and ULS SIT are all time-weighted processed and the calculations are conducted when all three products have valid data. The statistics show that the mean differences of ICESat-ULS and Envisat-ULS are all large at three sites. At site 207, the differences are 1.63 m for Envisat-ULS and 1.73 m for ICESat-ULS. The respective differences
are 0.72 m and 0.42 m at site 229, while at site 231 are 1.11 m and 0.55 m. However, the numbers of valid data are too small to derive a reliable conclusion on the accuracy of both products.

The uncertainties of such comparisons cannot be ignored. The ULS measurements are recorded at fixed locations with approximately 6–8 m footprint in diameter, while Envisat (ICESat) has a footprint of 2–10 km (70 m) and the SIT data used in the comparison represents mean values over 50 km (100 km) grid cells. The large resolution differences can increase the
selection biases. When the ULS measures a single point like a ridge or the edge of thin ice, satellites will survey a large area, which is represented in the SIT retrieval. In addition, though the ULS SIT and satellite SIT are all monthly mean values, one satellite SIT grid cell is scanned only once or twice through a month (see Fig. 3 in Kern and Spreen, 2015). Averages based on such a small measurement population has a limited representation of the mean SIT throughout the whole month. Theoretically, the more valid measurements exist in one grid cell, the more accurate the mean SIT is. In general, uncertainties
from both spatial interpolation and temporal representation can affect the comparisons. However, considering the typical sea ice motion (Drucker et al., 2011) in the Weddell Sea, monthly average ULS SIT could be referred to as a spatial average value, representing 100 km around the fixed ULS positions.

## 3.2 Inter-comparisons between Envisat and ICESat

We first conduct an overall comparison between Envisat and ICESat effective SIT for each ICESat operating period in each
season, as shown in Fig. 4–6. The inter-comparisons are carried out by linearly interpolating Envisat SIT onto the ICESat polar-stereographic grid with 100 km grid resolution. The figures suggest that there are substantial inter-seasonal and interannual differences between the two SIT data sets.

In spring (ON), both positive and negative differences exist, shown in Fig. 4. Envisat and ICESat SIT are both able to capture the thick ice located in the western Weddell Sea and the Bellingshausen and Amundsen Sea. Deformed sea ice near the coast
of the western Pacific Ocean is also detected by both sensors, but is thicker than the ship-based observations (Worby et al., 2008a). Envisat does not show the young ice in the Ross Sea (Kern et al., 2018) while the Ross Ice Shelf polynya (which is indicated in Fig. 2) is present in the ICESat maps. The same feature is found for the Ronne Ice Shelf polynya in the Weddell Sea in 2007. There is a fringe with no data along most of the East Antarctic coast, which indicates that the 100 km ICESat



product possibly fails to see the sea ice close to the coast. Table 5 provides the respective SIT and their SDs, differences,

RMSDs, correlation coefficients (CCs) and the numbers of comparison pairs. In general, the difference between Envisat and ICESat spring SIT is close to zero, ranging from -0.16 m in 2006 to +0.10 m in 2007. However, some of the differences between Envisat and ICESat SIT may not be significant, because the SDs are ~0.6 m while the differences are near zero. The RMSD for seasonal average comparison is the smallest by 0.39 m and the correlation coefficient is 0.68, with the significance larger than 95%. Note that we calculate the seasonal mean values only with valid data that are available for at least three years.

At the end of summer melt (FM), the ice coverage is limited to the western Weddell Sea, Bellingshausen and Amundsen Sea along the coast and southern Ross Sea (Fig. 5). In the western Weddell Sea, Envisat shows that thick ice still exists and remains at least 3 m, while ICESat shows thinner ice. As for the Ross Ice Shelf polynya, ICESat displays thin ice lower than 1 m there in 2004, 2007 and 2008, while in 2005 and 2006 the data is missing due to low ice concentration. Envisat detects sea ice in the Ross Sea all the years, but with thickness estimates of up to 1.5 m. According to Table 5, the numbers of comparison pairs are

small. Generally, Envisat SIT exceeds ICESat SIT by 0.52 m in summer, with the largest RMSD by 0.68 m and the smallest correlation values by 0.40 among the three seasons.

In autumn (MJ), the thickness patterns of the two data sets are comparable, shown in Fig. 6. The differences between Envisat and ICESat SIT are consistently positive over all regions except some regions in the Eastern Antarctic, probably inferring that ICESat is more sensitive to thick ice than Envisat, i.e., higher spatial resolution is better able to resolve thick ice. Compared

to summer, the differences in the western Weddell Sea spread to the whole Weddell Sea sector and decrease from west to east. In addition, the positive differences in the Ross Ice Shelf polynya still exist, mostly due to Envisat's inability to capture the thin ice there (Comiso et al., 2011; Tian et al., 2020), which has been pointed out in Kern et al. (2018) identifying a substantial SIT difference between Envisat and CryoSat-2 in that region. According to Table 5, the mean difference in autumn is the largest by 0.57 m, with RMSD by 0.47 m and the largest CC by 0.71.

To investigate the development of sea ice thickness from two satellites from the end of melting to end of freezing, we provide the probability distribution of the Envisat SIT and the ICESat SIT for all the valid individual comparison pairs, shown in Fig. 7. The mean and modal ice thickness of both data sets are marked beside. In summer, Envisat has a positive difference with respect to ICESat, mainly due to their different performances towards thick ice above 3 m. Envisat presents a larger mean and modal thickness than ICESat in autumn with similar distributions. In spring, the two data sets coincide with each other,

represented by closer mean and modal thicknesses. We find that ICESat mean SIT increases while Envisat mean SIT decreases from autumn to spring. For Envisat SIT, the distribution indicates that more thin ice is present in spring than autumn, while for ICESat SIT more thick ice is found. Therefore, we further compare the SIT variations from summer (FM) to spring (ON) with the FDD results in 2004, 2005 and 2006, shown in Fig. 8 and Table 6. As introduced in Sect. 2.4, this indicator only represents thermodynamic growth and neglects any dynamic contributions to sea ice thickness. We calculate the period-

average SIT from the model corresponding to ICESat operating periods for the same spatial coverage. The results show that



Envisat SIT has opposite developments from ICESat and FDD during MJ-ON while shares similar variations during FM-MJ. Both products present decreasing SIT in Ronne Ice Shelf polynya and increasing SIT in the northwest Weddell Sea from FM to MJ. This pattern is probably formed through thin ice produced in the Ronne Ice Shelf polynya combined with the typical sea ice drift in the Weddell Sea (Drucker et al., 2011). The adverse patterns from MJ to ON infer that a lot of SIT changes
during this time are caused by deformation processes and snow-ice formation, and both products have different abilities to reveal them.

Figure 9 presents scatterplots of the individual data pairs between Envisat SIT and ICESat SIT for each region and each season. The respective correlation coefficients are indicated in the panels. Due to the limited measurements in the Indian Ocean and western Pacific Ocean, we combine them into the whole Eastern Antarctic. For all five panels, the regression lines have large
positive intercepts in all three seasons, indicating that Envisat SIT tends to exceed ICESat SIT for thin ice. From Fig. 9a, we can see that in the western Weddell Sea, the summer and autumn sea ice thickness splashes exceed the spring ones. This reveals that mean ICESat SIT are nearly constant through all three seasons in the western Weddell Sea, while mean Envisat SIT are noticeably larger in summer and autumn, which can also be seen from the statistical results shown in Table 7. From FM to ON, Envisat SIT changes from 3.01 m to 3.18 m to 2.23 m, while ICESat SIT changes from 2.04 m to 2.28 m to 2.23 m. The
numbers in the last column of the table are the weighted average, taking into account the actual number of values per season. Considering the regional average differences between Envisat and ICESat SIT, the largest (0.63 m) is found in the western Weddell Sea and the smallest is in the Bellingshausen and Amundsen Sea (0.09 m). Differences are small in spring for most regions except Eastern Antarctic, just as the seasonal results shown in Table 5. Regarding the spatial correlation, the largest coefficients are found in Bellingshausen and Amundsen Sea (0.54) and the smallest is in the western Weddell Sea and Ross
Sea by 0.36.

## 4 Potential reasons for the differences

There are two main differences between the two data sets. One is the different altimeter sensors to determine surface elevation and freeboard. Envisat is equipped with a Ku-Band radar altimeter (RA-2), whose backscatter is assumed to originate from snow/ice interface, though it is known that this assumption is flawed for snow that is not dry, cold and with a homogenous
stratigraphy. Instead, ICESat is equipped with a laser altimeter (GLAS) whose signals are reflected from the air/snow interface. In addition, considering the large pulse-limited footprint of about 2–10 km and smaller footprint of laser beams of about 70 m, there are very likely differences in the ability to resolve leads or open water required for an adequate representation of the local sea-surface height during the freeboard retrieval, as well as in the accurate representation of heterogenous sea ice surfaces. The other difference is that they apply different retrieval algorithms to convert freeboard into thickness. Envisat directly uses
the hydrostatic equilibrium with an extra AMSR-E snow depth climatology product, while ICESat also uses the hydrostatic





equilibrium but accompanied with a modified snow–ice density method to reduce the influence of the often regionally biased snow depth product. The effects of these differences between both products are discussed in the following.

**4.1 Differences due to sensors**

It is assumed that the dominant backscatter horizon for a Ku-Band radar altimeter is the snow/ice interface for cold and dry
snow (Beaven et al., 1995). However, this would not always be the case in the Antarctic according to the field investigations conducted by Willatt et al. (2010). They demonstrate that the dominant scattering surface of the Ku-Band radar lies within the snowpack, usually half of the mean snow depth, when the snow cover is not cold and dry. Wet conditions can affect the dielectric properties of snow and then decrease the penetration of radar altimeter signals into the snow. Another common property of snow in the southern hemisphere occurs when snow cover gets thicker and depresses sea ice below the sea surface,
seawater intrudes towards the base of the snow layer and before the slush freezes, the existence of wet snow will cause the extinction of signals from ice layer (Massom et al., 2001). Consequently, RA-2 range measurements could be biased high when the main scattering horizon is located within the snow pack, above the snow/ice interface, which will lead to larger sea ice freeboard and larger sea ice thickness. Salinity of the basal snow layer also contributes to this effect (Nandan et al., 2017). The existence of this bias is also shown by Kwok and Kacimi (2018), where they find that radar freeboards from CryoSat-2
are consistently higher than those computed using Operation Ice Bridge (OIB) measurements. Other studies that utilize CryoSat-2 radar data in the southern hemisphere have thus explicitly incorporated radar backscatter from the snow layer into their freeboard retrieval method (Fons and Kurtz, 2019).

Assuming the total freeboard derived from ICESat and the snow depth product used in the retrieval of Envisat SIT as a reference, we can calculate the sensitivity of Envisat SIT ($I$) to the sea ice freeboard ($F$) caused solely by the offset of the radar
main reflecting horizon from Eq. (1):

$$\frac{dI}{dF} = \frac{\rho_{water}}{\rho_{water} - \rho_{ice}} \tag{5}$$

We set $\rho_{water}$ to 1024 kg m$^{-3}$ and $\rho_{ice}$ to 916.7 kg m$^{-3}$. If the sea ice freeboard is overestimated by 7 cm (which is the nominal adjustment suggested by Nandan et al. (2017, 2020)) then the SIT is overestimated by 67 cm, inferring that sea ice thickness change is sensitive to the sea ice freeboard change. This could potentially account for the differences between Envisat and
ICESat SIT in summer and autumn. However, detailed sensitivity discussions are limited due to lack of the seasonal and regional sea ice density and adjustments to sea ice freeboard.

In addition to the different penetration depths, the two sensors have different footprints. Envisat is a pulse-limited radar altimeter with a large footprint of 2–10 km and ICESat is a laser altimeter with a small footprint of about 70 m. This makes them sensitive to a relative selection bias, primarily on the side of the altimeters with the lower resolution in the case of surface-
type mixing within the footprint, especially if the different surface types vary in their backscatter properties. Higher spatial





resolution will mitigate this issue and subsequently allow a better classification of lead and sea surface height retrieval in principle. Several studies have pointed that Envisat freeboard and thickness uncertainties are elevated with respect to other sensors due to sub-footprint scale surface type mixing (Schwegmann et al., 2016; Paul et al., 2018; Tilling et al., 2019). While this is also directly applicable to radar altimeters with different footprints, the difference of lead detection skill between laser

(ICESat) and radar (Envisat) altimeters is not directly a function of footprint size but also of altimeter concept. Leads amplify radar backscatter and thus cause over-representation, while the laser backscatter is a function of the surface albedo thus leads return lower laser backscatter power. For different ice surfaces, however, the smaller footprint of ICESat has the capability to provide more detailed observations in areas with heterogenous ice conditions than the pulse-limited Envisat footprint.

## 4.2 Differences due to AMSR-E snow depth

According to the retrieval algorithms applied by the two data sets described in Sect. 2, one of the largest uncertainties may come from the biased AMSR-E snow depth. AMSR-E snow depth is retrieved from brightness temperature based on the linear relation between brightness temperatures and in situ observations (Markus and Cavalieri, 1998; Comiso et al., 2003). According to Markus and Cavalieri (1998), their AMSR-E snow depth product masks out perennial ice and is limited to the maximal retrieval value being around 50 cm because of the similar radiometric signature of deep snow and multiyear ice

(Comiso et al., 2003). Previous study reveals that AMSR-E snow depth tends to considerably underestimate the actual value over deformed sea ice, which usually occurs in the Eastern Antarctic (Worby et al., 2008b; Ozsoy-Cicek et al., 2011). According to Frost et al. (2014), the differences that AMSR-E snow depths minus the ASPeCt observations are positive for snow below 15 cm and negative for snow above 30 cm. Environmental conditions have great effects on the snow physical properties such as density, wetness and salinity and the satellite passive microwave snow depth is sensitive to ice concentration

errors, weather effects, grain size, thaw and refreezing (Markus and Cavalieri, 1998). Especially, wet snow caused by melt or flooding could lead to underestimations while refreezing of wet snow could lead to overestimations. All of the above biases can affect the differences between Envisat and ICESat.

Assuming the sea ice freeboard derived from Envisat as a reference, we can calculate the required snow depth $\Delta S$ for compensating the difference between Envisat and ICESat SIT ($\Delta I$). According to Eq. (1), we can derive that:

$$\frac{dI}{dS} = \frac{\rho_{snow}}{\rho_{water} - \rho_{ice}} \qquad (6)$$

Setting $\rho_{snow} = 300$ kg m$^{-3}$, we can infer that if the snow depth is overestimated by 2 cm (which is the average monthly retrieval uncertainty according to Kern et al. (2015)) then the SIT is overestimated by 5.6 cm, suggesting that sea ice thickness change is insensitive to the snow depth. Therefore, one of the causes of differences is the bias of snow depth products, albeit not the dominating one.





The largest effect might not come from the impact of ice deformation on the snow-depth retrieval but might be due to the difference between actual snow depth from that represented by the climatology. In this study, the snow depth climatology is employed during Envisat SIT retrieval processes, which neglects the inter-annual snow variability. According to Bunzel et al. (2018), the impact of using a snow climatology is small when the snow depth is thin. The usage of snow climatology allows reducing the relative uncertainties compared with using the actual snow depth values. The latter ones are affected by many factors as discussed above and have large uncertainties. While a snow climatology allows reducing relative uncertainties, it can also have an adverse effect when being constructed from biased snow depth data, as is the case for the climatology used for the Envisat SIT data.

In general, passive microwave snow depth is valid over level ice. During FM, snow is deep, potentially wet and/or metamorphous on thick ice, causing substantial difficulties for radar altimeters. And for the same reasons, passive microwave snow depth is possibly underestimated on thick ice during FM but also during the other ICESat periods. These snow depths also underestimate actual snow depth over deformed ice mostly during ON in Eastern Antarctic, Bellingshausen and Amundsen Seas.

### 4.3 Differences due to ICESat uncertainties

Although we mainly discuss the possible biases of Envisat and AMSR-E above, it must be taken into account that also estimates from ICESat are subject to uncertainties. ICESat freeboard can be biased at the locations where the geoid or the sea surface height are inaccurate or where the elevation measurements are affected by ocean swell (Kern et al., 2016). Consequently, the total freeboard retrieved from ICESat has an uncertainty up to 0.1 m (Kern and Spreen, 2015). In addition, the modified density used in the Worby retrieval algorithm does not consider the small-scale or regional variability of the snow depth. Instead, only a seasonal constant density derived from the ASPeCt observations is given. Therefore, the largest uncertainty of ICESat comes from the potential underestimations of the sea ice and snow observations for the computation of the apparent ice density, representing the bulk density of the ice-snow column (Kern et al., 2016). This bias has been modified in Li et al. (2018), who derived first guess values of snow depth and sea ice thickness directly from ICESat data with empirical approaches, instead of the observation climatology used by Kern et al. (2016). This product has been demonstrated to be more reasonable than ICESat SIT used in this study because it takes into account local conditions. However, it is noted that the empirical approaches used by Li et al. (2018) were developed from a suite of historic in-situ observations of freeboard, snow depth and sea-ice thickness which in a way have the character of a climatology as well.



## 5 Summary

Antarctic sea ice thickness derived from Envisat RA-2 is firstly compared with ICESat sea ice thickness retrieved from a
modified ice density algorithm. Envisat and ICESat sea ice thickness has been compared with ULS SIT in the Weddell Sea,
with the mean differences and the standard deviations of 1.29 (0.65) m for Envisat-ULS while 1.11 (0.81) m for ICESat-ULS,
respectively. The results show that both Envisat and ICESat measurements are not comparable to the ULS sea ice thickness.
Then a systematic comparison between the two data sets has been carried out for three seasons except winter, based on the
ICESat operating periods. According to the results, the differences between Envisat and ICESat sea ice thickness are different
considering different seasons, years and regions. More specifically, we find the smallest monthly average difference (SD in
brackets) of 0.00 m (0.39 m) between Envisat SIT minus ICESat SIT in spring, while larger differences (SD) exist in summer
and autumn by 0.52 m (0.68 m) and 0.57 m (0.45 m), respectively. In spring, ICESat SIT maps reveal the Ross Ice Shelf
polynya, while it is not present in the Envisat data. In summer, Envisat shows that thick ice still exists in the western Weddell
Sea and remains at least 3 m thick through all the years, while ICESat shows thinner ice. Compared to summer, the differences
in the western Weddell Sea spread to the whole Weddell Sea sector and slightly decrease from west to east in autumn. From
the probability distribution, it is noted that Envisat and ICESat have different SIT variations from autumn to spring, i.e., ICESat
SIT increases while Envisat SIT does not, but share similar SIT growth from summer to autumn. Compared to the FDD results
in 2004, 2005 and 2006, we infer that thermodynamic growth dominates during FM-MJ and deformation processes dominate
during MJ-ON. With respect to different sea sectors, the regional mean differences (SD in brackets) are 0.63 m (0.91 m) in the
Western Weddell Sea, 0.34 m (0.58 m) in the Eastern Weddell Sea, 0.31 m (0.57 m) in the Ross Sea, -0.12 m (0.69 m) in the
Eastern Antarctic, and 0.09 m (0.71 m) in the Bellingshausen and Amundsen Sea. The potential overestimation of sea ice
freeboard caused by range biases accounts for much of the differences between Envisat and ICESat SIT in summer and autumn,
while the biases of snow depth are not the dominant cause of the differences.

Through the study, we acknowledge that there are differences between Envisat and ICESat sea ice thickness, which potentially
result from the uncertainties of both data sets. There is still more work to be done in the future to make better use of remote-
sensed sea ice thickness such as assimilating the Antarctic sea ice thickness observations and analyzing the sea ice volume
variations.

*Data availability.* The Envisat sea ice thickness data are available
at https://dx.doi.org/10.5285/b1f1ac03077b4aa784c5a413a2210bf5 (Hendricks et al., 2018b). The ICESat-1 sea ice thickness
data are available at http://icdc.cen.uni-hamburg.de/1/projekte/esa-cci-sea-ice-ecv0/esa-cci-data-access-form-antarctic-sea-
ice-thickness.html (Kern et al., 2016). The Weddell Sea upward looking sonar data are available
at http://doi.pangaea.de/10.1594/PANGAEA.785565 (Behrendt et al., 2013a; Behrendt et al., 2013b). The 2 meter air
temperature data are available at https://www.ecmwf.int/en/forecasts/datasets/reanalysis-datasets/era5.



*Author contributions.* JW, QY and QS developed the concept of the paper. JW analyzed the data and wrote the manuscript.

CM, RR, QY, QS, BH and SH assisted during the writing process.

*Competing interests.* The authors declare that they have no conflict of interest.

*Acknowledgments.* This study is supported by the National Key R&D Program of China (Grant No. 2019YFC1509102), the National Natural Science Foundation of China (No. 41941009, 41922044), the Fundamental Research Funds for the Central Universities (No. 19lgzd07). The authors would like to thank Alfred Wegener Institute Helmholtz Centre for Polar and Marine

Research for providing the Weddell Sea upward looking sonar data, the Integrated Climate Data Center at the University of Hamburg for providing the ICESat-1 sea ice thickness data and European Centre for Medium-Range Weather Forecasts for providing ERA-5 2 meter air temperature data.

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





**Figure 1:** An illustration of measuring freeboard using ICESat and Envisat, and the ULS measurement principle.



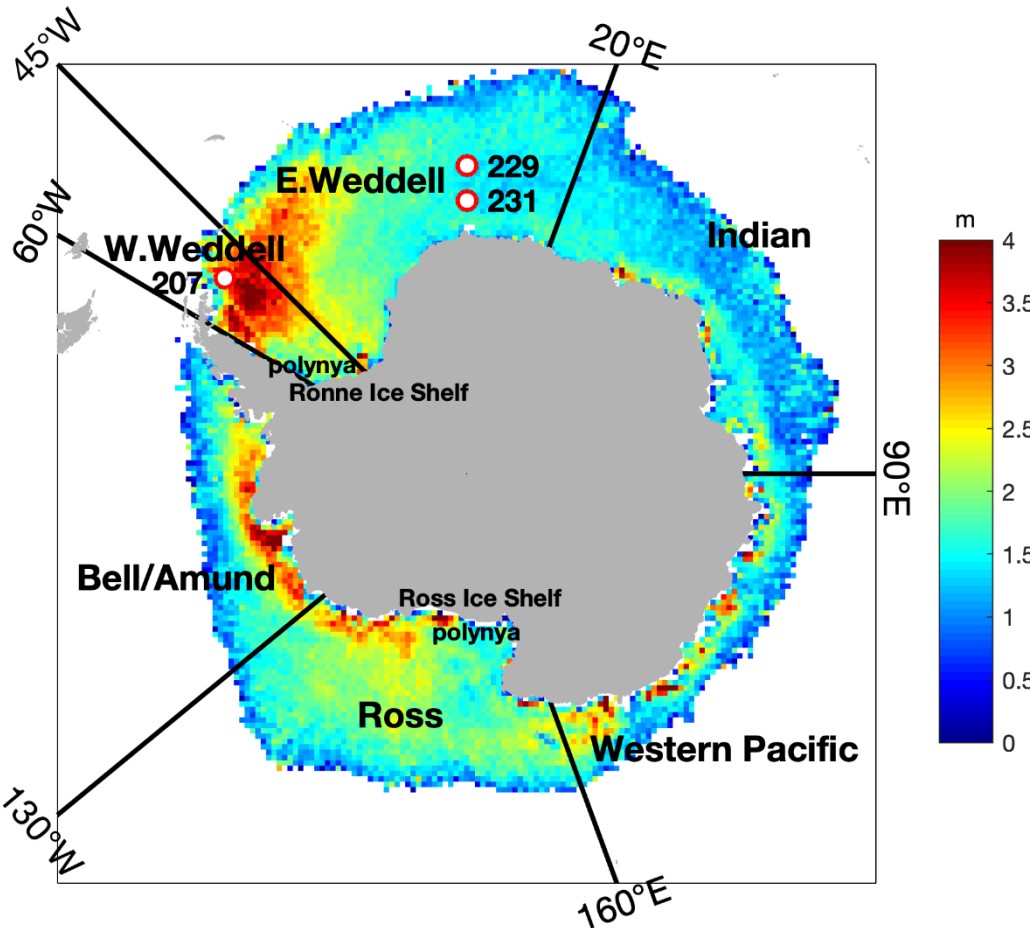

**Figure 2:** Map of the different sectors referred to in the study. The background is the average of the September sea ice thickness from Envisat
during 2003-2011 with 50 km grid size. Each sector and the two ice shelf polynyas are indicated in the figure. The circles and the
corresponding numbers refer to the sites of the ULS. The white grid cells stand for area with sea ice concentration less than 70% or missing
data.





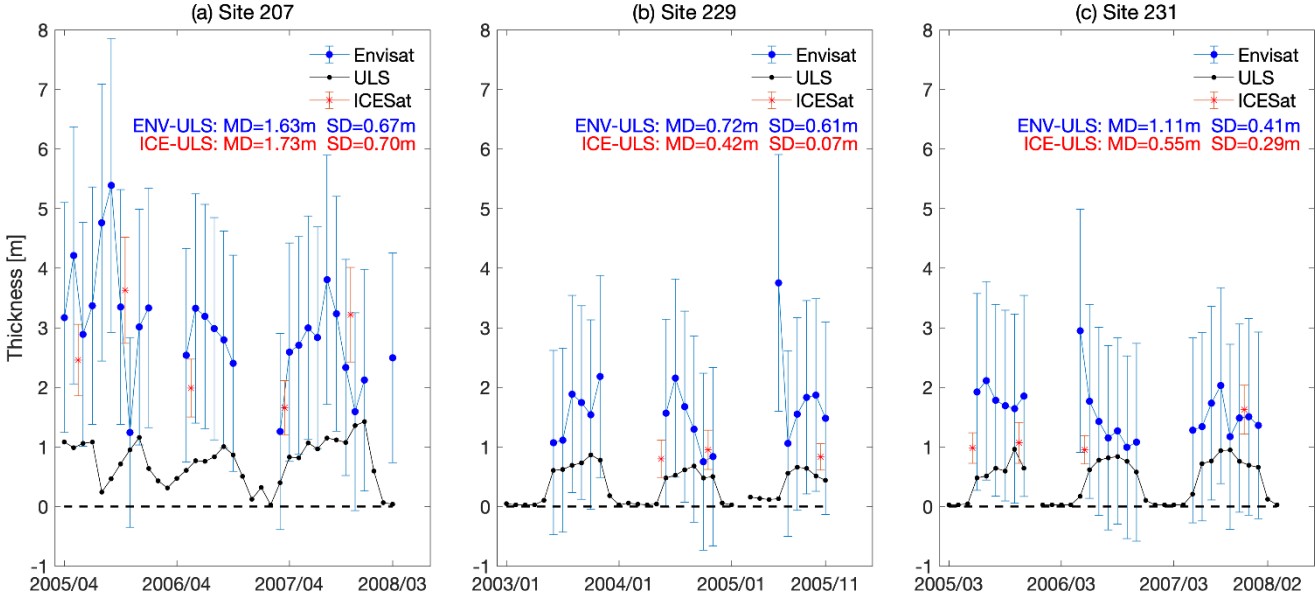

**Figure 3:** Time series of sea ice thickness for the Weddell Sea ULS data, Envisat and ICESat. The numbers on the top represent the location
of each site for the comparisons. The site locations can be searched in Fig. 2. ICESat SIT values are placed between the two months that
each period covers. The mean differences (MD) and their standard deviations (SD) are shown in the figures.


**Figure 4:** Comparisons of Envisat versus ICESat sea ice thickness for each ICESat operating period in spring (October & November). The first and second columns show the sea ice thickness distribution of Envisat and ICESat respectively, and the last column shows the difference map (Envisat minus ICESat) of sea ice thickness. Each row represents a year from 2004 to 2007. The sea ice thickness maps are at their native grid resolution while the difference map is interpolated onto the polar-stereographic grid of the ICESat product. The white cells denote sea ice concentration less than threshold or missing data.




**Figure 5:** Same as Fig. 4 but for summer (February & March).





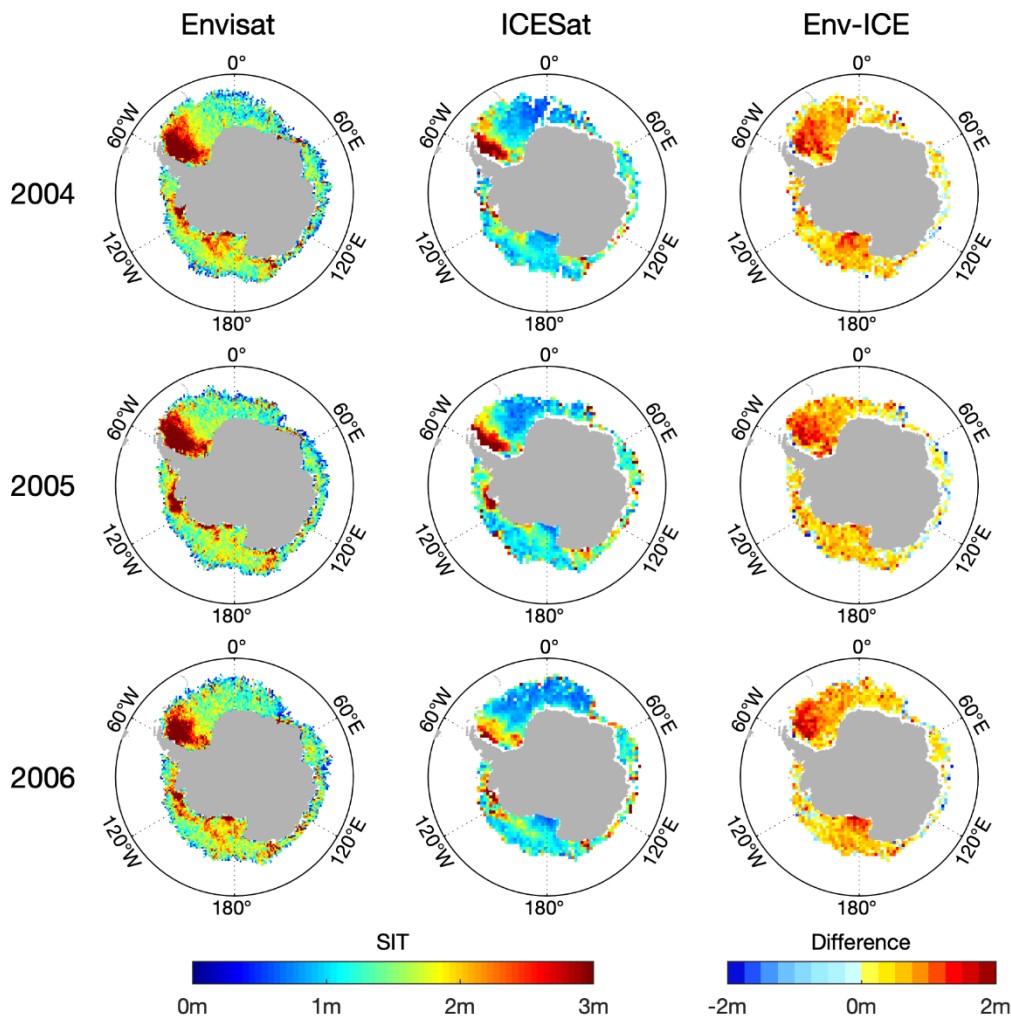


**Figure 6:** Same as Fig. 4 but for autumn (May & June).





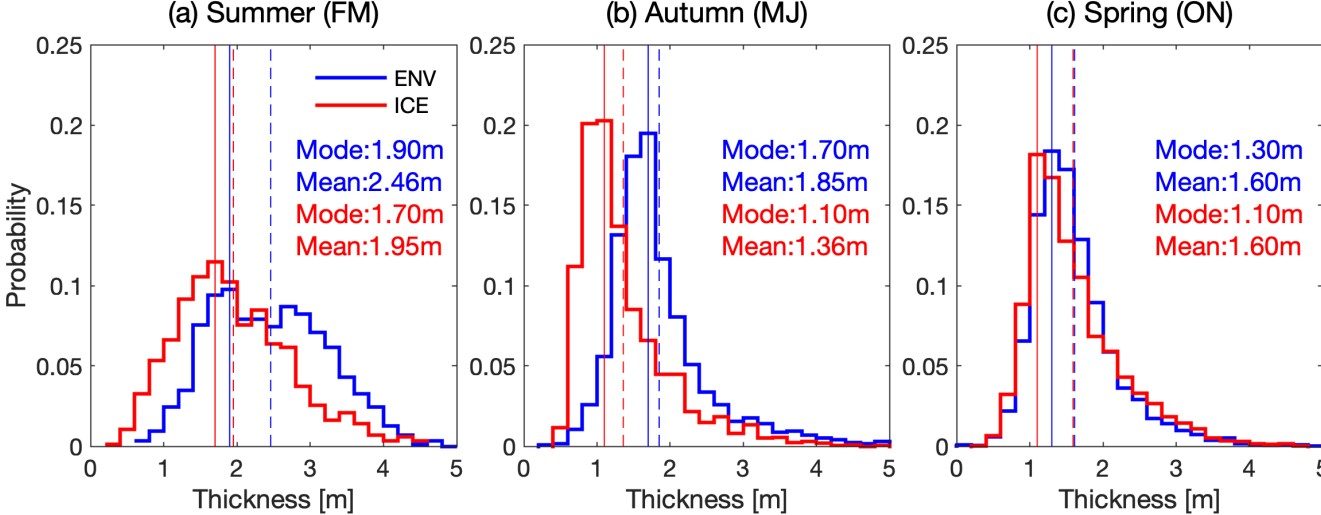

**Figure 7:** Probability of the Envisat SIT and the ICESat SIT for all the individual comparison pairs. The blue stairs represent Envisat ice thickness and the red stairs represent ICESat ice thickness. The solid lines indicate the modal ice thickness and the dashed lines indicate the mean ice thickness of both data sets.

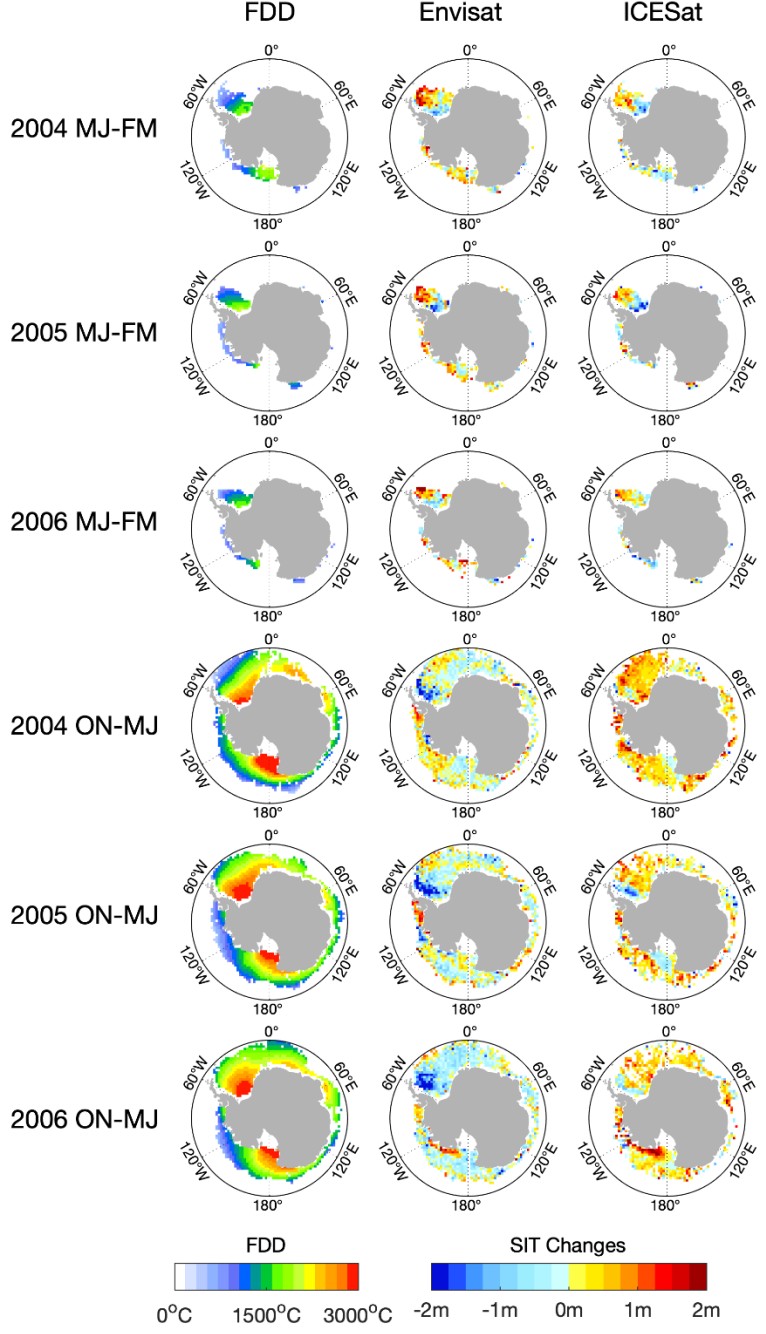

Figure 8. The FDD differences and sea ice thickness differences from summer to autumn (MJ-FM) and from autumn to spring (ON-MJ) derived from Envisat and ICESat in 2004, 2005 and 2006. Note that while the FDD represents thermodynamic ice growth, the right two columns simply show a difference in two SIT maps.




**Figure 9:** Scatterplots of the individual data pairs between Envisat SIT and ICESat SIT for each region and each season. Since the comparison pairs are too few in Indian Ocean and western Pacific Ocean, we combine these two regions into Eastern Antarctic. The respective correlation coefficients and RMSDs are indicated in the panels.


**Table 1**: ICESat operating periods and Envisat periods used for the comparisons. The three seasons are divided according to the ICESat operating periods. Note that ON is Oct-Nov, FM is Feb-Mar, and MJ is May-June.





| Years | Summer (FM) | | Autumn (MJ) | | Spring (ON) | |
|---|---|---|---|---|---|---|
| | ICESat | Envisat | ICESat | Envisat | ICESat | Envisat |
| 2004 | Feb 17 to Mar 20 | Feb 1 to Mar 31 | May 18 to Jun 20 | May 1 to Jun 30 | Oct 3 to Nov 8 | Oct 1 to Nov 31 |
| 2005 | Feb 17 to Mar 22 | Feb 1 to Mar 31 | May 20 to Jun 22 | May 1 to Jun 30 | Oct 21 to Nov 23 | Oct 1 to Nov 31 |
| 2006 | Feb 22 to Mar 26 | Feb 1 to Mar 31 | May 24 to Jun 25 | May 1 to Jun 30 | Oct 25 to Nov 26 | Oct 1 to Nov 31 |
| 2007 | Mar 12 to Apr 14 | Mar 1 to Apr 30 | – | – | Oct 2 to Nov 4 | Oct 1 to Nov 31 |
| 2008 | Feb 17 to Mar 20 | Feb 1 to Mar 31 | – | – | – | – |


**Table 2**: A summary of the sea ice thickness data used during the comparison, including different data sources, spatial resolution, temporal resolution and snow product.

| Source | Instrument | Operation time | Footprint | Grid resolution | Temporal resolution | Snow product |
|---|---|---|---|---|---|---|
| Envisat satellite | Radar altimeter | 2002-2011 | 2–10 km | 50 km grid | Monthly average | AMSR-E climatology |
| ICESat satellite | Laser altimeter | 2003-2009 | 70 m | 100 km grid | See Table 1 | ASPeCt observations |
| Weddell Sea ULS | Upward Looking Sonars | 1990-2010 | 6–8 m | Single point | Monthly average | built into Eq. (4) |


**Table 3:** Respective operation times of the ULS, Envisat and ICESat sea ice thickness data set during the comparison between ULS and satellite SIT.





|  | ULS | ENV | ICE |
|---|---|---|---|
| Site 207 | Apr 2005 to Mar 2008 | Apr 2005 to Mar 2008 | MJ05 to FM08 |
| Site 229 | Jan 2003 to Nov 2005 | Jan 2003 to Nov 2005 | FM04 to ON05 |
| Site 231 | Mar 2005 to Feb 2008 | Mar 2005 to Feb 2008 | FM05 to ON07 |

**Table 4:** Statistical results of the comparison between two satellite SIT data with ULS data.

| Site | Envisat-ULS | | | ICESat-ULS | | | N |
|---|---|---|---|---|---|---|---|
|  | MD (m) | SD (m) | RMSD (m) | MD (m) | SD (m) | RMSD (m) |  |
| 207 | 1.63 | 0.67 | 0.60 | 1.73 | 0.70 | 0.62 | 5 |
| 229 | 0.72 | 0.61 | 0.43 | 0.42 | 0.07 | 0.05 | 2 |
| 231 | 1.11 | 0.41 | 0.33 | 0.55 | 0.29 | 0.24 | 3 |
| Average | 1.29 | 0.65 | 0.62 | 1.11 | 0.81 | 0.77 | 10 |

**Table 5**: Statistical results of the comparisons between Envisat sea ice thickness and ICESat sea ice thickness for each ICESat operating period. The correlation coefficients (CC) in italic type have not passed the 95% significance test.

|  |  | ENV(SD) (m) | ICE(SD) (m) | Difference (SD) (m) | RMSD (m) | CC | N |
|---|---|---|---|---|---|---|---|
|  | Seasonal average | 2.51(0.66) | 1.99(0.58) | 0.52(0.68) | 0.68 | 0.40 | 170 |
| Summer (FM) | 2004 | 2.56(0.76) | 2.00(0.79) | 0.56(0.77) | 0.77 | 0.51 | 179 |
|  | 2005 | 2.82(0.82) | 2.35(0.82) | 0.47(0.85) | 0.84 | 0.47 | 139 |
|  | 2006 | 2.47(0.69) | 2.07(0.74) | 0.40(1.02) | 1.02 | *0.00* | 122 |
|  | 2007 | 2.16(0.76) | 1.69(0.80) | 0.47(0.88) | 0.88 | 0.36 | 236 |





|  |  |  |  |  |  |  |  |
|---|---|---|---|---|---|---|---|
|  | 2008 | 2.45(0.82) | 1.87(0.61) | 0.58(0.92) | 0.91 | *0.21* | 185 |
| Autumn (MJ) | Seasonal average | 1.92(0.65) | 1.35(0.55) | 0.57(0.45) | 0.47 | 0.71 | 735 |
|  | 2004 | 1.87(0.70) | 1.33(0.62) | 0.54(0.58) | 0.58 | 0.61 | 887 |
|  | 2005 | 1.88(0.76) | 1.42(0.68) | 0.46(0.66) | 0.66 | 0.58 | 903 |
|  | 2006 | 1.81(0.61) | 1.32(0.58) | 0.49(0.62) | 0.62 | 0.46 | 911 |
| Spring (ON) | Seasonal average | 1.62(0.48) | 1.62(0.50) | 0.00(0.39) | 0.39 | 0.68 | 886 |
|  | 2004 | 1.63(0.60) | 1.65(0.67) | -0.02(0.60) | 0.60 | 0.57 | 1057 |
|  | 2005 | 1.59(0.60) | 1.53(0.65) | 0.06(0.62) | 0.62 | 0.51 | 888 |
|  | 2006 | 1.48(0.54) | 1.64(0.66) | -0.16(0.58) | 0.58 | 0.55 | 828 |
|  | 2007 | 1.67(0.59) | 1.57(0.59) | 0.10(0.58) | 0.58 | 0.52 | 1124 |


**Table 6**: Statistical results of the FDD differences and sea ice thickness differences during FM-MJ and MJ-ON in 2004, 2005 and 2006.

|  | Envisat SIT (m) | ICESat SIT (m) | FDD (°C) | N |
|---|---|---|---|---|
| ON04-MJ04 | -0.11(0.59) | 0.47(0.55) | 1867.3(610.5) | 764 |
| MJ04-FM04 | 0.17(0.79) | -0.16(0.77) | 1266.5(410.5) | 176 |
| ON05-MJ05 | -0.22(0.70) | 0.22(0.69) | 1986.1(597.8) | 649 |
| MJ05-FM05 | 0.19(0.92) | -0.15(0.92) | 1273.9(348.5) | 136 |
| ON06-MJ06 | -0.33(0.65) | 0.35(0.64) | 1983.7(547.2) | 659 |
| MJ06-FM06 | 0.08(0.77) | -0.07(0.78) | 1224.6(340.1) | 118 |





**Table 7:** Statistical results of the comparisons between Envisat sea ice thickness and ICESat sea ice thickness for each region divided as Fig. 9.

|  |  | ENV(SD) (m) | ICE(SD) (m) | Difference(SD) (m) | RMSD (m) | CC | N |
|---|---|---|---|---|---|---|---|
| W. Weddell | Regional average | 2.80(0.87) | 2.17(0.72) | 0.63(0.91) | 0.91 | 0.36 | 892 |
|  | summer (FM) | 3.01(0.67) | 2.04(0.66) | 0.97(0.77) | 0.77 | 0.32 | 329 |
|  | autumn (MJ) | 3.18(0.88) | 2.28(0.74) | 0.90(0.77) | 0.77 | 0.57 | 263 |
|  | spring (ON) | 2.23(0.75) | 2.23(0.76) | 0.00(0.84) | 0.84 | 0.38 | 300 |
| E. Weddell | Regional average | 1.69(0.59) | 1.35(0.54) | 0.34(0.58) | 0.58 | 0.46 | 2405 |
|  | summer (FM) | 2.45(0.77) | 1.87(0.70) | 0.58(0.74) | 0.74 | 0.50 | 210 |
|  | autumn (MJ) | 1.76(0.51) | 1.08(0.40) | 0.68(0.45) | 0.44 | 0.55 | 921 |
|  | spring (ON) | 1.51(0.48) | 1.46(0.49) | 0.05(0.48) | 0.48 | 0.51 | 1274 |
| Eastern Antarctic | Regional average | 1.45(0.59) | 1.57(0.69) | -0.12(0.69) | 0.69 | 0.42 | 1535 |
|  | summer (FM) | 2.20(0.84) | 2.36(1.05) | -0.16(0.98) | 0.97 | 0.49 | 81 |
|  | autumn (MJ) | 1.55(0.53) | 1.49(0.61) | 0.06(0.71) | 0.71 | 0.23 | 521 |
|  | spring (ON) | 1.32(0.52) | 1.55(0.65) | -0.23(0.63) | 0.63 | 0.44 | 933 |
| Ross Sea | Regional average | 1.72(0.45) | 1.41(0.55) | 0.31(0.57) | 0.57 | 0.36 | 2047 |
|  | summer (FM) | 1.85(0.49) | 1.64(0.77) | 0.21(0.78) | 0.78 | 0.30 | 215 |
|  | autumn (MJ) | 1.72(0.37) | 1.20(0.37) | 0.53(0.49) | 0.49 | 0.10 | 749 |
|  | spring (ON) | 1.69(0.49) | 1.52(0.55) | 0.17(0.52) | 0.52 | 0.50 | 1083 |





| | | | | | | | |
|---|---|---|---|---|---|---|---|
| Bell/Amund | Regional average | 1.96(0.65) | 1.87(0.80) | 0.09(0.71) | 0.71 | 0.54 | 694 |
| | summer (FM) | 2.26(0.54) | 2.31(0.82) | -0.05(0.79) | 0.79 | 0.40 | 63 |
| | autumn (MJ) | 1.92(0.60) | 1.62(0.72) | 0.30(0.62) | 0.62 | 0.58 | 282 |
| | spring (ON) | 1.95(0.69) | 2.00(0.79) | -0.05(0.72) | 0.72 | 0.54 | 349 |