# Peer review of "A comparison between Envisat and ICESat sea ice thickness in the Southern Ocean"

_The Cryosphere, 2021_

## Author Comment (AC1)

Dear Reviewer:

We would like to express our gratitude to you for the helpful comments to improve this manuscript. For GC1 and GC2, we modify the descriptions carefully to avoid misunderstandings following your specific comments. For GC3, we add this "total thickness" information in the data description and discussion of ICESat SIT uncertainty. For GC4, we conduct further analyses towards the FDD results combining sea ice drift information. For GC5, we switch the descriptions to active mode as you suggest.
The specific responses and revisions are shown below. They are in blue font for clarity.

Qinghua Yang, Qian Shi, Robert Ricker, Stefan Hendricks
On behalf of all the authors

Specific Comments: (I abbreviate Line with L)
Title: Since sea ice is an integral part of the Southern Ocean I suggest to use "Southern Ocean" instead of "Antarctic" ... perhaps even throughout the entire paper.
Thanks for your comments. We replaced some of the "Antarctic" with "Southern Ocean", especially in the cases like "in the Antarctic".

L50: At this point I suggest to provide a summary sentence which states that all these various data sets - despite covering limited regions and/or time periods - are extremely useful for the evaluation of models and satellite retrieval methods. I suggest to also differentiate between data sets that provide sea-ice thickness information at one fixed location (ULS) and hence allow to check the consistency over time, and data sets which have a short duration but with high resolution cover comparably large regions (e.g. Operation ice bridge or AEM) and hence allow to check the spatial variability of the sea-ice thickness retrieved from satellite data.

(1) We added a summary sentence: "Despite covering limited regions and/or time periods, all these various observational data sets are extremely useful for the evaluation of models and satellite retrieval methods."

(2) We changed the expression to differentiate the data sets: "**One type of observation data** is in situ measurements providing sea ice thickness information at one fixed location and some allow to check the consistency over time. For example, drilling data (e.g., Meiners et al., 2012) are accurate but extremely limited in temporal and spatial coverage, and hence they cannot be used to obtain an understanding of large-scale Antarctic sea ice thickness processes. Upward-looking sonars (ULS), located at 13 different sites in the Weddell Sea, provide valuable temporal evolution of sea ice draft (Harms et al., 2001; Behrendt et al., 2013a; Behrendt et al., 2013b), but a basin-wide spatial distribution cannot be derived. **The other type of data sets** has a short duration but with high resolution cover comparably large regions and hence allow to check the spatial variability of the sea-ice thickness retrieved from satellite data. ......"

L50-56: I suggest to reorganize this information a bit. First of all Kurtz and Markus 2012 and Li et al. 2018 utilize laster altimetry and hence fall into what you describe in the last sentence of the lines referred to here; this should somehow be merged. Secondly, Bernstein et al. is a paper about trying to estimate sea-ice thickness in the Ross/Amundsen Sea only from a very limited set of sea-ice charts. This data does not have the same value as the data sets of the other two papers cited in the same sentence.

Thanks for your comments. We removed the references for the first sentence: "More recently, satellite remote sensing has been widely applied to investigate the spatial coverage and long-term trend of sea ice thickness in the whole Southern Ocean." Meanwhile, we cited "Kurtz and Markus, 2012" after the sentence "Satellite altimetry, including radar and laser altimetry, have been used in the Southern Ocean to retrieve sea ice thickness".

L63/64: While I am totally fine with the sentence that snow affects radar altimetry SIT retrievals in two ways, you should first tell the reader the two ways before you come up with details of the shortcoming. First i) snow depth is required to a) correct the radar wave speed in snow and hence to appropriately convert the radar freeboard into the seaice freeboard and to b) convert sea-ice freeboard into sea-ice thickness. In both cases, but mostly in b) also the snow density plays a role. Secondly ii) the presence of snow simply modifies how the radar signal is reflected in / by the ice-snow system; the assumption of Beaven et al. is for DRY snow only. Hence, in addition to the more physical/mathematical influence of snow depth, there is this potential violation of the full-penetration assumption made by Beaven et al as is demonstrated by Willatt et al. These issues need to be specified first before you can come up with the details in Lines 65+

Thanks for your comments. We modified the description as you suggested: "Firstly, snow depth is required to correct the radar wave speed in snow and hence to appropriately convert the radar freeboard into the sea ice freeboard, as well as to convert sea ice freeboard into sea ice thickness. Secondly, the presence of snow modifies how the radar signal is reflected by the ice-snow system. Specifically, over Antarctic sea ice, the complex snow stratigraphy and frequent snow flooding associated with the formation of snow ice and superimposed ice affect radar altimetry measurements (Willatt et al., 2010), i.e. the assumption of Beaven et al. (1995) is for DRY snow only. Besides, the snow depth climatology used in the retrieval of Envisat and CryoSat-2 SIT can cause additional uncertainties due to neglecting inter-annual variability in snow depth (Bunzel et al., 2018)."

L81/82: Here you please need to check recent literature because Kwok and Kacimi or Kacimi and Kwok came up with more VERY useful work based on ICESat-2 data. You should include these references here as well - and ideally already point to the fact that the coverage with ICESat-2 is much better than with ICESat.

We modified this sentence: "However, ICESat-2, which has been in orbit since 2018, provides a new source of year-round observations of total freeboard and thus better coverage than ICESat (Kwok et al., 2019; Kacimi and Kwok, 2020)."

L106/107: If I am not mistaken, then the Paul et al reference point to some data analysis and algorithm development but is not specifically the reference to cite the sensor properties of Envisat RA-2. Please find a more appropriate reference which also details the footprint issue. I doubt that also Connor et al. 2009 is the adequate reference here. I am sure that are papers from the early 2000s when the altimeter was just up or about to be launched in which the system specifications are laid out well.

We removed the citation of "Paul et al., 2017" and added the citation of "Peacock and Laxon, 2004".

L115: It might make sense to add that Laxon et al. applied this method to ERS altimeter data, i.e. the predecessor of the Envisat RA-2 instrument.

We added the information: "Sea ice thickness is retrieved from ice freeboard based on the hydrostatic

equilibrium approach as first used by Laxon et al. (2003), which applying this method to ERS altimeter data, i.e., the predecessor of the Envisat RA-2 instrument."

L120/121: "revised version ... Cavalieri et al (2014)" I recommend to not refer to a data set description here but refer to the main core paper of the apporach used which is the one by Markus and Cavalieri, 1998, and then it is the Comiso et al (2003) reference which points to the AMSRE sea ice processing. I suggest to make clear what the "revision" is (different tie point retrieval plus addition of retrieval errors). It would also be good if you could tell the reader on data of which years the snow depth climatology is based - because it extends well into the AMSR2 period. Finally, you may please change the URL into https://www.cen.uni-hamburg.de/icdc .

We checked the snow depth climatology data and found it only covered AMSR-E period. Therefore, we removed the AMSR2 information and modified this part as you suggested: "A snow depth climatology **(based on 2002-2011)** is employed to retrieve sea ice thickness from sea ice freeboard here (**Markus and Cavalieri, 1998; Comiso et al., 2003**). This snow-depth climatology is derived from the passive microwave sensor Advanced Microwave Scanning Radiometer-EOS (AMSR-E) for the Antarctic and is based on a revised version of the approach **with different tie point retrieval plus addition of retrieval errors** and provided by the Integrated Climate Data Center (ICDC, https://www.cen.uni-hamburg.de/icdc)."

L122/123: "the actual SIT (... mean thickness ... of the grid cell area)" --> this does not go together well. The actual SIT would be the thickness of the ice floes as they float around in the grid cell. The mean SIT takes into account that the grid cell might not be fully covered by sea ice. Hence the actual SIT is always larger or equal than the mean SIT and it is important that you write this down in a clear way.

We corrected the explanation of actual SIT: "ice thickness of the ice-covered fraction of the grid cell area".

L138-140 / Eq. 3: I guess it is important to check this equation and the wording. If I am not mistaken, then the authors of these data claim on the respective web page that it is actually not the sea-ice thickness that is retrieved with this equation but it is the total (sea ice plus snow) thickness. Hence it is in a way the same type of thickness as is observed by that famous airborne EM sensor (see your introduction). In order to obtain the sea ice thickness from I retrieved using (3) one should possibly substract the snow depth and/or reformulate equation (3) such that this effect is somehow included.

We checked the equation and corrected the mistake: "And total thickness (sea ice thickness plus snow depth) can be determined from it."

L141: Please check whether this product contains the mean gridded sea-ice freeboard or whether this is perhaps in fact the total (sea ice + snow) freeboard.

We corrected the mistake by changing "sea ice freeboard" to "total freeboard".

L147: "at more than 900 m underwater" --> I don't think that this is a relevant information because the actual sensor is mounted further up anyways - otherwise the comparably small footprint would not be possible to achieve and the footprint would possibly also change between ULS sensor locations.

Thanks for your comments. We agreed and removed this information.

L166: When I look at Fig. 8 I have difficulties to fully understand what you did. First of all, the annotation in the Figure is opposite to what you write here. Secondly, what are the start and end days for the FDD computation using, e.g. the period from FM to MJ? The same question for MJ to ON. I find it strange and not easy to understand that you kept the FDD in degrees C and did not attempt to translate this into a net ice thickness growth. With that it remains a very qualitative comparison.

(1) We added the explanation for the annotation: "We compare FDD with the SIT variations from summer to autumn (FMMJ) and from autumn to spring (MJON) represented by Envisat and ICESat SIT. Specifically, FMMJ represents the differences between mean FDD in MJ and mean FDD in FM following ICESat operating periods and so does MJON." For example, during the period from FM to MJ in 2004, the FDD computation is conducted by subtracting mean summer SIT (from Feb 17 to Mar 20) from mean autumn SIT (from May 18 to Jun 20).

(2) We don't translate the FDD into thickness growth because we think FDD is a robust measure for potential thermodynamic ice growth (or melt). Converting FDD into thickness would require a model and additional assumptions with uncertainties that we cannot constrain sufficiently. Therefore, we think it is reasonable to only consider FDD.

L167: "neglects ice growth from snowfall, freezing rain or ridging" --> I suggest to be more specific with your formulation. "snowfall" per se does not lead to ice growth. It requires the process of flooding. "freezing rain" does not trigger ice growth - at least not to my knowledge. While melting of ice crystals requires energy, formation of ice from undercooled water releases energy; hence freezing rain, although contributing millimeters of ice - mostly on top of snow - warms the snow / ice. Finally ridging is no form of ice growth. It causes dynamic thickening of the ice using ice which is already there.

Thanks for your comments. We changed "ice growth" to "SIT variations" to avoid the misunderstanding.

L174-177: While this is possibly a good approach it leads to the observed partly considerably larger coverage with Envisat SIT data in Figs. 4 to 6, particularly Fig. 5, which at first glance is a bit puzzling. It is of course not relevant for the comparison as long as you only consider grid cells where both, Envisat and ICESat provide values. But as shown it implies that Envisat, e.g., has much more ice in summer 2005 (Ross Sea) or 2007 (several regions) but this is just because your Envisat SIT map shows data of the entire month, e.g. April, into which an ICESat period overlaps. You could include a comment about this into your text or, alternatively, only show Envisat SIT values where both satellites provide a SIT estimate.

We considered it necessary to show more Envisat SIT data and we chose to add a discussion sentence to point out the problem: "It is noted that this approach can lead to considerably larger coverage with Envisat SIT data than ICESat when Envisat SIT maps show data of the entire month into which an ICESat period overlaps, e.g., summer 2005 (Ross Sea) or 2007 (several regions) in Fig. 5."

L186-189: What is the motivation to use these sea-ice concentration data which I assume are based on the ASI algorithm? If you keep this product please make sure that you refer to the algorithm name and to also provide information about the native spatial resolution of this product (which is much finer than 100 km). It might also make sense to provide the URL to the data set web page at ICDC if there is any.

Since this product is contained in the ICESat SIT data, we use it to convert the mean gridded SIT to actual SIT. We modified the introduction as you suggested: "The sea ice concentration data are derived from Special Sensor Microwave/Imager (SSM/I) and Special Sensor Microwave/Imager Sounder (SSM/IS) **based on ASI algorithm** provided by ICDC (Kaleschke et al., 2001; https://www.cen.unihamburg.de/en/icdc/data/cryosphere/seaiceconcentration-asi-ssmi.html) **with 12.5 km spatial resolution**, interpolated to 100 km grid NSIDC polar-stereographic grid and averaged over respective ICESat measurement periods."

L207: The statement about the SIT uncertainties in the Worby 1-layer SIT data set is potentially not correct. I checked the data set and found uncertainties for both freeboard and thickness. Reading the paper Kern et al. 2016 it seems relatively clear that their computation of the SIT uncertainty included in the product is similar to their SICCI-2 SIT product from ICESat and hence based on uncertainties in densities and freeboard; only and here you are correct - snow depth uncertainty is not included. You might want to rephrase you text accordingly. Also, if I am not mistaken, then the uncertainty estimates provided with the Envisat SIT data set are possibly too large because the data set producers those days did not adequately take potential correlations between the error contribution into account. I am quite sure that, for instance, for the currently available (from AWI) CS-2 sea-ice thickness data the uncertainty is considerably smaller than for the SICCI-2 project data set and I am sure the same applies to the Envisat RA-2 data set. But you have the producers among your co-authors. So you simply need to ask!

(1) We amended the sentence about the SIT uncertainties: "ICESat SIT uncertainties are also calculated based on the uncertainties of densities and freeboard (Kern et al., 2016)."

(2) We rewrote the potential reasons: "The large differences in the error bars between Envisat and ICESat mainly result from the inclusion of snow depth uncertainty in Envisat SIT, and without adequate regard for potential correlations between the error contribution."

L215/216: I suggest to differentiate a bit better here between ICESat and Envisat - because Envisat provides a larger data set and hence your comparison is based on more data pairs. While not possible for ICESat it would be possible for Envisat SIT to come up with a statement about the agreemen of the seasonal cycle. Do ULS and satellite data sets provide the same seasonal cycle qualitatively?

We modified the summary sentence of the comparisons with ULS: "However, the numbers of valid data are too small to derive a reliable conclusion on the accuracy of ICESat. The comparison is based on more data pairs for Envisat, but the agreement of the seasonal cycle between ULS and satellite data sets is bad qualitatively (Fig. 3)."

L221/222: "one satellite SIT grid cell is scanned only one of twice through a month" --> Please make sure to be more specific here. Not all these grid cells are covered only one / twice a month. Also this is valid for ICESat but possibly not for Envisat.

We modified this sentence: "one ICESat SIT grid cell is scanned once or twice on average through a measurement period."

L225-227: "However ... fixed ULS positions." --> While I agree that thanks to the ice motion and the integration period used the ULS point measurement kind of gains a larger representativity, it might be worthwhile to check i) how large the ice drift actually was and what their average direction was. You could use the NSIDC V4.1 sea-ice motion data set to figure this out.

As we know, ULS indeed measures the continuous ice draft in a fixed location with diameter of several meters. Considering the ice motion, ULS actually acquired dozens to hundreds of kilometers records along the trajectory of sea ice motion on monthly basis, which have enough spatial representativeness compared with ICESat-1/Envisat. Here, we track the source of sea ice that flowing over the ULS on

specified month by backward tracking method based on NSIDC Pathfinder data sets. We find the ice draft records included in ULS monthly mean calculation come from a wide range area (Fig. 1). Therefore, we think this is enough to prove that the spatial representativeness of the monthly average ULS data can be compared with that of ICESat-1/Envisat.

Besides, ULS data is one of the main reference dater sources used for assessing ice thickness from remote sensing in the past decades. ULS are used for comparison with the ice thickness derived from AVHRR (Yu and Rothrock, 1996; Drucker et al., 2003). It was also used to compare with ICESat-1 ice thickness in the Fram Strait (Spreen et al., 2009). In addition, the ULS data sets have been also used for comparison with reanalyses data in the polar region (Mu et al., 2018; Shi et al., 2021). In addition, the comparison with ULS data sets is also a convention for assessing the quality of ice thickness derived from altimeters in the European Space Agency (Kern et al., 2018).

[Figure]

Fig. 1 The 30-day origins of the sea ice passing the three ULS sites in July 2011. The red dots stand for the ULS locations and the blue dots stand for the original locations of the sea ice using backward tracking method.

L237: Not clear what you mean by "The same feature is found ..." --> Are you referring to the existance of a polynya? Or are you referring to the fact that for both polynya regions, Ross Sea and Weddell Sea Envisat SIT is much higher than ICESat SIT? Please be more specific.

We clarified this sentence: "The same feature is found for the Ronne Ice Shelf polynya in the Weddell Sea in 2007, with the polynya appearing only in the ICESat map."

L239: "possibly fails ..." --> This is not a specific enough wording. There are two things involved with that. A) using a 100 km grid naturally results in a land mask at the same grid resolution. Hence it is very likely that the land mask used in the ICESat product extends further into the open ocean than the landmask which is used in the Envisat product. B) As stated in Kern and Spreen, it is not overly bad to not take ICESat freeboard estimates close to the coast not into acount because there the freeboard often is less accurate here compared to the open ocean due to various issues, mostly because of a lack of enough

open leads detected by ICESat and hence a less accurate approximation of the local sea surface height and with that less accurate total freeboard.

We added the discussion here as you suggested: "Using a 100 km grid naturally results in a land mask at the same grid resolution and it is very likely that the land mask used in the ICESat product extends further into the open ocean than the landmask which is used in the Envisat product. Besides, as stated in Kern and Spreen (2015), it is not overly bad to not take ICESat freeboard estimates close to the coast not into acount because there the freeboard often is less accurate here compared to the open ocean due to various issues, mostly because of a lack of enough open leads detected by ICESat and hence a less accurate approximation of the local sea surface height and with that less accurate total freeboard."

L247/248: This apparent discrepancy could be mitigated by showing Envisat SIT only for those grid cells where ICESat has SIT values - as I mentioned earlier already. Otherwise it might be difficult to understand why the small difference between the sea-ice concentration thresholds used (60% vs. 70% ?) has such a large impact on the spatial coverage with SIT data.

Thanks for your advice. However, we think that showing larger coverage of Envisat can help achieving improved understanding of Envisat product itself. Besides, the sea ice concentration threshold for Envisat is 70 % while 60 % for ICESat. Therefore, we don't think the different thresholds are the cause of the different spatial coverage in Ross Ice Shelf Polynya.

L253-254: "probably ... resolve thick ice" --> while the statement made is correct for along-track data you need - in my eyes - to consider two issues here. The first one is that the ICESat product is gridded on a 100 km grid. Given the sparseness of ICESat overpasses with valid data such a 100 km grid SIT estimate in that region might be biased by the presence of thick landfast ice. The second one is that thanks to its finer along-track resolution ICESat can expected to be more sensitive to ocean swell. Ocean swell can result in anomalously high freeeboard values which then convert into too high sea-ice thickness values. While this is a local phenomenon again the sparseness of ICESat overpasses with valid data can results in a similar effect as described above for landfast ice.

Thank you for your comments. We considered the two issues you suggested here carefully and decided to remove this statement.

Fig. 8: I am wondering whether you could perhaps change the color table used for the FDD. It is not intuitive. A high number of FDD denotes cold conditions while a low number comparably warm conditions. I suggest you use a color table which goes from white (0 FDD) to blue (3000degC FDD). Please check whether it is common to express FDD this way. I find it strange to read about temperatures of 1500 and 3000 deg C. Also switching to the unit Kelvin does not solve the problem; ideally, as mentioned earlier, you would translate this to a net growth of sea ice (in meters). Did you check that the FDD shown for MJ-ON is in fact for that period and not for the full FM to ON period? Please note that the notation MJ-FM and ON-MJ is opposite to what you write in the text. Since you aim is to express that the maps in the right two columns show a SIT difference of, e.g. ON minus MJ you might need to invest more annotation elements to not confuse the reader.

We replotted the FDD figures by changing the color table, modifying the expression of the notations and removing the unit of FDD based on the explanation of FDD. We kept analyzing FDD instead of the converted SIT because we think FDD is a robust measure for potential thermodynamic ice growth (or melt).

[Figure]

Fig. 2 The FDD differences and sea ice thickness differences from summer to autumn (FMMJ) and from autumn to spring (MJON) derived from Envisat and ICESat in 2004, 2005 and 2006. The FDD patterns are derived by forward tracking daily FDD with sea ice motion data.

L273/274: "This pattern ..." --> I suggest to add the fact that the thick ice found in the southwestern

Weddell Sea at the end of summer is advected northward. If you look at the SIT distributions it is both the tail at large SIT which is decreasing and the tail at small SIT which is increasing. In the particular case you mention here, the thick old ice is replaced by the thin younger ice formed in the polynya (plus other comparably thin ice that is recirculated from the Eastern Weddell Sea in winter.

We added this information: "The thick ice found in the southwestern Weddell Sea at the end of summer is advected northward and the thick old ice is replaced by the thin younger ice formed in the polynya and other comparably thin ice that is recirculated from the Eastern Weddell Sea in winter."

L274-276: "The adverse ... reveal them" --> I would have wished for a more detailed discussion here because one can interpret a lot from these maps - provided one takes into account knowledge about typical snow fall patterns and ice motion. Here you could substantially add some more interesting information and interpretation to your paper.

We conduct the forward tracking on daily FDD with the NSIDC sea ice motion data to add the dynamic effects on the purely thermodynamic growth pattern (Fig. 2). We find that with the aid of sea ice motion, thick ice in the Weddell Sea and Ross Sea can be moved northward. However, the Envisat SIT decrease during MJON still cannot be explained considering the dynamic processes. Therefore, we assume the main reason of the SIT decrease is the overestimation of Envisat SIT in autumn. As for the snow fall patterns, we think it difficult to quantify the impact of snow, i.e., in which way the snow fall would lead to the sea ice thickness growth.

Fig. 9: Please add to the caption what the black line and the dashed colored lines stand for. You might also give the information whether you took data from all seasons available or whether we only look at data of years 2004, 2005 and 2006 as only from these years data from all three seasons are available from ICESat.

We added the information you mentioned: "The data are taken from all seasons available." "The black line is the 1-to-1 fit line and the dashed colored lines stand for linear regression lines."

L322-326: Please note that the "nominal adjustement" suggested by Nandan et al. is derived for cases in the Arctic which might be special and not necessarily transferrable to the Southern Ocean. You could mitigate focussing too much on this exact value of 7 cm by providing a table into which you put sea-ice thickness changes in response to freeboard biases between 2 amd 10 cm in steps of 2 cm. You choose typical first-year sea-ice density. Did you expereiment with other density values to see how dominant the freeboard change is compared to a density change? You could use densities between 880 and 940 kg/m3 in steps of 20 kg/m3 to illustrate this. Why can the differences found here not also account for the differences between Envisat and ICESat in spring? And why do you consider the end of summer a season when this difference might apply?

We experimented with different density values following your suggestions and changed the way to present the sensitivity of the SIT changes to freeboard biases, snow depth biases and sea ice density. From Fig. 3, we can see that the SIT changes are more sensitive to sea ice freeboard biases than to snow depth biases. Besides, with the increase of sea ice density, the SIT changes rise.

Considering the average freeboard biases, the corresponding SIT changes can be up to 0.5 m, which matches with the consistent positive differences in summer and autumn but not in spring.

[Figure]

Fig. 3 Sensitivity of sea ice thickness changes to sea ice freeboard biases and snow depth biases as function of sea ice density. (a) SIT changes computed with Eq. (1) for different sea ice freeboard biases (2 cm to 10 cm). (b) Similar to (a) but computed for different snow depth biases (5 cm to 30 cm).

L356-359: Please state that you took the same values for water and sea-ice density as in Eq. 5. While your computation is of course correct, I am wondering whether the 2 cm bias assumed isn't a strong under-estimation. Yes, the analysis is based on monthly data, I agree. But the recommendation of Nandan et al you used in Eq. 5 is not tied to monthly data, is it? The monthly mean retrieval uncertainty you used should be considered the precision and not the potential bias which can be much larger - as you learned from Worby et al., Ozsoy-Cicek et al and as you could also see in the Kern and Ozsoy-Cicek paper in Remote Sensing from 2016; there we easily talk about 20 cm bias. Also te work of Kwok and Maksym from 2014 supports the notion that biases can be much higher over large regions. Hence, considering that also on a monthly scale the bias is an order of magnitude larger does not hurt and I invite you to, as suggested for Eq 5 provide a table into which you put sea-ice thickness changes in response to snow biases between 5 and 30 cm in steps of 5 cm; that would provide a much more realistic view of the potential bias due to using a snow depth data set that does not reflect the actual conditions.
We modified the analyses following your comments and the revision is shown above.

L365-367: "While a snow ..." --> I agree to this and suggest to also stress one more time that sea-ice thickness differences you observe in your paper between different summer seasons (e.g. between Feb/Mar 2004, 2005, ... 2008) might, to a large extent, also simply be the result that the climatology does not match the actual conditions.
We stressed this point as you suggested: "The SIT differences between different summer seasons might, to a large extent, also simply be the result that the climatology does not match the actual conditions. In addition, snow-depth dependent radar signal delay is applied to convert the radar freeboard into the sea-ice freeboard but the delay correction is based on a conventional assumption that has been revised (Mallett et al. 2019) since the generation of the SICCI data."

L374-386: You might want to mention here that possibly the approach by Kern et al. (2016) is providing the total (sea ice plus snow depth) thickness. Taking this into account, the actual 1-layer sea-ice thickness

values shown in this paper would possibly even be a bit smaller - with the respective consequences for your results. See also my comments in the context of Eq. 3.

We added this discussion: "In addition, the approach by Kern et al. (2016) is actually providing the total (sea ice plus snow depth) thickness. Taking this into account, the actual 1-layer sea-ice thickness values shown in this paper would possibly even be a bit smaller."

L402-404: "Compared to the FDD ..." --> In order to make this quite general statement you should investigate these maps in more detail and write more text in the respective section. See also my comments about your usage of FDD.

We modified the statement based on the new results we achieve: "Compared to the FDD results in 2004, 2005 and 2006, we find that the Envisat SIT decrease during MJON cannot be explained considering the dynamic processes and we assume the main reason of the SIT decrease is the overestimation of Envisat SIT in autumn."

Editoral remarks / Typos:

L31: Actually, to obtain the sea-ice volume you need to combine the sea-ice thickness with the sea-ice area. I strongly recommend to change the working accordingly.

We changed "sea ice extent" to "sea ice area" since the slight increase and the abrupt decrease occur on sea ice area (Maksym, 2019).

L57++: Please check the paper for the typo: CyroSat-2. It needs to read "CryoSat-2"

We corrected these mistakes.

L104: "aboard on" --> either "aboard" or "on".

We deleted "on" here.

L112/113: "The delay correction ... " I suggest to delete this sentence here and instead add it in the discussion section when you discuss error sources / the uncertainties of the Envisat data.

We moved this sentence to the discussion section as you suggested.

L129: As ICESat is not operating anymore it is grammatically possibly more correct to write "lasted" instead of "lasts".

We corrected these mistakes.

L153/154: "The uncertainty ... height calibration" --> I suggest to rewrite this: "The uncertainty in summer is smaller than in other seasons because open water occurs more frequently in the ULS footprint and with that the estimate of the sea surface height is more accurate.

We modified this sentence as you suggested.

L241/242: "However, ... near zero." -->perhaps better: "However, these differences have to be seen in the light of the standard deviations of ~0.6 m."

We modified this sentence as you suggested.

L258/259: "According to Table 5 ..." --> you could point out better that DESPITE the large difference and RMSD the correlation is actually the highest of the three seasons investigated.

We modified this sentence: "According to Table 5, despite the largest mean difference in autumn of 0.57 m and large RMSD of 0.47 m, the correlation is actually the highest of the three seasons investigated of 0.71."

L281: What are "splashes"?

We replaced "splashes" with "cloud" here.

L294/295: "though it is known ..." --> please support this knowledge with respective references.

We added the reference here: Willatt et al., 2010.

L296: "footprint of" --> "footprint of the radar altimeter of"

We modified this expression.

L372: The perfect place for the Kwok and Maksym paper from 2014 (JGR-Oceans I think) and possibly for one of his more recent papers where he looked into ICESat-2.

We added the references here: Kwok and Maksym, 2014; Kacimi and Kwok, 2020.

L389/390: --> This sentence reads a bit strange in the context of what follows. My suggestion: "In this study, we compare estimates of the sea-ice thickness obtained from satellite altimeter observations by Envisat RA-2 (radar) and ICESat GLAS (laser) in the Southern Ocean."

We modified this sentence as you suggested.

L391: "Envisat-ULS" --> please make sure the reader understands the "-" as a minus so that it is clear that ULS sea-ice thickness values are smaller than Envisat (and ICESat) values. Currently, this is not clear from the text.

We spelled out the "-" here to make it clearer.

L392: "The results ..." --> I don't understand this sentence in the context of the previous one. Consider to remove.

We removed this sentence.

L394/395: "According ..." --> three time usage of difference / different. Consider to rephrase.

We rephrased this sentence as: "According to the results, the differences between Envisat and ICESat sea ice thickness vary in each season, year and region".

L395/396: "difference of ... between Envisat SIT minus ICESat SIT" reads strange. Please consider rephrasing. I note: In contrast to L391 here you spell out the "-".

We modified this sentence as: "More specifically, the smallest monthly average difference (SD in brackets) for Envisat SIT minus ICESat SIT exists in spring of 0.00 m (0.39 m), while larger differences (SD) exist in summer and autumn by 0.52 m (0.68 m) and 0.57 m (0.45 m), respectively."

L406-408: You might want to re-phrase this sentence after you have considered by comments in the context of Eq. 5 and 6.

We modified the sentence as: "Through the sensitivity experiments, we find that Envisat SIT changes are more sensitive to sea ice freeboard biases than to snow depth biases. Besides, with the increase of sea ice density, the SIT changes rise."

Figure 3: I suggest that you avoid to write "ENV-ULS" and the like because it is easily misinterpreted as a difference Envisat SIT minus ULS SIT which I doubt is the quantity you are showing here.

Thanks for the comments but here "ENV-ULS" does represent the difference between Envisat SIT and ULS SIT.

---

## Author Comment (AC2)

Dear Reviewer:

We would like to express our gratitude to you for the comments to improve this manuscript. However, we need to clarify that the purpose of this paper is to give a comprehensive and statistical comparison between Envisat and ICESat sea ice thickness data. Only when the significant differences are admitted, the importance of dealing with the uncertainties of these products is revealed. Besides, we have already discussed the probable causes of the differences in section 4 and we supply more experiments following your suggestions.

The specific responses and revisions are shown below. They are in blue font for clarity.

Qinghua Yang, Qian Shi, Robert Ricker, Stefan Hendricks
On behalf of all the authors

**Major concerns:**

L90-92. I think the comparison of the two SIT products with ULS is not appropriate since the single measurement point (6-8 m) cannot represent a grid with 50 km or even 100 km. Moreover, only the uncertainty of sea ice draft derived with ULS 5-12 cm is presented (L152-153), the uncertainty of SIT derived with Eq. 4 is missing and Fig.3 also lacks error bars for ULS, thus making the comparison unreliable. "Both Envisat and ICESat SIT have been interpolated onto each ULS location in the nearest neighbour way" (L183-184) further introduces huge uncertainties. Based on these considerations, it is not recommended to use ULS as a comparison data source. ULS can be used if the Envisat or ICESat footprints spatio-temporally coincide with it, and the uncertainty of SIT derived with ULS is clear.

Thanks for your comments. However, we think that you are biased in denying the feasibility of using ULS data as comparison data with ICESat-1/Envisat due to their relatively narrow footprint. As we know, ULS indeed measures the continuous ice draft in a fixed location with a diameter of several meters. Considering the ice motion, ULS acquired dozens to hundreds of kilometers records along the trajectory of sea ice motion on a monthly basis, which have enough spatial representativeness compared with ICESat-1/Envisat. Here, we track the source of sea ice that flows over the ULS in a specified month by backward tracking method based on NSIDC Pathfinder data sets. We find the ice draft records included in ULS monthly mean calculation come from a wide range area (Fig. 1). Therefore, we think this is enough to prove that the spatial representativeness of the monthly average ULS data can be compared with that of ICESat-1/Envisat.

Besides, ULS data was generally used for ice thickness comparison in the previous studies. ULS is used for comparison with the ice thickness derived from AVHRR (Yu and Rothrock, 1996; Drucker et al., 2003). It was also used to compare with ICESat-1 ice thickness in the Fram Strait (Spreen et al., 2009). In addition, the ULS data sets have also been used for comparison with reanalyses data in the polar region (Mu et al., 2018; Shi et al., 2021). In addition, the comparison with ULS data sets is also a convention for assessing the quality of ice thickness derived from altimeters in the European Space Agency (Kern et al., 2018).

In summary, we think that the reason for rejecting us due to the spatial representativeness of ULS ice thickness is untenable. Previous studies (referred to above) have shown that using ULS for validation of satellite-derived sea-ice thickness data sets can be considered as state of the art.

[Figure]

Fig. 1 The 30-day origins of the sea ice passing the three ULS sites in July 2011. The red dots stand for the ULS locations and the blue dots stand for the original locations of the sea ice using backward tracking method.

The difference between the Envisat-based actual SIT, i.e., the mean thickness of the icecovered fraction of the grid cell area (without open water areas) (L122-123), and the ICESat effective sea ice thickness, i.e., mean thickness per grid cell including open water areas (L141-142), is not tackled nor discussed for the two datasets.

We point out the different thickness representations for Envisat and ICESat. And we choose to compare the effective sea ice thickness during the intercomparison process. We have clarified this in the paper (L229).

Considering the huge differences between Envisat and ICESat SIT products (as can be seen in Fig. 9 and Table 7), the main object of this work should not stay at just comparing those products, but concentrating on the qualitative and quantitative analysis of the causes leading to the differences. Currently, these issures are only simply discussed in Section 4. Following works may be considered by the authors:

L253-254 About the sentence "Probably inferring that …" Is it really the key reason for SIT overestimation of Envisat than ICESat in autumn? The similar doubt also appears in summer (L262-263). L21 and L256-257. Why on earth the mean Envisat SIT decreases while the mean ICESat SIT increases from autumn to spring? This should be supported with supplement experiments.
L360-361. "The largest effect might not come from the impact of ice deformation on the snow-depth retrieval but might be due to the difference between actual snow depth from that represented by the climatology." Can the influence of climatology quantified?
I didn't see solid evidences supporting the statement "The potential overestimation of sea ice freeboard caused by range biases accounts for much of the differences between Envisat and ICESat SIT in summer and autumn, while the biases of snow depth are not the dominant cause of the differences."

(1) We realized that the statement is correct only for along-track data. Firstly, given the sparseness of ICESat overpasses with valid data such a 100 km grid SIT estimate in that region might be biased by the presence of thick landfast ice. Besides, ocean swell can result in anomalously high freeboard values which then convert into too high sea-ice thickness values. While this is a local phenomenon, the sparseness of ICESat overpasses with valid data can results in a similar effect as for landfast ice. Therefore, we considered the two issues here carefully and decided to remove this statement.

(2) We conduct the forward tracking on daily FDD with the NSIDC sea ice motion data to add the dynamic effects on the purely thermodynamic growth pattern (Fig. 2). We find that with the aid of sea ice motion, thick ice in the Weddell Sea and Ross Sea can be moved northward. However, the Envisat SIT decrease during MJON still cannot be explained considering the dynamic processes. Therefore, we assume the main reason of the SIT decrease is the overestimation of Envisat SIT in autumn. As for the snow fall patterns, we think it difficult to quantify the impact of snow, i.e., in which way the snow fall would lead to the sea ice thickness growth.

[Figure]

Fig. 2 The FDD differences and sea ice thickness differences from summer to autumn (FMMJ) and from autumn to spring (MJON) derived from Envisat and ICESat in 2004, 2005 and 2006. The FDD patterns are derived by forward tracking daily FDD with sea ice motion data.

(3) We quantify the contribution of the usage of snow depth climatology instead of actual snow depth

during the Envisat SIT retrieval, in response to one of the RC2's major concerns. We redo the retrieval of Envisat SIT by replacing the snow depth climatology with SICCI AMSR-E snow depth on level-3 sea ice freeboard data. The new Envisat SIT is compared with ICESat SIT and the variations of their differences are shown in Fig. 3. This figure reveals that the impact of snow depth climatology is relatively small, with a maximum variation of 0.34 m. The variations are larger in the Amundsen and Bellingshausen Sea and the Western Weddell Sea, compared to other sectors. Among the three seasons, the variations are larger in summer.

[Figure]

Fig. 3 Changes in the differences between Envisat and ICESat SIT for each comparison period and each region under the experiment of the snow depth climatology impacts.

(4) We conclude the sensitivity of the SIT changes to freeboard biases, snow depth biases and sea ice density in Fig. 4 by analyzing Eq. (1):

$$I = \frac{F\rho_{water} + S\rho_{snow}}{\rho_{water} - \rho_{ice}} \tag{1}$$

The sensitivities to freeboard biases and to snow depth biases are calculated by:

$$\frac{dI}{dF} = \frac{\rho_{water}}{\rho_{water} - \rho_{ice}} \tag{2}$$

$$\frac{dI}{dS} = \frac{\rho_{snow}}{\rho_{water} - \rho_{ice}} \tag{3}$$

From Fig. 4, we can see that though the magnitudes of the resulting thickness changes are quite similar, the SIT changes are more sensitive to sea ice freeboard biases than to snow depth biases. Besides, with the increase of sea ice density, the SIT changes rise. For typical sea ice freeboard biases (7 cm for the Arctic nominal adjustment suggested by Nandan et al. (2017, 2020)), the sea ice density variations induce

the thickness changes ranging from ~0.5 m to ~0.8 m. For typical snow depth biases (20 cm for the monthly mean retrieval uncertainty in Kern and Ozsoy-Cicek (2016)), the thickness changes from ~0.4 m to ~0.7 m. Although this sensitivity analysis is not solid enough for the explanation for the SIT differences in three seasons, it can provide a reasonable conjecture that freeboard biases are the main cause of the positive differences in summer and autumn.

[Figure]

Fig. 4 Sensitivity of sea ice thickness changes to sea ice freeboard biases and snow depth biases as function of sea ice density. (a) SIT changes computed with Eq. (1) for different sea ice freeboard biases (2 cm to 10 cm). (b) Similar to (a) but computed for different snow depth biases (5 cm to 30 cm).

L124 The sea ice thickness derived with the modified ice density approach, i.e., Eq.3 can be considered to be updated to the new OLMi method (Xu, et al. (2021). "Deriving Antarctic Sea-Ice Thickness from Satellite Altimetry and Estimating Consistency for NASA's ICESat/ICESat-2 Missions." Geophysical Research Letters. http://dx.doi.org/10.1029/2021GL093425), which showed the modified ice density approach in Kern et al. (2016) would overestimate SIT.

Thanks for your information. We conduct the comparison between the Envisat SIT and the new ICESat SIT derived by Xu et al. (2021). Figure 5 shows consistent positive variations, with larger ones in summer, especially in the Amundsen and the Bellingshausen Sea and the Western Weddell Sea. However, we decide not to focus on the quality of different ICESat SIT, but investigate the causes of the differences between Envisat and ICESat SIT. Therefore, we keep using the ICESat product from Kern et al. (2016) and discuss its uncertainties in section 4.3.

[Figure]

Fig. 5 Changes in the differences between Envisat and ICESat SIT for each comparison period and each region under the experiment of the new OLMi ICESat SIT.

**Minor concerns:**

L22-24 Please quantify the percentage of the uncertainties caused by the radar backscatter and snow depth products respectively.

The new sensitivity analyses are shown above. Since the exact freeboard biases and snow depth biases cannot be quantified, we can only achieve a general SIT change coming from either of them.

L64 "the radar altimetry SIT retrievals" to "SIT retrieval by the radar altimetry"

We modified it as you suggested.

Are the densities used in Eq. 1 and Eq.2/3 the same? If not, how does it influence the SIT retrieved by the two sensors?

Yes, the densities are similar. The $\rho\_water$ in Eq.1 is 1024 kg m$^{-3}$ while in Eq.3 is 1023.9 kg m$^{-3}$. The $\rho\_ice$ in Eq.1 is 916.7 kg m$^{-3}$ while in Eq. 2 is 915.1 kg m$^{-3}$. The $\rho\_snow$ in Eq.1 and Eq.2 are the same of 300 kg m$^{-3}$.

L166 and L271 MF-MJ or MJ-MF? MJ-ON or ON-MJ? Please unify them throughout the paper, such as those 'MJ-ON' (in the text) or 'ON-MJ' (Fig. 8).

We unified the descriptions to "FMMJ" and "MJON" throughout the paper.

Why is it called snow depth climatology (L66, L118), snow-depth climatology (L119), or snow climatology (L363), and what is the real difference between them and the actual snow depth? Besides, what is the meaning of "have the character of a climatology" (L386)?

We unified the descriptions to "snow depth climatology". This snow depth climatology is an average snow depth based on 2002-2011 and it neglects the interannual variations of the snow depth. "Have the

character if a climatology" means that the sea ice thickness derived from Li et al. (2018) is still affected by the usage of climatology data.

L270 "from the model" is unclear.

We corrected this sentence to: "We calculate the period-average FDD corresponding to ICESat operating periods for the same spatial coverage."

L274-276 I don't think it is an adverse pattern comparing MJ-ON with FM-MJ. Please also make "different abilities" clear.

We replaced "adverse" with "opposite". "Different abilities" represent the ability to detect small-scale deformation processes.

L284-285 the weighted average is in the first row instead of in the last column?

We meant to explain the numbers in the last column and we moved this sentence to the caption in Table 7: "N is the numbers of comparison pairs, taking into account the actual number of values per season."

L379-381. The sentence "Therefore, …the ice-snow column" is hard to understand. For example, "underestimations of sea ice and snow observations" is not clear, is it sea ice thickness and snow depth underestimation? What is the "apparent ice density"?

Since visual ship-based observations of sea ice thickness and snow depth are used in the ICESat SIT retrieval, the underestimations of these data can have effects on the modified ice-snow density. We amended the sentence to: "Therefore, the largest uncertainty of ICESat comes from the potential underestimations of the ship-based sea ice thickness and snow depth observations for the computation of the bulk density of the ice-snow column (Kern et al., 2016)."

Fig.8 Suggest to use the same Antarctica background (in grey) as that in the other figures such as Fig. 4/5/6 since we can notice the big blank area along the Ross Sea coast in this figure.

We modified this figure as shown in Fig. 2.

Table 4 what's N? It should be introduced in the title. Same happens in Table 5/6/7.

N is the number of comparison pairs. We added this introduction in the captions of the tables.

Table 5. I suggest to also compute the difference between ENV and ICE at grid scale instead of just subtract with the computed statistical values (the "Difference" column). I mean, the mean of the third column of Figure 4/5/6 should be computed. Based on the figures, I think the two values would be different.

The averages of the difference patterns are calculated only for the grid cells with both available Envisat and ICESat SIT. Therefore, the two ways would lead to the same results.

Table 6. "sea ice thickness differences" should be followed by "with standard deviation in brackets".

We added this information in the caption.

References:

Yu, Y., and Rothrock, D. A. (1996), Thin ice thickness from satellite thermal imagery, J. Geophys. Res., 101( C11), 25753– 25766, doi:10.1029/96JC02242.

Drucker, R., Martin, S., and Moritz, R. (2003), Observations of ice thickness and frazil ice in the St. Lawrence Island polynya from satellite imagery, upward looking sonar, and salinity/temperature moorings, J. Geophys. Res., 108, 3149, doi:10.1029/2001JC001213, C5.

Spreen, G., Kern, S., Stammer, D., and Hansen, E. (2009), Fram Strait sea ice volume export estimated between 2003 and 2008 from satellite data, Geophys. Res. Lett., 36, L19502, doi:10.1029/2009GL039591.

Mu, L., Losch, M., Yang, Q., Ricker, R., Losa, S. N., and Nerger, L. (2018). Arctic-wide sea ice thickness

estimates from combining satellite remote sensing data and a dynamic ice-ocean model with data assimilation during the CryoSat-2 period. J. Geophys. Res., 123, 7763– 7780, doi: 10.1029/2018JC014316

Shi, Q., Yang, Q., Mu, L., Wang, J., Massonnet, F., and Mazloff, M. R.: Evaluation of sea-ice thickness from four reanalyses in the Antarctic Weddell Sea, The Cryosphere, 15, 31–47, https://doi.org/10.5194/tc-15-31-2021, 2021.

Kern, S., Khvorostovsky K., and Skourup, H.: D4.1 Product Validation & Intercomparison Report (PVIR-SIT), available at: http://icdc.cen.uni-hamburg.de/fileadmin/user_upload/ESA_Sea-Ice-ECV_Phase2/SICCI_P2_PVIR-SIT_D4.1_Issue_1.1.pdf, 2018.

---

## Author Comment (AC3)

Dear Reviewer:

We would like to express our gratitude to you for the helpful comments to improve this manuscript. We have carefully modified the discussion and the expression following your comments. The specific responses and revisions are shown below. They are in blue font for clarity.

Qinghua Yang, Qian Shi, Robert Ricker, Stefan Hendricks
On behalf of all the authors

GC1: Why not use same snow product and methods for ICESat and Envisat? Now the discussion of the differences due to sensors and snow depth only include a sensitivity of the hydrostatic equilibrium to snow depth/freeboard, but not the actual effect. It would be an option to calculate sea ice thickness from ICESat with the AMSR-E snow depths as well so you can compare what part of the difference is a direct effect from the difference in sensors and what is caused by the difference in snow depth. I understand that this involves quite some more work, but I think a the statement that is now made in the summary (L406-408) is a bit strong for the amount of proof you have for this, as you've not made the actual comparison.

Finally, we retrieve a new ICESat SIT using the same snow depth climatology product and the same hydrostatic equilibrium approach as Envisat SIT in response to GC1 suggested by RC3. The new ICESat SIT is compared with Envisat SIT and the variations of their differences are shown in Fig. 4. Compared to Fig. 4-6 in the manuscript, we can find that the positive differences in the Weddell Sea in summer and autumn increase, while the rest of the differences decrease.

However, we don't think this experiment is available to distinguish the impacts from the sensors and from snow depth as stated by RC3, since the new ICESat SIT still employs a possibly biased snow depth product. It can only be used to clarify the difference between the hydrostatic equilibrium retrieval method and the modified density retrieval method.

[Figure]

Fig. 1 Changes in the differences between Envisat and ICESat SIT for each comparison period and each region under the experiment of the new ICESat SIT.

GC2: Why is a different sea ice concentration threshold used for ICESat (60%, L143) than for Envisat (70%, L123)?

The usage of different SIC thresholds is because of the different thresholds used in the retrieval of the two data sets. Envisat SIT employs a SIC threshold of 70% during the retrieval while the ICESat SIT uses 60%. Only areas with sea ice concentrations greater than the threshold are considered a valid area for detection of leads and sea ice. We also tested the difference between using 60% and 70% SIC threshold for ICESat during the comparison with Envisat SIT. According to Table 1, this different threshold does not play an important role in the results of this paper. D(60) refers to Envisat minus ICESat (ENV-ICE) applying 60% SIC threshold for ICESat, while D(70) refers to ENV-ICE when SIC threshold for ICESat is 70%. Since the ice concentration gradients are usually quite steep, there will not be a lot of area with values 60% < SIC < 70%.

Table 1. Statistical results of the comparison between Envisat SIT and ICESat SIT using 60% and 70% SIC threshold at each operating period.

| | ON04 | ON05 | ON06 | ON07 | FM04 | FM05 | FM06 | MA07 | FM08 | MJ04 | MJ05 | MJ06 |
|---|---|---|---|---|---|---|---|---|---|---|---|---|
| D(60) (m) | 0.00 | 0.05 | -0.19 | 0.14 | 0.89 | 0.74 | 0.47 | 0.61 | 0.92 | 0.61 | 0.55 | 0.60 |
| D(70) (m) | 0.00 | 0.06 | -0.21 | 0.15 | 0.79 | 0.66 | 0.43 | 0.61 | 0.89 | 0.60 | 0.55 | 0.61 |

GC3: In lines 381-386 you introduce an improvement of the method you have used to obtain ICESat sea

ice thickness. What is the reason for not using this improved method?

We investigated the ICESat product that Li et al. (2018) produces by comparing with ICESat from Kern et al. (2016) and the ULS SIT used in this study. From Fig. 2 we can see that the differences between two ICESat products are small in general, with some larger differences in the West Weddell Sea and Amundsen Sea. Table 2 shows that compared with ULS SIT, ICESat SIT from Kern et al. (2016) performs even better. Based on these analyses, we think that the inter-comparison results with Envisat SIT in this study are not affected by the choice of ICESat product. Therefore, our work is still based on the data produced by Kern et al. (2016).

[Figure]

Fig. 2 Maps of differences that ICESat SIT from Kern et al. (2016) minus ICESat SIT from Li et al. (2018) at each operating period.

Table 2. The differences and RMSD between ULS SIT and the two ICESat SIT at each site. ICE(K) refers to ICESat SIT from Kern et al. (2016) and ICE(L) refers to ICESat SIT from Li et al. (2018).

|  | ICE(K)-ULS | | ICE(L)-ULS | |
| --- | --- | --- | --- | --- |
|  | D (m) | RMSD (m) | D (m) | RMSD (m) |
| Site 207 | 0.60 | 0.66 | 0.68 | 0.34 |
| Site 229 | -0.04 | 0.07 | 0.17 | 0.15 |
| Site 231 | 0.27 | 0.34 | 0.33 | 0.44 |

GC4: L229: 'an overall comparison between Envisat and ICESat effective SIT'. In the methods it said the Envisat SIT product 'represents the actual SIT (i.e., mean thickness of the ice-covered fraction of the grid cell area)' (L122-123) and that the ICESat SIT product is the 'effective sea ice thickness (i.e., mean thickness per grid cell including open water ares)' (L141-142). Are these two products compared here? This would not be a fair comparison, as the effective sea ice thickness is by definition going to be thinner than the actual sea ice thickness. If you are comparing actual sea ice thickness products please clarify here and in the methods section.

During the comparison with ULS observations, we compare Envisat and ICESat actual SIT to exclude zero thickness measured by ULS. Therefore, we divide ICESat SIT by the sea ice concentration contained in the ICESat data for each grid.

Then, we compare the effective SIT of Envisat and ICESat during the intercomparison work by multiplying Envisat SIT by the sea ice concentration contained in the Envisat data. We added the

information in section 3.2: "The effective Envisat SIT is calculated by multiplying the sea ice concentration contained in the data for each grid which come from OSI-SAF Global Sea Ice Concentration (OSI-409) until April 16, 2015 and the OSI-SAF Global Sea Ice Concentration continuous reprocessing offline product (OSI-430) afterwards (http://osisaf.met.no)."

GC5: What values or products have been used for water, snow, and ice density in the calculations of sea ice thickness with Eq 1, Eq 2, and Eq 3? Are they the same for Envisat and ICESat? If not, discuss the effect on the results.

The $\rho\_water$ in Eq.1 is 1024 kg m$^{-3}$ while in Eq.3 is 1023.9 kg m$^{-3}$. The $\rho\_ice$ in Eq.1 is 916.7 kg m$^{-3}$ while in Eq. 2 is 915.1 kg m$^{-3}$. The $\rho\_snow$ in Eq.1 and Eq.2 are the same of 300 kg m$^{-3}$.

GC6: ULS and satellite altimetry SIT distributions would be interesting to see as well, if possible.

Thanks for your advice. However, since there are little corresponding ULS data, the large-scale SIT distributions cannot be presented.

GC7: There are some significant issues with interpreting sentences throughout the manuscript. I have added some key examples below, but the clarity of the manuscript could improve from a thorough read-through.

We checked through the manuscript and made our best to interpret every sentence clearly.

L77-78: 'Several freeboard- ... compared (Kern et al., 2016).' This sentence feels unrelated to the rest of the paragraph and is therefore confusing. If you want to go into this you need to explain the different retrieval algortihms. But I think it's better to leave this sentence out of the introduction and leave this to the methods (as you've explained this more clearly in section 2.2).

We agreed and removed this sentence as you suggested.

L148-149: 'The signals ... travel time.' This sentence makes it sound like travel time is used to differentiate observations of sea ice bottom vs. sea surface, but I think you are trying to say that the distance is determined from the travel time measurement. It also sounds like only two measurements are made, one from the sea ice bottom and one from the sea surface. Please rewrite this.

We rewrote the sentence: "The sensors transmit sound pulses upwards with a footprint of 6–8 m in diameter and the signals are reflected either by the sea ice bottom or the sea surface, **yielding two-way travel time which can be converted into distances.**"

L254: 'ICESat is more sensitive to thick ice than Envisat', but the Envisat SIT product is thicker than ICESat? You describe this bias well in section 4.1, but here it's a bit confusing, as you seem to say that ICESat should show thicker ice.

We realized that the statement is correct only for along-track data. Firstly, given the sparseness of ICESat overpasses with valid data such a 100 km grid SIT estimate in that region might be biased by the presence of thick landfast ice. Besides, ocean swell can result in anomalously high freeboard values which then convert into too high sea-ice thickness values. While this is a local phenomenon, the sparseness of ICESat overpasses with valid data can results in a similar effect as for landfast ice. Therefore, we considered the two issues here carefully and decided to remove this statement.

L262: 'Envisat has a positive difference with respect to ICESat'. I do not understand what this means. Have a look at the suggestions for technical corrections too.

We wrote the sentence: "In summer, the agreement between Envisat SIT and ICESat SIT is not good, mainly due to their different performances towards thick ice above 3 m."

*Techinical corrections*
L24: 'while the uncertainties of *the* snow depth product are' or 'while the uncertainties of snow depth product*s* are'

We modified this sentence: "while the uncertainties of the snow depth product are not the dominant cause of the differences".

L30: 'it is still unclear if ... sea ice thickness'. Change 'also associated with' to 'accompanied by', these changes do not have to be related (or associated) but can be seperate.

We modified this sentence: "However, it is still unclear if the recent increase in Antarctic sea ice area is also accompanied by a similar change in sea ice thickness."

L39-40: 'from the ASPeCt can provide' change to 'by the ASPeCt expert group can provide'

We modified this sentence.

L42: 'airborne electromagnetic data which measure total freeboard', data don't measure things, maybe rephrase.

We changed "measure" to "provide".

L52: Remove 'basically', this sounds very unscientific.

We removed this word.

L54-55: Consider more recent studies that have retrieved Antarctic sea ice thickness, e.g. Kurtz & Markus (2012) and Kacimi & Kwok (2020).

We added Kacimi & Kwok (2020) here and removed Zwally et al. (2008).

L83-84: 'also how the different ... distribution.' Very vague, what are 'the different retrieval methods', ICESat and Envisat?

Yes, the different retrieval methods that Envisat and ICESat SIT products use have impacts on their differences.

L88: 'the former inter-comparison study', which study is this?

The study is Kern et al. (2016), and we modified the sentence to make it clear: "Based on the former inter-comparison study (Kern et al., 2016), we choose the ICESat sea ice thickness data derived from the modified ice density approach for comparison."

L147: change 'underwater' to 'below sea level'

We agreed with your comment but we found it unnecessary to mention the mooring location in detail.

L150-151: 'once several minutes', do you mean 'every several minutes'? Please rewrite and maybe be more specific (what is several minutes)?

We modified the sentence: "The intervals of sea ice draft measurements are between 3 and 15 minutes from November 1990 to March 2008."

L153: seasons -> season

We corrected this issue.

L166: 'FM-MJ and MJ-ON'. I guess you are referring to February/March-May/June and May/June-October/November. Please specify the first time you mention these abbreviations.

We modified the sentence: "We compare FDD with the SIT variations from February/March to May/June (FM-MJ) and from May/June-October/November (MJ-ON) represented by Envisat and ICESat SIT."

L182: 'Before ... first.' Repetitive, just use 'before' or 'first'.

We deleted "first".

L193: Remove 'during the comparison'.

We removed this phrase.

L197: 'We provide ... SIT products.' Rewrite this sentence. I would suggest something like 'The error bars in the figure show the uncertainty estimates of/from the SIT products'.

We rewrote the sentence as: "The error bars in the figure show the uncertainty estimates from the SIT products."

L197-200: 'The Envisat SIT ... Li et al., 2018).' Move these sentences to the methods? Also: I think adding an estimate of the ULS uncertainty to Figure 3 as well would improve the interpretation of this figure. You mentioned an estimate of the ULS uncertainty in

We added the ULS uncertainty of ± 0.05 m following Belter et al. (2020) in our paper.

L152-154. You now mention when the error bars of the altimetry sensors do not overlap with the ULS points, but it would be interesting to see if they do overlap with the ULS error bars.

We find that the ULS uncertainty cannot explain these differences.

L207: Why are the uncertainties of freeboard and snow depth not considered for the ICESat SIT uncertainties?

Snow depth uncertainty is not included because the ICESat SIT retrieval method does not require additional snow-depth information. However, we checked Kern et al. (2016) and their computation of the SIT uncertainty included in the product is based on uncertainties in densities and freeboard. Also, the uncertainty estimates provided with the Envisat SIT data set are possibly too large because the data set producers those days did not adequately take potential correlations between the error contribution into account. Therefore, we rewrote the sentence as: "The large differences in the error bars between Envisat and ICESat mainly result from the inclusion of snow depth uncertainty in Envisat SIT, and lack of adequate regards for potential correlations between the error contribution."

L208-209: 'ICESat does not capture ... on thicker ice.' I'm not sure where I can see this in Figure 3?

We rewrote the sentence as: "In the eastern Weddell Sea (at sites 229 and 231), ICESat has a few overestimations on thicker ice."

L210: 'error bars can cover' -> remove 'can'

We removed "can".

L210: 'However, since many contributions are not well characterized and quantified'. What contributions is this about and how are they not well characterized and quantified?

These uncertainty contributions include spatial and temporal variability on snow depth as well as snow and sea ice density. Few information about these data exists in the Antarctic. Besides, the coverage of sea ice type (first and multi-year) products is incomplete for the Envisat observation period.

L225-226: 'considering the typical sea ice motion'. Briefly characterize this typical sea ice motion (fast, direction?), so the reader can see why the monthly average ULS SIT can be referred to as a spatial average value.

We added the sea ice motion information in the Weddell Sea: The climatological cyclonic atmospheric circulation and the ocean gyre in the Weddell Sea result in westward ice advection along the southern coast and northward advection of ice along the eastern peninsula.

L235: What are 'the ship-based observations'? This is not introduced in the paper before.

The ship-based observations are the ASPeCt data from Worby et al. (2008). According to their Table 3, the average ice thickness in spring West Pacific is 0.68 m, smaller than Envisat and ICESat SIT in our

study.

L237: change 'feature' to 'dissimilarity' or another more descriptive word.

We changed "feature" to "dissimilarity".

L249: 'but with thickness estimates of up to 1.5 m'. Make sure it is clear to the reader that this is thinner than elsewhere.

We added a sentence to clarify it: "Envisat detects sea ice in the Ross Sea all the years, but with thickness estimates of up to 1.5 m, much larger than expectant seasonal ice thickness."

L264: 'the two datasets coincide with each other', this sounds a bit like they temporally coincide instead of the distributions being similar (which is I think what you want to say here). Please rewrite.

We rewrote the sentence: "In spring, the two data sets have similar distributions, represented by closer mean and modal thicknesses."

L269-270: 'We calculate the period-average SIT from the model'. This might be my lack of experience with freezing-degree-days: the FDD in Figure 8 and Table 6 show the total negative temperatures between these months right? I do not understand how it shows SIT. I understand that FDD and SIT are related but I don't see how the model actually calculates average SIT? Please make this more clear in the methods. If 'the model' is not FDD, maybe specify what model you mean?

FDD is calculated by daily degrees below freezing summed over the total number of days the temperature was below freezing. According to Lebedev (1938), a simple model is constructed to produce sea ice thickness:

Thickness (cm) = 1.33 * FDD (℃)$^{0.58}$

Note that the calculated thickness only accounts for the freezing of sea water and excludes ice variations from snowfall, freezing rain or ridging. Therefore, we don't translate the FDD into thickness growth in this study because i) we think FDD is sufficient to stand for thickness growth and ii) using the very simple translation equation adds uncertainties into our analyses.

L271: 'Envisat SIT has opposite developments from ICESat and FDD during MJ-ON'. Envisat and ICESat do not really show the opposite? They both show the strongest thinning in the western Weddell Sea and both show thickening near the coast in the Amundsen Sea. Please rewrite this to describe the difference, I think something like that Envisat shows more thinning all around the Southern Ocean and ICESat generally more thickening?

We modified the description: "The results show that as FDD is always positive, Envisat SIT shows more thinning all around the Southern Ocean and ICESat SIT generally shows more thickening during MJON, while they share similar variations during FMMJ."

L271, Figure 8, and Table 6: Please be consistent in how you refer to these periods (MJON or ON-MJ and FM-MJ or MJ-FM). I would suggest for summer to autumn you use FMMJ (instead of the subtraction MJ-FM you used in Figure 8) as this order is more intuitive.

Thanks for your advice. We used FMMJ and MJON to replace the previous usage.

L272: 'both products', which two products? Envisat and ICESat, or satellite altimetry and FDD?

Envisat and ICESat. We changed it to "both satellite products" to clarify.

L274: 'The adverse patterns', adverse (preventing success or development; harmful; unfavourable) might not be the right word here?

We changed "adverse" to "opposite".

L279-280: 'the regression lines have large positive intercepts in all three seasons, indicating that Envisat SIT tends to exceed ICESat SIT for thin ice'. I can see in Figure 9 that this is true, but the latter does not necessarily follow from the former. A large positive intercept could also be caused by Envisat SIT being

lower than ICESat SIT for thick ice. Again, in the figure I can see this is not the case here, but maybe just rephrase the explanation to just say 'For all five locations, Envisat SIT tends to exceed ICESat SIT for thin ice', without referring to the intercept.

We rewrote the sentence as you suggested: "For all five locations, Envisat SIT tends to exceed ICESat SIT for thin ice."

L281: change 'splashes' to 'cloud' which is more often used to describe a collection of points in scatterplots. 'Exceed' in what way? Envisat or ICESat or both?

We changed "splashes" to "cloud". This kind of "exceed" reveals that mean ICESat SIT are nearly constant through all three seasons in the western Weddell Sea, while mean Envisat SIT are noticeably larger in summer and autumn

L284-285: 'The numbers in the last ... values per season'. This might be something to replace to the caption of the table. Also, in the table it does not look like this is in the last column, but in the first row?

We moved this sentence to the caption. We meant to explain the numbers in the last column and we modified the description: "The numbers in the last column of the table are the sample sizes of the comparisons, taking into account the actual number of values per season."

L294-295: 'it is known that ... homogenous stratigraphy'. This statement could use a citation.

We added a citation: Willatt et al. (2010).

L296: 'considering the large ... of about 70 m'. Maybe specify that the pulse-limited footprint is Envisat and the laser beams ICESat.

We modified the sentence: "considering the Envisat large pulse-limited footprint of about 2–10 km and smaller footprint of ICESat laser beams of about 70 m".

L341: maybe just say 'may come from the AMSR-E snow depth' here as you haven't yet discussed why it might be biased.

We removed "biased".

L347-348: 'the differences that AMSR-E snow depths minus the ASPeCt observations are positive ...', rephrase this sentence to something like 'AMSR-E snow depth minus the ASPeCt oservations is positive'

We rephrased the sentence as you suggested.

L349: 'the satellite passive microwave snow depth'. Maybe introduce AMSR-E as a passive microwave sensor in the methods, so readers that don't know the AMSR-E snow depth product know what you are referring to here.

Thanks for your advice. We have introduced AMSR-E snow depth in section 2.1 and 4.2. We just added the passive microwave information in section 2.1: "This snow-depth climatology is derived from the passive microwave sensor Advanced Microwave Scanning Radiometer-EOS (AMSR-E) and AMSR-2 for the Antarctic".

L351: '... lead to underestimations' and '... lead to overestimations', under- and overestimations of what? SIT?

We meant to say under- and overestimations of snow depth and we added "of snow depth" in the sentence.

L357: The retrieval uncertainty of AMSR-E?

Yes. According to Kern et al. (2015) the average monthly retrieval uncertainty of AMSR-E is 2 cm. However, we realized this value is the precision but not the potential bias which can be much larger. Therefore, we checked sea-ice thickness changes in response to snow biases between 5 and 30 cm in steps of 5 cm.

L357-358: 'suggesting that sea ice thickness change is insensitive to the snow depth', I would suggest change to 'the sensitivity is low', as SIT does change with snow depth, just not by a lot.

We modified the sentence: "suggesting that the sensitivity of sea ice thickness change to the snow depth is low".

L363-364: 'The usage of snow climatology allows reducing the relative uncertainties', it's a bit unclear what these 'relative uncertainties' are and how they are reduced.

Generally, using a snow climatology for converting ice freeboard to thickness neglects any interannual snow variability. It can reduce the actual snow depth biases to some extent.

L389: Remove 'firstly'

We removed it.

L392: change 'not comparible to' to 'overestimating' or something else more descriptive of the difference between the two.

Thanks for your advice, but we delete this sentence since the previous sentence has stated the result of the comparison with ULS.

---

## Author Comment (AC4)

Dear Reviewer:

We would like to express our gratitude to you for the comments to improve this manuscript. According to your and other reviewers' comments, we have conducted further research on the issues that you suggest. Please find the specific responses and revisions shown below. They are in blue font for clarity.

Qinghua Yang, Qian Shi, Robert Ricker, Stefan Hendricks
On behalf of all the authors

Major comments:
-One of my main concerns has to do with the actual validity and usefulness of the comparison between the satellite estimates and the ULS data. As clearly stated by the authors, there are significant differences in temporal and spatial sampling. The authors even point out that the results are not consistent. I believe it would be more beneficial to the paper to focus solely on the intercomparison between Envisat and ICESat data.

Thanks for your comments. However, we think that you are biased in denying the feasibility of using ULS data as comparison data with ICESat-1/Envisat due to their relatively narrow footprint. As we know, ULS indeed measures the continuous ice draft in a fixed location with a diameter of several meters. Considering the ice motion, ULS acquired dozens to hundreds of kilometers records along the trajectory of sea ice motion on a monthly basis, which have enough spatial representativeness compared with ICESat-1/Envisat. Here, we track the source of sea ice that flows over the ULS in a specified month by backward tracking method based on NSIDC Pathfinder data sets. We find the ice draft records included in ULS monthly mean calculation come from a wide range area (Fig. 1). Therefore, we think this is enough to prove that the spatial representativeness of the monthly average ULS data can be compared with that of ICESat-1/Envisat.

Besides, ULS data was generally used for ice thickness comparison in the previous studies. ULS is used for comparison with the ice thickness derived from AVHRR (Yu and Rothrock, 1996; Drucker et al., 2003). It was also used to compare with ICESat-1 ice thickness in the Fram Strait (Spreen et al., 2009). In addition, the ULS data sets have also been used for comparison with reanalyses data in the polar region (Mu et al., 2018; Shi et al., 2021). In addition, the comparison with ULS data sets is also a convention for assessing the quality of ice thickness derived from altimeters in the European Space Agency (Kern et al., 2018).

In summary, we think that the reason for rejecting us due to the spatial representativeness of ULS ice thickness is untenable. Previous studies (referred to above) have shown that using ULS for validation of satellite-derived sea-ice thickness data sets can be considered as state of the art.

[Figure]

Fig. 1 The 30-day origins of the sea ice passing the three ULS sites in July 2011. The red dots stand for the ULS locations and the blue dots stand for the original locations of the sea ice using backward tracking method.

- Another major concern is the way that the comparison between the Envisat and ICESat-2 SIT is carried out. I think the paper would be more robust if a comparison of the actual freeboards and snow depths (total freeboards for ICESat) was introduced. The assumption made on snow depth can have a huge impact on the mean and variability of the derived sea ice thickness.

Thanks for your comments. Indeed, the involvement of snow depth can have a huge impact on the retrieved sea ice thickness. However, the purpose of this paper is to give a comprehensive and statistical comparison between Envisat and ICESat sea ice thickness data, and to highlight the importance of dealing with the uncertainties of these products. Additionally, comparing the total freeboard still needs an additional snow depth product, since the radar altimeter aboard Envisat detects sea ice freeboard.

- While the authors explored the possible causes of the observed differences between the two satellite datasets, I think this should be looked at more carefully and in more detail. Based on their uncertainty analysis, the authors conclude that most of the bias is probably explained by radar penetration issues. I do not believe that the authors successfully demonstrated this, especially given that the assumptions on snow depth and snow density are different for the two instruments.

We need to clarify that the snow densities used by both Envisat and ICESat SIT retrieval are 300 kg m$^{-3}$. In addition to the radar penetration biases, we also discuss the causes of the snow depth product and ICESat uncertainties in section 4. For modification, we conclude the sensitivity of the SIT changes to freeboard biases, snow depth biases and sea ice density in Fig. 2 by analyzing Eq. (1):

$$I = \frac{F\rho_{water} + S\rho_{snow}}{\rho_{water} - \rho_{ice}} \tag{1}$$

The sensitivities to freeboard biases and to snow depth biases are calculated by:

$$\frac{dI}{dF} = \frac{\rho_{\text{water}}}{\rho_{\text{water}} - \rho_{\text{ice}}} \tag{2}$$

$$\frac{dI}{dS} = \frac{\rho_{\text{snow}}}{\rho_{\text{water}} - \rho_{\text{ice}}} \tag{3}$$

From Fig. 2, we can see that though the magnitudes of the resulting thickness changes are quite similar, the SIT changes are more sensitive to sea ice freeboard biases than to snow depth biases. Besides, with the increase of sea ice density, the SIT changes rise. For typical sea ice freeboard biases (7 cm for the Arctic nominal adjustment suggested by Nandan et al. (2017, 2020)), the sea ice density variations induce the thickness changes ranging from ~0.5 m to ~0.8 m. For typical snow depth biases (20 cm for the monthly mean retrieval uncertainty in Kern and Ozsoy-Cicek (2016)), the thickness changes from ~0.4 m to ~0.7 m. **Although this sensitivity analysis is not solid enough for the explanation for the SIT differences in three seasons, it can provide a reasonable conjecture that freeboard biases are the main cause of the positive differences in summer and autumn.**

[Figure]

Fig. 2 Sensitivity of sea ice thickness changes to sea ice freeboard biases and snow depth biases as function of sea ice density. (a) SIT changes computed with Eq. (1) for different sea ice freeboard biases (2 cm to 10 cm). (b) Similar to (a) but computed for different snow depth biases (5 cm to 30 cm).

- Some of the phrasing needs to be reviewed carefully. Especially in the introductory part of the paper, some sentences are poorly constructed and lack clarity. It challenges the understanding of the paper.
We apologize for the language problems in the original manuscript. We have carefully amended the phrasing and modified the expressions throughout the paper.

Minor comments:
P1L9: the sentence" The crucial role that Antarctic sea ice plays in the global climate system is strongly linked to its thickness" does not really mean anything. Maybe you mean that thickness is important to evaluate the role of Antarctic sea ice in the global climate system?
Yes, we meant to point out that sea ice thickness is a critical component in assessing the role of Antarctic sea ice in the global climate system. Therefore, we focus on sea ice thickness in this study.
P1L10-11: What do you mean by "on a hemispheric scale, satellite radar altimetry data can be applied with a promising prospect"? Do you mean that large scale estimates of SIT are achievable with radar

altimetry? Again revise the wording to make clearer statements.

This sentence means satellite radar altimetry can be used to achieve large-scale and long-term SIT variations, while field observations cannot.

P1L28: Replace "declines" by "decline".

We corrected this word.

P2L59: Replace "CyroSat-2" by "CryoSat-2".

We corrected this word.

P2L60-61: I suggest rephrasing this sentence:" The SICCI product covers the entire Antarctic sea ice for the complete annual cycle from 2002 to 2017, and it is finally a combined data set of Envisat and CyroSat-2" to "The SICCI product is derived using measurements from Envisat and CryoSat-2 and covers the entire Antarctic sea ice for the complete annual cycle from 2002 to 2017".

We rewrote the sentence as you suggested.

P3L76:" This data set has been investigated for many years". I believe this dataset has been used in several investigations, not investigated.

We modified the sentence: "This dataset has been used in several investigations."

P4L94:"between the two datasets" please specify that you are referring to the satellite data.

We changed it to "the inter-comparisons between the two satellite data sets".

P5L127: Replace "are conducted with" by "are characterized by"

We amended this phrase.

P6L163: Replace "derived" by "from"

We removed "derived".

P6L171: Please revise:" For each period, we choose the corresponding time period during which Envisat monthly data are used".

We revised the sentence: "For each ICESat operating period, we choose the corresponding Envisat monthly data."

P6, L175-177: Please revise :" The weighting has taken into account periods where only Envisat SIT of one month are present, i.e., we use this equation for grid cells where we have valid SIT data from both months, while we only use the Envisat SIT of the respective month without weighing for those grid cells where we only have valid data from either month."

We simplified the sentence: "We use this weighing equation for grid cells where valid Envisat SIT data exist in both months, while the weighing is not conducted for grid cells where valid data only exist in one month."

P8L236: I suggest to replace "Envisat does not show the young ice in the Ross Sea" by "Thin ice in the Ross Sea is not captured by Envisat".

We rewrote the sentence as you suggested.

P9L244-255: Revise "Compared to summer, the differences in the western Weddell Sea spread to the whole Weddell Sea sector and decrease from west to east.". The statement is not clear.

We clarified the sentence: "Compared to summer, the positive differences between Envisat and ICESat SIT in the western Weddell Sea turn to positive differences over the whole Weddell Sea sector, and the differences decrease from west to east."

P12L345: Replace "Previous study reveals" by "Previous studies show".

We corrected this word.

P14L389: Remove "Firstly". The comparison to ULS data is carried out first.

We removed this word.

---

## Author Response (AR2)

Dear Editor,

We truly appreciate your and two anonymous reviewers' valuable comments and suggestions for the paper 'A comparison between Envisat and ICESat sea ice thickness in the Southern Ocean' submitted to The Cryosphere. We have already made a substantial revision according to these comments and suggestions, and reply to them one by one below.

Qinghua Yang and Qian Shi

On behalf of all the authors

**Responses to editor**

1. A refinement of the pairing (in time and location) of Envisat and ICESat data is required. Pls provide further explanation.

   *Response:* Thank you for your comment. We have refined the description and the figures of paring between Envisat and ICESat SIT following the referee's and your comments: "We employ a time-weighted average of the monthly Envisat data to match the ICESat period. For example, considering the ON04 period from Oct 3 to Nov 8 in 2004, which is 37 days long – 29 days in October and 8 days in November, we calculate the corresponding Envisat SIT as: $SIT_{ON04} = (29/37)*(SIT_{October}) + (8/37)*(SIT_{November})$. We use this weighing equation only for grid cells where valid Envisat SIT data exist in both months, while the weighing is not conducted for grid cells where valid data only exist in either one month. It is noted that this approach can lead to considerably larger coverage of Envisat SIT data than ICESat, thus we only show grid cells where both Envisat and ICESat have valid SIT and only take those values in the statistical computation." *(please see P6 line 166-172 in the revised manuscript)* "Both Envisat and ICESat SIT have been interpolated onto each ULS location in the nearest neighbour way." *(please see P7 line 182-183 in the revised manuscript)*

2. Expand and consolidate the definition and validity of the dynamic freezing-degree-day [FDD]. The dyn FDD concept is introduced in very brief notes (9lines), but as it carries some weight in the argumentation this requires expansion. The "seasonal" approach taken here requires justification.

   *Response:* Thank you for your valuable comment.

   (1) Our procedures on deriving dynamic FDD are listed as follows: a. On Day1, the historical FDD and the newly increased FDD are interpolated to the regular NSIDC ice velocity grid position (X1, Y1); b. Accumulated FDD moves to irregular positions (X2, Y2) with sea ice motion derived from NSIDC v4; c. At the beginning of Day2, the FDD distribution of the irregular Day1 is interpolated to the regular grid (X1, Y1), then repeating steps 1-3. The ice divergence and deformation situations cannot be represented through our method.

   (2) However, according to another referee's suggestions, we have decided to remove the FDD parts and focus on the intercomparison between Envisat and ICESat since the results of FDD cannot explain the reason for the differences. However, according to one of the referee's comments, we have decided to remove this part and conducted further research on the difference during MJON. The results according to the new figure are added as follows: "Therefore, we further compare the mean variations of Envisat SIT, ICESat SIT, Envisat freeboard, ICESat freeboard and snow depth climatology used in Envisat retrieval from autumn (MJ) to spring (ON), shown in Fig. 8. The average fields are calculated with grid cells where both Envisat and ICESat SIT have valid

values in all three years from 2004 to 2006. Figure 8 shows that Envisat SIT experiences general decreases from May/June-October/November (MJON) except Bellingshausen Sea and part of Amundsen Sea. Significantly large decreases exist in Western Weddell Sea. In contrast, ICESat SIT present large-scale increases except Western Weddell Sea and Ross Sea where slight decreases exist. By comparing the SIT and freeboard changes of both products, we find that the different changes of freeboard dominantly explain the SIT differences. One thing we can give a speculation based on the analyses in autumn and the regular rule during freezing seasons is that Envisat freeboard is probably overestimated in autumn, which has been pointed out in several studies before (e.g., Willatt et al., 2010; Kwok and Kacimi, 2018; Kacimi and Kwok, 2020). Moreover, the snow depth climatology also shows a decrease in Western Weddell Sea and Ross Sea (Fig. 8e), which has been reported by Kern and Ozsoy-Cicek (2016) that AMSR-E snow depth is likely to underestimate the snow depth evolution during MJON, also contribute to the Envisat SIT decrease." *(please see P9-10 line 269-281 in the revised manuscript)*

[Figure]

**RFig. 1:** The average changes of Envisat SIT, ICESat SIT, Envisat freeboard, ICESat freeboard and snow depth climatology used in Envisat retrieval from autumn to spring (MJON) calculated from 2004, 2005 and 2006.

3. The footprints of ULS vs IceSat vs Envisat are stretching magnitudes. I suggest to show the full thickness distribution of ULS-derived data and contrast this with the

ULS-derived means.

*Response:* We appreciate for your valuable comment. We have investigated the ULS thickness distribution of three sites (excluding open water) following your comments and find that the modal and mean thickness of all distributions agree with each other well and usually present a little lower than the modal ice thickness. Therefore, we think the mean ice thickness of ULS can represent the dominant ice thickness condition during the study period.

[Figure]

**RFig. 2:** Probability distribution function of the daily ULS SIT excluding open water. The dashed lines indicate the mean ice thicknesses. The bin size is 0.1 m and the probability distribution is normalized.

4. Fig1: Some indication about the uncertainty of the reflectance horizon in Envisat radar signals should be included. I.e., Willatt et al. (2010): "The authors suggest that one reason for this is that the radar does not always penetrate to the snow/ice interface but sometimes to somewhere between the air/snow and snow/ice interfaces. The range resolution of satellite radar altimeters is not sufficient to resolve layers within the snow pack."

*Response:* Thank you for your comment. We have added the information about the penetration uncertainty in Fig. 1.

[Figure]

**RFig. 3:** An illustration of measuring freeboard using ICESat and Envisat, and the ULS measurement principle. Noted that radar altimeter on Envisat usually penetrates to somewhere between the air/snow and snow/ice interfaces (Willatt et al., 2010).

5.  Colourbars for Fig 4 - 6 should share the same max/min bounds for "SIT" and the "Difference".
    *Response:* Thank you for your comment. We have unified the bounds for color bars in Fig. 4-6., setting 0~3 m as the "SIT" bounds and -2~2 m as the "Difference" bounds.

**Responses to referee #1**

Dear Reviewer:

We would like to express our gratitude to you for the helpful comments to improve this manuscript. We have modified the spatial pairing between Envisat and ICESat SIT by showing grid cells where both products have valid values. Besides, we have replaced the FDD analyses by the freeboard and snow depth changes during MJON. We also added a sensitivity analysis of ICESat SIT to sea ice density. The specific responses and revisions are shown below. They are in blue font for clarity.

Qinghua Yang and Qian Shi
On behalf of all the authors

1.  L118-122: Both, the ATBD as well as the CRDP data set description state that the snow depth climatology is based on AMSR-E and AMSR2 data. You might ask your co-authors from AWI who have been part of the CCI project for this information. Therefore, you should correct your wording accordingly.

    ***Response:*** Thank you for pointing this out and we are sorry for making the mistake. We have checked the description of dataset and corrected the sentence as: "This snow depth climatology is derived from the passive microwave sensor Advanced Microwave Scanning Radiometer-EOS/2 (AMSR-E, 2002-2011; AMSR2, 2012-2017) and is based on a revised approach with different tie point retrieval plus addition of retrieval errors and provided by the Integrated Climate Data Center (ICDC)." *(please see P4-5 line 121-124 in the revised manuscript)*

2.  L163-165: I recommend that you explain this in more detail, because ERA-5 reanalysis temperatures are available at a specific temporal sampling (hourly, 3-hourly, 6-hourly, daily?) and you need to explain which temporal resolution applies and how you computed the FDD from presumably sub-daily resolution air-temperatures.

    ***Response:*** Thank you for your comments. We used daily average temperature fields from ERA-5 to calculate the FDD values because the definition is based on daily resolution. Albeit too simple, FDD shows how cold it has been and how long it has been cold. However, according to your suggestions, we have decided to remove the FDD parts and focus on the intercomparison between Envisat and ICESat since the results of FDD cannot explain the reason for the differences.

3.  L167: "we add sea ice motion data and convert it to a dynamic FDD" --> This has to be explained in more detail. It is not clear how you derive this dynamic FDD. You should invest a paragraph describing what you did. How did you, in particular, cope with divergent or convergent situations?
    In addition, in my view, the motivation you wrote in L166 (snowfall, freezing rain,

ridging) to carry out this forward advection of grid cells having a specific FDD does not match with what you did. All you take into account here is that the sea ice is moving. While being at a certain location at day 1 it will be at a different location at day 2 and hence may experience a different forcing by the air temperature. With your approach you don't take into account any of the mentioned processes that could also be responsible for a SIT change. Hence, the resulting FDD field or distribution might be more realistic than a FDD field without advection but not for the reasons laid out. Still you only take into account thermodynamic ice growth. Please correct your writing accordingly.

*Response:* We gratefully appreciate for your comment.

(1) Our procedures on deriving dynamic FDD are listed as follows: a. On Day1, the historical FDD and the newly increased FDD are interpolated to the regular NSIDC ice velocity grid position (X1, Y1); b. Accumulated FDD moves to irregular positions (X2, Y2) with sea ice motion derived from NSIDC v4; c. At the beginning of Day2, the FDD distribution of the irregular Day1 is interpolated to the regular grid (X1, Y1), then repeating steps 1-3. The divergent and convergent situations cannot be represented through our method, which might have impacts on the FDD changes in western Weddell Sea and Amundsen-Bellingshausen Seas.

(2) We agree with your assessment that we only improve spatial distribution of the thermodynamic growth and we apologize for our incorrect descriptions. What we should say is: "In order to achieve a more realistic FDD field with advection, we convert it to a dynamic FDD." However, according to your suggestions, we have decided to remove the FDD parts.

4. L170-172: The areas covered by the sea ice differ substantially between the seasons. Did you carry out the comparison only for those parts of the sea-ice covered regions where you have valid SIT (or FDD) data in both seasons? Or do you derive, for instance, a mean FM SIT and a mean MJ SIT (regardless of the sea-ice cover) and compute the difference from these values. Please be more specific in the description of the methodology.

*Response:* Thank you for your careful reading and we are sorry for not clarifying the comparison method. We conduct the comparisons for the ice-covered regions where valid SIT exist in both seasons during FMMJ and MJON. Although we didn't show the same coverage for Envisat and ICESat/FDD in Fig. 8, we calculate the statistical values for the same coverage in Table. 6. Our methodology should be described as: "We compare the dynamic FDD with the SIT variations from February/March to May/June (FMMJ) and from May/June-October/November (MJON) represented by Envisat and ICESat SIT. Specifically, FMMJ represents the differences that mean FDD/Envisat SIT/ICESat SIT in MJ minus that in FM consistent with ICESat operating periods and so does MJON. The comparisons are conducted for the same coverage where all three products have valid SIT difference values." However, according to your suggestions, we have decided to remove the FDD parts.

5. L181/182: I keep my notion of my previous review that it is confusing to see that basically all Envisat SIT maps show a sea ice coverage that corresponds to the maximum sea ice cover of the two months considered. I can understand that you don't want to redo the maps but particularly for summer this seems to be very recommendable in order to avoid further misleading comments such as the one in Lines 254-256 wherein you state that ICESat SIT maps of summers 2005 and 2006 don't show sea ice in the Ross Sea because the SIC is below 60% while Envisat detects sea ice (even though the threshold used is 70%). This is misleading.

Therefore: Please redo Figures 4 to 6, showing Envisat SIT values only for those grid cells where ICESat also shows SIT values - similar to the spatial distribution of the difference map - and only take those values in the computation of the statistics.

*Response:* We appreciate your kind suggestions and we have amended the Figures 4 to 6 accordingly as follows. Besides, we have also amended the descriptions of comparison method here: "It is noted that this approach can lead to considerably larger coverage of Envisat SIT data than ICESat, thus we only show grid cells where both Envisat and ICESat have valid SIT and only take those values in the statistical computation." *(please see P6 line 170-172 in the revised manuscript)*

[Figure]

**RFig. 1:** Comparisons of Envisat versus ICESat sea ice thickness for each ICESat operating period in spring (October & November). The first and second columns show the sea ice thickness distribution of Envisat and ICESat respectively, and the last column shows the difference field (Envisat minus ICESat) of sea ice thickness. Each row represents a year from 2004 to 2007. The maps are all interpolated onto the polar-stereographic grid of the ICESat product and only show grid cells where both data have valid SIT. The white cells denote sea ice concentration less than threshold or missing data.

[Figure]

**RFig. 2:** Same as RFig. 1 but for summer (February & March).

[Figure]

**RFig. 3:** Same as RFig. 1 but for autumn (May & June).

6. L204 / Fig. 3: I am certainly a fan of error bars. But you can simply delete those for the ULS data because i) the spatial scales are so different that it does not make sense to add ULS SIT error bars, ii) "following Belter et al. (2020)" is likely not the right thing to do (and cite) here because that paper is using different instruments and a different geographic region (Arctic, Laptev Sea). You need to figure out what is written inside the 3 or 4 papers related to the ULS measurements in the Weddell Sea. One solution to obtain a more reasonable ULS SIT error bar could be the compute the standard deviation of the SIT at the mooring location recorded over the period of the respective months.

*Response:* We appreciate for your comment. We have calculated the standard deviations of the ULS SIT for each month as the error bars since no uncertainty values are given in former studies on Weddell ULS. Therefore, we amended the descriptions to: "We also add the ULS error bars by calculating SDs of the ULS SIT for each month." *(please see P7 line 195 in the revised manuscript)*

[Figure]

**RFig. 4:** Time series of sea ice thickness and their uncertainties for the Weddell Sea ULS data, Envisat and ICESat. The numbers on the top represent the location of each site for the comparisons. The site locations can be searched in Fig. 2. ICESat SIT values are placed between the two months that each period covers. The error bars of ULS thickness are the standard deviations of daily ULS SIT in each comparison period.

7. L205/206: From what you know about sea-ice conditions in the western Weddell Sea: Do you believe that these SIT values are reasonable? In addition: Please make a comment why you don't observe a clear seasonal cycle at this location.
*Response:* Thank you for your comments.
(1) Williams et al. (2015) compared the ULS data at site 207 with autonomous underwater vehicles (AUV) data in November 2010. Their Fig. 2b shows that a large proportion of thin ice and undeformed ice exist in the northwestern Weddell Sea, having modal drafts of 1.2–1.5 m. Therefore, although this site is located in the western Weddell Sea, the existence of thin ice ranging between 0 and 1.5 m are reasonable.
(2) This site is located at the boundary between first-year ice to the east and predominantly multiyear ice to the west and the sea ice here are experiencing strong motion due to Weddell Gyre. According to the large SDs of ULS SIT which is an indication of large sea ice temporal variability, we think that a mixture of deformed and undeformed sea ice is the main reason for the lack of seasonal cycle at this location. Therefore, we have amended this sentence as: "In the western Weddell Sea along the coast of the Antarctic Peninsula (at site 207), the ULS thickness ranges between 0 and 1.5 m, without a clear seasonal cycle due to a mixture of deformed and undeformed sea ice (Williams et al., 2015)." *(please see P7 line 196-198 in the revised manuscript)*

8. L208/209: "ICESat thickness ... exceeds ... ICESat has a few overestimations ..." --> What is missing here is the notion that the ICESat SIT is the total SIT and hence

includes the snow depth. Given typical snow depths on sea ice in the two regions considered it appears reasonable to assume that particularly at site 207 the "true" ICESat SIT would be considerably (by 30-40cm - depending on the season of course) smaller than the total ice thickness. While you do not need to speculate about the actual difference it is important to make the statement that ICESat is the total thickness.

*Response:* Thank you for your careful check and we have added a sentence here to point out this problem: "Noted that the realistic ICESat SIT would be considerably smaller due to the retrieval method mentioned in Sect. 2.2, about 0.2–0.4 m at site 207 and 0.15–0.3 m at site 229/231 depending on the seasons (Fig. S3)." *(please see P7 line 201-203 in the revised manuscript)*

9. L229-231: These lines require more writing and more explanations. Please write explicitly in the paper how you computed the region from within during the period of one month sea ice arrives at the respective ULS. You need to explain how you did carry out this computation and how you end up with your number of 100 km. You also need to justify why doing this computation for one sample month is representative for the entire period of overlap between ULS and satellite SIT data.

I note that the blue data points for the ULS moorings along the Greenwich Meridian are located to the southwest of the ULS locations and hence NOT upstream but off the track - in contrast to the blue points for the ULS near the Antarctic Peninsula which are located upstream. To me this figure (S1) provides not a credible justification for your approach.

Then you write "proves the heterogenity of the sea ice measured by each ULS" --> I don't understand what you want to stress here. I don't see any heterogenity in SIT given the location of the blue dots in Figure S1. Certainly it is not a proof. Also, "proves ... the validity of ULS data usage in comparison with satellite products" is not a credible statement given how Figure S1 looks like and given the fact that you did not describe adequately what you did. This must be improved.

*Response:* Thanks for your careful check.

(1) In Figure S1, we plot the locations after 30-days backtracking of sea ice at the ULS sites for the period from July 2 to July 31 and the background monthly sea ice motion (from July 1 to July 30) only corresponding to the data in July 31. Here, we present the complete trace (black dots) for the period from July 2 to July 31 (RFig. 5) and they are consistent with the monthly mean sea ice motion field, though the trace presents high variation. So, other blue dots seem not consistent with the sea ice motion because the time doesn't match.

(2) The "heterogeneity" indeed made reader confused. We want to emphasize that though the footprint of ULS is smaller than Envisat and ICESat, continuous ULS observations can still represent the sea ice thickness over a large area due to its continuous sampling and irregular sea ice motion. In a sense, ULS sampling can be analogized to randomly drilling in a range of sea ice cover. In

order to make our explanation clearer, we amended the statement as: "With the sea ice motion data from NSIDC introduced in Sect. 2.4, the 30-day origins of the sea ice passing the three ULS sites from July 2 to July 31, 2011 is shown in Fig. S1 and it is spatially representative". *(please see P8 line 222-223 in the revised manuscript)*

[Figure]

**RFig. 5:** 30-days backtracking of sea ice at the ULS sites for the period from July 2 to July 31.

10. L232: As written in the context of Section 2.5 I ask you to redo figures 4 to 6 to avoid confusion and misleading statements about the different coverage with valid data (see L254-256).
    *Response:* Thank you for your valuable suggestion. We have reproduced figures 4 to 6 (please see our answers for Comment #5) and amended the statements about the SIT differences in the Ross Ice Shelf polynya carefully (please see our answers for Comment #13).

11. L241/242: "Deformed sea ice along the coast ..." --> How do you know this sea ice is deformed?
    "but thicker than ..." --> How did you compare the data? The Worby et al data set ends in March 2005. So, is this statement related to ship-based observations of months October and November from the entire Worby et al data set basically BEFORE your time period of investigation? Please either be more specific about what you did to arrive at this statement or delete it.
    *Response:* Thank you for your valuable suggestions.
    (1) We are sorry for using an inappropriate word here and we have replaced

"Deformed sea ice" with "Thick sea ice". *(please see P8 line 233 in the revised manuscript)*

(2) We compared the SIT in the western Pacific Ocean from both satellites with ship-based observations in 2004 spring achieved by *Aurora Australis* and we have clarified the comparison following your comment: "……but their mean SIT in 2004 are thicker than the ship-based observations (0.63 m; Worby et al., 2008a)." *(please see P8 line 233-234 in the revised manuscript)*

12. L250/251: It is not clear how you derived the statistics. Does that mean that in order to obtain a seasonal mean SIT value, valid SIT values need to be available in the ICESat-measurement-period-mean maps (shown in Figs. 4 to 6) for all 3 years in fall, at least 3 of 4 years in spring and at least 3 of 5 years in summer? How many grid cells are used to compute the seasonal mean values then?

ICESat has more data gaps than Envisat. Do you compute the seasonal mean values only from grid cells where BOTH data sets are available and fulfil the condition just mentioned above?

*Response:* Thank you for your careful check.

(1) Yes, we only used grid cells if valid SIT values are available for all 3 years in fall, at least 3 of 4 years in spring and at least 3 of 5 years in summer. The numbers of grid cells used in the calculation are listed in the last column in Table. 5 and also listed below.

| Season | N |
|---|---|
| Summer (FM) | 170 |
| Autumn (MJ) | 735 |
| Autumn (MJ) | 886 |

(2) Yes, we computed the seasonal mean values only from grid cells where both data sets are available.

To clarify the statistical method, we have amended the description here: "Note that in order to obtain the seasonal mean SIT, we compute the seasonal mean SIT only from grid cells where values are available from both data sets and available for all 3 years in autumn, at least 3 of 4 years in spring and at least 3 of 5 years in summer. The numbers of grid cells used in the calculation are listed in the last column in Table. 5." *(please see P9 line 243-246 in the revised manuscript)*

13. L254-256: As mentioned earlier, the fact that the Envisat SIT maps show values in 2005 & 2006 where ICESat doesn't, is not caused by the fact that ICESat cuts off the retrieval at 60 % (Actually, Envisat does so already at 70% so that, in principle, ICESat should show more SIT values than Envisat) but is the result of the way how you constructed these SIT maps. All Envisat SIT maps shown in Figs. 4 to 6 show the maximum sea ice cover of the two months considered.

*Response:* We gratefully appreciate for your valuable comment. We are sorry for incorrectly explaining the reasons for the SIT differences in the Ross Ice Shelf polynya. The SIT differences in that region mainly come from the way we construct

our time-weighted SIT maps because we considered all valid SIT data from both months. Therefore, we have amended the statements as follows: "As for the Ross Ice Shelf polynya, ICESat displays thin ice lower than 1 m in 2004, 2007 and 2008, while Envisat detects sea ice of up to 1.5 m, much larger than expected seasonal ice thickness." *(please see P9 line 249-250 in the revised manuscript)*

14. L266: Looking back at the these three paragraphs I did not find any notion that the ICESat SIT values compared here denote the total thickness, hence include snow depth.

    ***Response:*** Thank you for your careful check. We are sorry for missing this point in the analyses and we have complemented the information in the manuscript following your comments:

    "However, these differences have to be seen in the light of the large SDs (~0.6 m) and the fact that ICESat SIT values include the snow depth." *(please see P9 line 241-242 in the revised manuscript)*

    "Considering the ICESat SIT excluding snow depth, the real differences should be larger." *(please see P9 line 252-253 in the revised manuscript)*

15. L272-274: As it is not entirely clear from Figs. 4 to 6 how you did compute these values I ask you to one more time provide the information whether both data sets presented here, i.e. Envisat mean SIT and ICESat mean SIT are based on the same number of identical grid cells.

    In addition I find it worthwile to add the information that for Envisat in addition to the mean SIT also the modal SIT decreases from autumn to spring.

    ***Response:*** Thank you for your comments.

    (1) Yes, both Envisat and ICESat mean SIT presented here are based on the same number of identical grid cells and we have added this in the manuscript: "To investigate the development of two SIT data from the end of melting to end of freezing, we provide the probability distribution of the Envisat SIT and the ICESat SIT for all the valid individual comparison pairs where both Envisat and ICESat have valid SIT, shown in Fig. 7." *(please see P9 line 262-264 in the revised manuscript)*

    (2) We have added the information that Envisat modal SIT decreases from autumn to spring. We amended the sentence as: "In addition, we find that ICESat mean SIT increases while Envisat mean and modal SIT decreases from autumn to spring." *(please see P9 line 267-268 in the revised manuscript)*

16. L325-327: Please confirm from these two papers by Nandan et al. that both the season as well as the type of ice for which this bias in CS2 sea-ice freeboard is suggested, match Antarctic conditions discussed in your paper. I am wondering whether particularly during Antarctic summer sea ice and snow conditions are not completely different from those touched upon in the two cited papers.

    ***Response:*** Thank you for your comments. These two papers focus on the role of **saline snow** on first-year sea ice (FYI) in late winter (April and May) from 2004 to

2017. This condition is more consistent with our **autumn** Antarctic conditions with large-scale FYI. Meanwhile, former studies reporting radar altimeter reflection biases (e.g., Massom et al., 1997; Willatt et al., 2010; Kwok and Kacimi, 2018; Kacimi and Kwok, 2020) conducted their experiments in autumn and winter, leaving the potential biases in summer unsolved. Haas et al. (2001) showed that summer snow salinities are significantly lower than measurements in other seasons, mostly below 0.1‰, but the wetness is high at the snow bottom. Liquid water content is sufficient to dominate the snow dielectric properties (Barber et al., 1995). Therefore, we amended and complemented the discussion accordingly: "For typical sea ice freeboard biases caused by saline snow (7 cm for the Arctic nominal adjustment for first-year ice suggested by Nandan et al. (2017, 2020)), the sea ice density variations induce the thickness changes ranging from ~0.5 m to ~0.8 m. This could potentially account for the differences between Envisat and ICESat SIT in autumn (0.57 m). Therefore, we assume that in autumn freeboard-biases-induced SIT changes happen frequently. In summer, when snow salinities are significantly lower than measurements in other seasons (mostly below 0.1‰) but the wetness is high at the snow bottom (Haas et al., 2001), the freeboard biases also matters because liquid water content affects the snow dielectric properties (Barber et al., 1995). Besides, based on previous studies (Willatt et al., 2010; Kwok and Kacimi, 2018; Kacimi and Kwok, 2020), the displacements of radar retracking points and thus the freeboard biases are significant in spring. Considering the small differences between Envisat and ICESat SIT, we suggest that underestimations of snow depth and biases in ICESat total thickness might play an important role in spring. However, detailed sensitivity discussions are limited due to lack of seasonal and regional sea ice density and adjustments to sea ice freeboard." *(please see P11-12 line 323-333 in the revised manuscript)*

17. L337-342: I am not sure the lead detection capability is really the relevant issue here. As you write, the specular reflection of the radar signal over a smooth lead can be quite effective and the lead detection is possibly not that bad - it is worse though for Envisat than for CS-2. More problematic might be reflections from off-nadir leads which are not of a problem for ICESat but have considerable potential to cause elevation biases for radar altimeters such as the Envisat RA-2. I would not call this "over-representation" or leads, though. Note that for ICESat the lead detection is a combination of low albedo and low elevation. Hence, in short, I recommend to give your argumentation more weight into the direction of surface type mixing and, as described by Tilling et al., the preferential sampling of large (and hence comparably thick) ice floes that applies to Envisat RA-2.
What I am missing in Section 4.1 are considerations towards the sensitivity of the ICESat total freeboard with respect to variations in sea-ice density (and additional parameters?) You might want to put this into section 4.3.
*Response:* We gratefully appreciate for your valuable comment.
(1) We have rephrased the term "lead detection skill" to "lead surface response" and expanded the description in the paragraph to improve the explanation how

the laser and radar altimeter characteristics differ over lead surfaces. Our point is that a single lead may dominate the radar backscatter for several consecutive Envisat footprint and thus indeed result in an overrepresentation of leads in Envisat waveform data.

The preferential sampling of large and comparable thick floes in Envisat data described in Tilling et al., is not a physical characteristic of pulse-limited radar altimetry, which preferentially samples high backscatter targets. It rather needs to be understood as a matter of specific surface type classification and filtering algorithms designed to exclude off-nadir returns in the freeboard retrieval. The Envisat retrieval algorithm in Tilling et al. differs from the one in the CCI sea ice project and thus the finding may not be directly applicable to our results.

Therefore, we amended the statements as: "While this is also directly applicable to radar altimeters with different footprints, the response to lead surfaces of laser (ICESat) and radar (Envisat) altimeters is not directly a function of footprint size but also of altimeter concept. Leads dominate radar backscatter even if the leads are already covered by thin sea ice for nadir and off-nadir cases and thus cause an overrepresentation of lead detections with range biases for off-nadir leads in Envisat data. Off-nadir reflections are usually detected and removed from the freeboard retrieval resulting in an underrepresentation of areas with mixed surface types in the Envisat freeboard statistics. Lead laser backscatter instead is a function of the surface albedo thus leads return lower laser backscatter power and since ICESat footprints do not overlap, the lead oversampling and necessary filtering of off-nadir reflections is not an issue for laser altimetry. For variable ice surfaces, the smaller footprint of ICESat has the capability to provide more detailed observations in areas with heterogeneous ice conditions than the pulse-limited Envisat footprint." *(please see P12 line 340-349 in the revised manuscript)*

(2) We have added a sensitivity discussion of ICESat total freeboard with respect to variations in sea-ice density in section 4.3. We set $\rho_{snow}$ = 300 kg m$^{-3}$, R = 6.8(FM), 6.0(MJ), 5.4(ON), freeboard = 0.2/0.3m, and sea ice density ranging from 880 to 940 kg m$^{-3}$ to analyse the sensitivity (RFig. 6). Therefore, we stated the results as: "We analyse the sensitivity of ICESat SIT to sea ice density and find an increase of ~0.2–0.4 m SIT when sea ice density rises from 880 to 940 kg m$^{-3}$ under seasonal R values and total freeboard." *(please see P14 line 418-419 in the revised manuscript)*

[Figure]

**RFig. 6:** Sensitivity of ICESat SIT to sea ice density under different total freeboard and R values.

18. L376-377: I don't understand which AMSR-E data set is used here. Please provide a reference and describe in more detail how this data was obtained (daily / monthly) and what you did to compute the SIT shown in Fig. S2.

    *Response:* Accepted and corrected. We have clarified the source and computing method of AMSR-E snow depth in the text as: "To further quantify the differences between snow depth climatology and actual snow depth contributions, we conduct the retrieval of Envisat SIT by replacing the snow depth climatology with monthly SICCI AMSR-E snow depth provided by SICCI (Kern et al., 2015). The new Envisat SIT is converted through Eq. (1) with Envisat monthly gridded sea ice freeboard data, monthly AMSR-E snow depth and the same density values mentioned above. The new Envisat SIT is compared with ICESat SIT and the changes to former Envisat-ICESat differences are shown in Table. S1." *(please see P13 line 382-386 in the revised manuscript)*

19. L392-393: I am not sure I understand why you add snow depth to an assumingly correct product (Envisat) instead of subtracting the snow depth from the ICESat total SIT. This would be much more straightforward and - in my eyes - the more correct thing to do. Please change this issue accordingly by computing a new Fig. S3 and changing the text accordingly.

    *Response:* Accepted and corrected. We have changed our comparison method following your suggestion and turned the heatmap to tables following another referee's comment. We have amended the description as: "To examine this issue, we subtract the snow depth climatology (used in Envisat retrieval) from the ICESat data and compare them with Envisat thickness. The changes to the former differences are shown in Table. S2, which is also a representation of the snow depth climatology itself." *(please see P14 line 406-408 in the revised manuscript)*

    RTable. 1 Changes to the differences that Envisat minus ICESat SIT for each

comparison period and each region by subtracting snow depth climatology (used in Envisat retrieval) from ICESat SIT. (Unit: m)

|      | ABS | WW | EW | EA | RS |
|------|-----|-----|-----|-----|-----|
| FM04 | 0.31 | 0.42 | 0.35 | 0.28 | 0.24 |
| FM05 | 0.32 | 0.44 | 0.38 | 0.27 | 0.23 |
| FM06 | 0.31 | 0.44 | 0.35 | 0.24 | 0.22 |
| FM07 | 0.23 | 0.40 | 0.28 | 0.18 | 0.13 |
| FM08 | NAN | 0.45 | 0.34 | 0.26 | 0.22 |
| MJ04 | 0.17 | 0.35 | 0.20 | 0.12 | 0.15 |
| MJ05 | 0.17 | 0.35 | 0.21 | 0.12 | 0.15 |
| MJ06 | 0.18 | 0.36 | 0.20 | 0.12 | 0.15 |
| ON04 | 0.20 | 0.23 | 0.17 | 0.12 | 0.17 |
| ON05 | 0.20 | 0.22 | 0.17 | 0.12 | 0.16 |
| ON06 | 0.19 | 0.21 | 0.16 | 0.12 | 0.16 |
| ON07 | 0.19 | 0.24 | 0.17 | 0.12 | 0.18 |

20. L398-400: The work of Li et al. is a substantial enhancement of the approach proposed in Kern et al. (2016). I see less a problem in their attempt to justify and evaluate their approach with comparably old in-situ observations than in combining two ICESat approaches making assumptions that kind of exclude their parallel usage. The Kern and Ozsoy-Cicek 2016 snow depth from ICESat approach assumes, in a way, that the sea-ice freeboard is more or less zero and that the total freeboard measured by ICESat is a reasonably good measure of the snow depth on sea ice. In contrast, the Kern et al. 2016 approach very well considers that there are non-zero sea-ice freeboards. But this is of course not your issue here. I would say it does not harm to delete this "in addition ... as well." sentence.

*Response:* Accepted and corrected. We have removed this statement following your suggestion.

21. L401-403: What you write here is not entirely correct. The procedure that is used in the ICESat retrieval to approaximate the SSH is not necessarily influenced by the geoid or the SSH itself - the latter is approximated from residuals which are already void of a geoid influence.

Also, as Kern and Spreen (2015) is the work published earlier, I doubt that "Consequently" is the right beginning of the second sentence here. I suggest to read these two papers once again, to perhaps also read Kern et al., The Cryosphere, 2015, and then come up with a more elaborated estimation of the uncertainties and/or sensitivities of the ICESat SIT product used - including statements about the sensitivity to densities.

*Response:* Accepted and corrected. We gratefully appreciate for your comment and these paper articles inspire us a lot. We have corrected our statements and modified the discussion of ICESat SIT uncertainties as: "Apart from the uncertainties from ICESat retrieval method mentioned above, Kern et al. (2016) also discussed the potential biases due to total freeboard and sea ice density. In comparison with the freeboard from Kurtz and Markus (2012), modal and mean total freeboard values

of this product are slightly higher, which might be a potential source of SIT positive biases. The total freeboard retrieved from ICESat has an uncertainty of up to 0.1 m, mainly due to the choice of percentage of observations used as sea surface height tie points (Kern and Spreen, 2015). Meanwhile, a smaller sea ice density will lead to smaller modified ice-snow density and SIT according to Eq. (2) and Eq. (3). **We analyse the sensitivity of ICESat SIT to sea ice density and find an increase of ~0.2–0.4 m SIT when sea ice density rises from 880 to 940 kg m$^{-3}$ under seasonal R values and total freeboard.**" *(please see P14 line 413-419 in the revised manuscript)*

22. Figure 7: Please add the number of the valid data pairs used.
    Please add the used binsize. How many data are truncated by not showing values above 5 m? Is the probability shown normalized such that its sum is equal to 1?
    ***Response:*** Accepted and corrected. We have modified Fig. 7 following your suggestions. Few data are truncated, 2 for summer and spring and 6 for autumn. The probability distribution is normalized and we have added this in the caption.

[Figure]

**RFig. 7:** Probability of the Envisat SIT and the ICESat SIT for all the individual comparison pairs. The blue stairs represent Envisat ice thickness and the red stairs represent ICESat ice thickness. The solid lines indicate the modal ice thickness and the dashed lines indicate the mean ice thickness of both data sets. The bin size is 0.2 m and the probability distribution is normalized.

23. L655 / Figure 8: I don't see what we can learn from this figure. The combination of SIT differences between ON and MJ periods with the "dynamic FDD" is not really enlightening, is it? Neither in this figure nor in the text in which you describe / interpret the figure solid information comes across. The FDD maps basically show where (a lot) of (new/thin) ice is produced thermodynamically and advected away from its source regions. It is your take, though.
If you decide to keep this aspect then I recommend to:
i) skip the top part with the differences between summer and autumn;
ii) make sure that the difference maps only show values where both periods' ENV-ICE maps (Fig. 4 and Fig. 6) have valid values;
iii) enlarge the maps;

iv) switch the color table of the SIT differences such that an increase in SIT gets blue colors while a decrease gets the red colors.

*Response:* Thank you for your comment. We have carefully considered your suggestions and agreed that this figure cannot explain the different MJON SIT changes. Therefore, we have removed this figure and analyzed the freeboard and snow depth changes from MJ to ON. Besides, for point *iv* we guess that you want the positive/negative differences to represent sea ice freezing/melting, so labelling the freezing/melting as blue and red does make it more physical. However, considering that most readers will subconsciously associate red with increase, we still want to keep the original colormap.

The results according to the new figure are added as follows: "Therefore, we further compare the mean variations of Envisat SIT, ICESat SIT, Envisat freeboard, ICESat freeboard and snow depth climatology used in Envisat retrieval from autumn (MJ) to spring (ON), shown in Fig. 8. The average fields are calculated with grid cells where both Envisat and ICESat SIT have valid values in all three years from 2004 to 2006. Figure 8 shows that Envisat SIT experiences general decreases from May/June-October/November (MJON) except Bellingshausen Sea and part of Amundsen Sea. Significantly large decreases exist in Western Weddell Sea. In contrast, ICESat SIT present large-scale increases except Western Weddell Sea and Ross Sea where slight decreases exist. By comparing the SIT and freeboard changes of both products, we find that the different changes of freeboard dominantly explain the SIT differences. One thing we can give a speculation based on the analyses in autumn and the regular rule during freezing seasons is that Envisat freeboard is probably overestimated in autumn, which has been pointed out in several studies before (e.g., Willatt et al., 2010; Kwok and Kacimi, 2018; Kacimi and Kwok, 2020). Moreover, the snow depth climatology also shows a decrease in Western Weddell Sea and Ross Sea (Fig. 8e), which has been reported by Kern and Ozsoy-Cicek (2016) that AMSR-E snow depth is likely to underestimate the snow depth evolution during MJON, also contribute to the Envisat SIT decrease." *(please see P9-10 line 269-281 in the revised manuscript)*

[Figure]

**RFig. 8:** The average changes of Envisat SIT, ICESat SIT, Envisat freeboard, ICESat freeboard and snow depth climatology used in Envisat retrieval from autumn to spring (MJON) calculated from 2004, 2005 and 2006.

24. Figure 9: Maybe you did this already but I recommend to use a specific order of plotting the data. I would first plot the data with the highest population, while the data (presumably summer) with the smallest population I would plot last. That way they would be visible better in the scatterplots. I also recommend to decrease the size of the symbols.

    *Response:* Accepted and corrected. We have modified Fig. 9 following your suggestions.

[Figure]

**RFig. 9:** Scatterplots of the individual data pairs between Envisat SIT and ICESat SIT for each region and each season. The data are taken from all seasons available. Since the comparison pairs are too few in Indian Ocean and western Pacific Ocean, we combine these two regions into Eastern Antarctic. The respective correlation coefficients and RMSDs are indicated in the panels. The black line is the 1-to-1 fit line and the dashed colored lines stand for linear regression lines.

Editoral comments / Typos:

L21-23: There is a paper by Tilling et al., JGR-Oceans, from 2019, (actually you cite that further down) which suggests preferential sampling of thicker, larger ice floes by Envisat RA-2 which, even though demonstrated in that paper for Arctic conditions, almost certainly also applies to Antarctic conditions (see e.g. Paul et al., 2018). I suggest to mention this additional bias source here in the abstract but also at the respective place within the text further below.

*Response:* We gratefully appreciate for your valuable suggestion. However, as what we have replied in No. 17 above, the preferential sampling of large and comparable thick floes in Envisat data described in Tilling et al., is not a physical characteristic of pulse-limited radar altimetry, which preferentially samples high backscatter targets. It rather needs to be understood as a matter of specific surface type classification and filtering algorithms designed to exclude off-nadir returns in the freeboard retrieval. The Envisat retrieval algorithm in Tilling et al. differs from the one in the CCI sea ice project and thus the finding may not be directly applicable to our results. However, we have

added the biases source of sea ice density here as: "Our findings suggest that both overestimation of Envisat sea ice freeboard potentially caused by radar backscatter originating from inside the snow layer, **as well as the AMSR-E snow depth biases and sea ice density uncertainties** can possibly account for the differences between Envisat and ICESat SIT." *(please see P1 line 21-24 in the revised manuscript)*

L28: Since the Antarctic sea ice cover has been quite dynamic in recent years, I suggest to also add the paper by Parkinson and DiGirolamo, 2022, Remote Sensing of Environment, which illustrates well that Southern Ocean sea-ice coverage did not yet recover from that drop in 2015/16.
*Response:* According to your suggestion, we have added the information here as: "During 2016-2020, the sea ice coverage in the Southern Ocean did not recover and set eight new Antarctic monthly record lows instead (Parkinson and DiGirolamo, 2021)." *(please see P2 line 30-31 in the revised manuscript)*

L207: The Envisat SIT error bars are not trustable; these are too large. The ICESat error bars do match better what one would expect. The notion "cover few observations" in the context of the error bars is not well chosen. I recommend that you write something along the lines: "Only few ULS observations fall within the possible Envisat SIT range indicated by the error bars" ... but actually you should only write this for the ICESat data.
*Response:* Thank you for your comment. We have removed the statement of Envisat error bars and modified that of ICESat error bars as: "Envisat thickness exceeds ULS, with a maximum value larger than 5 m. In comparison, ICESat thickness also exceeds ULS and only few ULS observations fall within the possible ICESat SIT range indicated by the error bars." *(please see P7 line 198-200 in the revised manuscript)*

L212/213: I don't second this notion. I am sure that the data set producers did their best to take into account well characterized and quantified contributions and I am sure that you find reasonable descriptions of how the uncertainties are computed in the respective papers and reports. The mere problem I see is the mentioned correlation and the different spatial scales involved.
*Response:* Accepted and corrected. We have amended the sentence to: "The differences in the error bars between Envisat and ICESat mainly result from their different spatial scales, the inclusion of snow depth uncertainty and lack of adequate regards for potential correlations between the error contribution in Envisat SIT, hence making it difficult to estimate realistic uncertainties." *(please see P7 line 203-205 in the revised manuscript)*

L303: "with an extra ... product" --> "together with a snow depth climatology derived from AMSR-E and AMSR2 data"
*Response:* Accepted and corrected. We have amended the sentence to: "Envisat directly uses the hydrostatic equilibrium together with a snow depth climatology derived from AMSR-E and AMSR2 data". *(please see P11 line 301-302 in the revised manuscript)*

L346-348: I suggest to reformulate things here because I have the impression that you are mixing some information. While the algorithm has in fact been developed for the Antarctic originally it has later been applied to the Arctic as well. Only there it runs into problems with multiyear ice. In addition, the maximum retrievable value is not caused by the similar signature of thick snow and multiyear ice (which is a problem in general) but is due to the saturation of the signal. Above a certain snow depth there is no change in the brightness temperature gradient ratio used that can be reliably related to snow depth.

*Response:* Accepted and corrected. According to your comment, we have amended the description as: "According to Markus and Cavalieri (1998), their AMSR-E snow depth product is limited to the maximal retrieval value being around 50 cm because of the saturation of the signal, i.e., there is no change in the brightness temperature gradient ratio with increasing snow depth over a certain limit." *(please see P12 line 353-355 in the revised manuscript)*

L349: You might want to add "Kern and Ozsoy-Cicek, Remote Sensing, 2016" here because they kind of rounded up the problems that exist with AMSR-E snow depths over Antarctic sea ice. This might replace the report by Frost et al.

*Response:* Accepted and corrected. We have replaced Frost et al. (2014) with Kern and Ozsoy-Cicek (2016). *(please see P12 line 257 in the revised manuscript)*

L381-383: Please provide the reader with an idea of the significance of this correction for Antarctic conditions. Is this an impact of the same size as the one described before?

*Response:* Accepted and corrected. The biases of this correction are smaller than the ones described before according to Mallett et al. (2019). We have added more information about this correction: "Moreover, the distance between sea ice surface elevations and the sea surface height is computed with vacuum light speed, which is defined as radar freeboard (RFB). A geometric correction used to correct the slower wave propagation speed in the snow layer is applied to convert the radar freeboard into the sea ice freeboard (FB):

$$FB = RFB + 0.22 \times SD \tag{7}$$

But the delay correction is based on a conventional assumption that has been assessed by Mallett et al. (2019), which pointed out that it introduced systematic underestimation of up to 15 cm into SIT estimates. While this systematic bias is small compared to other sources, uncertainties of snow depths and incomplete radar wave penetration would cause larger biases in this way." *(please see P13-14 line 390-397 in the revised manuscript)*

L412-414: Albeit late I need to make the comment that there are many more ICESat SIT products available. Alone in the Kern et al. (2016) paper several are presented and discussed. In that context it might make sense to revisit that paper and check what other approaches exist, e.g. the SICCI-SIT product which uses actual AMSR-E snow depth data, or the Kurtz and Markus product which assumes zero sea-ice freeboard and takes

the total freeboard as snow depth. These different products kind of span a possible range within which ICESat SIT values may be located and I recommend that you clearly state that the ICESat product you used is just ONE possible realization of many.

*Response:* Accepted and corrected. We have added this statement following your suggestion: "While we choose one of the ICESat SIT products to conduct the comparison, there are many other ICESat SIT products using different retrieval algorithms available, e.g., the SICCI product discriminating between positive and negative sea ice freeboard (Kern et al., 2016), or the one assuming zero sea ice freeboard in freeboard-to-thickness conversion (Kurtz and Markus, 2012). These different products provide a range of values within which ICESat SIT may be located, thus the differences between Envisat and ICESat SIT in this study are just one of the possible outcomes." *(please see P15 line 443-447 in the revised manuscript)*

**References:**

Barber, D. G., Reddan, S. P., and LeDrew, E. F.: Statistical characterization of the geophysical and electrical properties of snow on Landfast first-year sea ice, J. Geophys. Res., 100(C2), 2673– 2686, https://doi.org/10.1029/94JC02200, 1995.

Frost, T., Heygster, G., and Kern, S.: ANT D1.1 Passive Microwave Snow Depth on Antarctic sea ice assessment v1.0, available at: https://icdc.cen.uni-hamburg.de/fileadmin/user_upload/ESA_Sea-Ice-ECV/SICCI_ANT_SIT_Option_D1.1_Issue_1.0.pdf, 2014.

Haas, C., Thomas, D., and Bareiss, J.: Surface properties and processes of perennial Antarctic sea ice in summer. Journal of Glaciology, 47(159), 613-625, https://doi.org/10.3189/172756501781831864, 2001.

Kacimi, S. and Kwok, R.: The Antarctic sea ice cover from ICESat-2 and CryoSat-2: freeboard, snow depth, and ice thickness, The Cryosphere, 14, 4453–4474, https://doi.org/10.5194/tc-14-4453-2020, 2020.

Kern, S., and Ozsoy-Cicek, B.: Satellite Remote Sensing of Snow Depth on Antarctic Sea Ice: An Inter-Comparison of Two Empirical Approaches. Remote Sens., 8, 450, https://doi.org/10.3390/rs8060450, 2016.

Kwok, R. and Kacimi, S.: Three years of sea ice freeboard, snow depth, and ice thickness of the Weddell Sea from Operation IceBridge and CryoSat-2, The Cryosphere, 12, 2789–2801, https://doi.org/10.5194/tc-12-2789-2018, 2018.

Mallett, R. D. C., Lawrence, I. R., Stroeve, J. C., Landy, J. C., and Tsamados, M.: Brief communication: Conventional assumptions involving the speed of radar waves in snow introduce systematic underestimates to sea ice thickness and seasonal growth rate estimates, The Cryosphere, 14, 251–260, https://doi.org/10.5194/tc-14-251-2020, 2020.

Massom, R. A., Drinkwater, M. R., and Haas, C.: Winter snow cover on sea ice in the Weddell Sea, J. Geophys. Res., 102, 1101–1117, https://doi.org/10.1029/96jc02992, 1997.

Willatt, R. C., Giles, K. A., Laxon, S. W., Stone-Drake, L., and Worby, A. P.: Field Investigations of Ku-Band Radar Penetration into Snow Cover on Antarctic Sea Ice, IEEE Trans. Geosci. Remote Sens., 48, 365–372,

https://doi.org/10.1109/TGRS.2009.2028237, 2010.

Williams, G., Maksym, T., Wilkinson, J., Kunz, C., Murphy, C., Kimball, P., and Singh, H.: Thick and deformed Antarctic sea ice mapped with autonomous underwater vehicles, Nature Geosci., 8, 61–67, https://doi.org/10.1038/ngeo2299, 2015.

**Responses to referee #2**

Dear Reviewer:

We would like to express our gratitude to you for the helpful comments to improve this manuscript. We have carefully modified the expressions following your comments. The specific responses and revisions are shown below. They are in blue font for clarity.

Qinghua Yang and Qian Shi
On behalf of all the authors

P3L77-78: Replace "ICESat-1, hereinafter ICESat" by ICESat as this the name of the mission.
*Response:* Accepted and corrected. According to your suggestion, we have replaced "ICESat-1, hereinafter ICESat" by "ICESat" as: "The Geoscience Laser Altimeter System (GLAS) aboard the Ice, Cloud and land Elevation Satellite (ICESat) allows estimating the total freeboard through the determination of the surface elevation from 2003 to 2009, illustrated in Fig. 1." *(please see P3 line 80-81 in the revised manuscript)*

P3L91-92: Can you specify which independent observations you are referring to?
*Response:* We have specified the independent observations here according to Kern et al. (2016) as: "…we choose the ICESat SIT derived from the modified density approach for comparison, which seems to agree with average SIT from independent observations **like ASPeCt, ULS and AEM** and has a reasonable winter-to-spring growth (Kern et al., 2016)." *(please see P3-4 line 93-95 in the revised manuscript)*

P4L118: Remove "cover" in snow cover.
*Response:* Accepted and corrected. We have changed the "snow cover" to "snow". *(please see P4 line 120 in the revised manuscript)*

P6L163: Replace "accumulative" by cumulative.
*Response:* Accepted and corrected. However, we have removed this sentence because according to another referee's suggestions, we have decided to remove the FDD parts and focus on the intercomparison between Envisat and ICESat since the results of FDD cannot explain the reason for the differences.

P6L166-169: Can you give more details on how the "dynamic FDD" is computed? You mention that "the NSIDC Polar Pathfinder daily 25 km EASE-grid sea ice motion data are applied to produce the forward tracking on daily FDD" but how is this implemented really? How do you link the number of freezing degree days to dynamics and ice deformation?
*Response:* Our procedures on deriving dynamic FDD are listed as follows: a. On Day1, the historical FDD and the newly increased FDD are interpolated to the regular NSIDC

ice velocity grid position (X1, Y1); b. Accumulated FDD moves to irregular positions (X2, Y2) with sea ice motion derived from NSIDC v4; c. At the beginning of Day2, the FDD distribution of the irregular Day1 is interpolated to the regular grid (X1, Y1), then repeating steps 1-3. The ice divergence and deformation situations cannot be represented through our method. However, according to another referee's suggestions, we have decided to remove the FDD parts and focus on the intercomparison between Envisat and ICESat since the results of FDD cannot explain the reason for the differences.

P7L174: Replace "realized in" by "done for".
*Response:* Accepted and corrected. We have changed the "realized in" to "done for". *(please see P6 line 164 in the revised manuscript)*

P7L189: Replace "here compare" by "compare here".
*Response:* Accepted and corrected. We have modified this sentence according to your next comment.

P7L189-190: Replace "We here compare Envisat and ICESat actual SIT with ULS observations, thus we divide ICESat SIT by SIC contained in the data for each grid." By "In order to compare the SIT from the two satellites with the ULS observations, we first compute the ICESat effective SIT by dividing the SIT by the SIC at each grid cell".
*Response:* Accepted and corrected. We have changed the original sentence to "In order to compare the SIT from the two satellites with the ULS observations, we first compute the ICESat effective SIT by dividing the SIT by the SIC at each grid cell". *(please see P7 line 179-180 in the revised manuscript)*

P8L208: Please rephrase "and the smaller error bars of ICESat also cannot cover the observations" as this is not clear.
*Response:* Accepted and corrected. "In comparison, ICESat thickness also exceeds ULS and only few ULS observations fall within the possible ICESat SIT range indicated by the error bars." *(please see P7 line199-200 in the revised manuscript)*

P8L215-216: Replace "largest/smallest" by either "the largest/the smallest" or "larger/smaller".
*Response:* Accepted and corrected. We modified the sentence to: "The statistics show that both MDs are the largest at site 207 (1.63 m for Envisat-ULS and 1.73 m for ICESat-ULS) and the smallest at site 229 (0.72 m for Envisat-ULS and 0.42 m for ICESat-ULS)." *(please see P8 line 207-209 in the revised manuscript)*

P8L229-231: The following statement seems somehow contradictory. You want to prove the validity of the ULS data for comparison with satellite measurements, yet point to the heterogeneity of the sea ice measurements at each ULS? "With the sea ice motion data from NSIDC introduced in Sect. 2.4, the 30-day origins of the sea ice passing the three ULS sites in July 2011 is shown in Fig. S1 and it proves the heterogeneity of sea

ice measured by each ULS and the 230 validity of ULS data usage in comparison with satellite products."

*Response:* Thanks for your comment. The "heterogeneity" indeed made reader confused. We want to emphasize that though the footprint of ULS is smaller than Envisat and ICESat, continuous ULS observations can still represent the sea ice thickness over a large area due to its continuous sampling and irregular sea ice motion. In a sense, ULS sampling can be analogized to randomly drilling in a range of sea ice cover. In order to make our explanation clearer, we amended the statement as: "With the sea ice motion data from NSIDC introduced in Sect. 2.4, the 30-day origins of the sea ice passing the three ULS sites from July 2 to July 31, 2011 is shown in Fig. S1 and it is spatially representative". *(please see P8 line 222-223 in the revised manuscript)*

P9L243: Replace "ICESat maps" by "ICESat fields" here and throughout the text.
*Response:* Accepted and corrected. We have replaced "ICESat maps" by "ICESat fields" throughout the text. *(please see P8 line 235-236 in the revised manuscript)*

P9L246: Can you clarify the following statement: "consideration of less accurate total freeboard there". Do you mean that ICESat does not provide freeboard estimates within a certain distance of the coast because of freeboard quality requirements?
*Response:* Yes, and we clarified and amended the sentence as: "This can be attributed to a different land mask used in the ICESat product and consideration of lower freeboard quality there." *(please see P9 line 238-239 in the revised manuscript)*

P9L256: Replace "expectant" by expected.
*Response:* Accepted and corrected. We have replaced "expectant" by "expected". *(please see P9 line 250 in the revised manuscript)*

P10L269-270: "In summer, the agreement between Envisat SIT and ICESat SIT is not good, mainly due to their different performances on thick ice above 3 m." Can give more details on why the instruments have different performances over thick ice?
*Response:* Thank you for your comment. Based on the discussion on the differences in Sect. 4 and previous studies (e.g., Willatt et al., 2010), during FM, snow is deep, potentially wet and/or metamorphous on thick ice, causing substantial penetration difficulties for radar altimeters.

P10L272: "Envisat mean SIT decreases from autumn to spring". Could this be related to the snow depth climatology? For a same freeboard value, a higher snow loading would provide thicker ice. I wonder what is the variability from fall to spring of the Envisat freeboards and snow depth.
*Response:* Thank you for your comment. We have carefully considered your suggestions and agreed that this figure cannot explain the different MJON SIT changes. Therefore, we have removed this figure and analyzed the freeboard and snow depth changes from MJ to ON.
The results according to the new figure are added as follows: "Therefore, we further

compare the mean variations of Envisat SIT, ICESat SIT, Envisat freeboard, ICESat freeboard and snow depth climatology used in Envisat retrieval from autumn (MJ) to spring (ON), shown in Fig. 8. The average fields are calculated with grid cells where both Envisat and ICESat SIT have valid values in all three years from 2004 to 2006. Figure 8 shows that Envisat SIT experiences general decreases from May/June-October/November (MJON) except Bellingshausen Sea and part of Amundsen Sea. Significantly large decreases exist in Western Weddell Sea. In contrast, ICESat SIT present large-scale increases except Western Weddell Sea and Ross Sea where slight decreases exist. By comparing the SIT and freeboard changes of both products, we find that the different changes of freeboard dominantly explain the SIT differences. One thing we can give a speculation based on the analyses in autumn and the regular rule during freezing seasons is that Envisat freeboard is probably overestimated in autumn, which has been pointed out in several studies before (e.g., Willatt et al., 2010; Kwok and Kacimi, 2018; Kacimi and Kwok, 2020). Moreover, the snow depth climatology also shows a decrease in Western Weddell Sea and Ross Sea (Fig. 8e), which has been reported by Kern and Ozsoy-Cicek (2016) that AMSR-E snow depth is likely to underestimate the snow depth evolution during MJON, also contribute to the Envisat SIT decrease." *(please see P9-10 line 269-281 in the revised manuscript)*

[Figure]

**RFig. 1:** The average changes of Envisat SIT, ICESat SIT, Envisat freeboard, ICESat freeboard and snow depth climatology used in Envisat retrieval from autumn to spring (MJON) calculated from 2004, 2005 and 2006.

P10L273: replace "more thick" by thicker.

*Response:* Thank you for your comment. We have realized the misunderstanding here and amended the sentence as: "For Envisat SIT, the distribution indicates that more ice is in thinner categories in spring than autumn, while more ice in thicker categories is found for ICESat SIT." *(please see P9 line 268-269 in the revised manuscript)*

P10L276-278: Please rephrase "We use FDD rather than converted SIT with an empirical equation because they represent the same mechanism and we cannot constrain the uncertainties sufficiently caused by additional assumptions." I do not understand the meaning or purpose of this statement.

*Response:* Accepted and corrected. This sentence is used to explain why we didn't convert the FDD values to SIT values with an empirical equation suggested by Lebedev (1938):

$$\text{Thickness (cm)} = 1.33 * \text{FDD (°C)}^{0.58}$$

We have rephrased this sentence following your comment: "Although the FDD can be converted to SIT with an empirical equation suggested by Lebedev (1938), we use FDD rather FDD-derived sea ice thickness because: (1) FDD represent the same mechanism on the evolution of sea ice thickness as the FDD SIT; (2) avoid the large uncertainties caused by additional assumptions." However, according to another referee's suggestions, we have decided to remove the FDD parts and focus on the intercomparison between Envisat and ICESat since the results of FDD cannot explain the reason for the differences.

P10L282-283: "one thing we can give a speculation based on the analyses in autumn and the regular rule during freezing seasons is that the main reason for Envisat SIT overall decrease during MJON is the overestimation of Envisat SIT in autumn." Can you provide citations that point to the overestimation of Envisat SIT in autumn?

*Response:* Accepted and corrected. We have provided related citations in the context as: "one thing we can give a speculation based on the analyses in autumn and the regular rule during freezing seasons is that the main reason for Envisat SIT overall decrease during MJON is the overestimation of Envisat SIT in autumn, which is reported in several studies before (e.g., Willatt et al., 2010; Kwok and Kacimi, 2018; Kacimi and Kwok, 2020)." *(please see P10 line 276-279 in the revised manuscript)*

P13L366-367: Can you clarify what you refer to in "the other ICESat period". Do you mean ICESat-2? Please clarify.

*Response:* Thank you for your comment. We are sorry for the inappropriate expression here and we have clarified the sentence as: "And for the same reasons, passive microwave snow depth is possibly underestimated on thick ice during FM but also during other seasons." *(please see P13 line 373-374 in the revised manuscript)*

P13L381-383: "Moreover, snow-depth dependent radar signal delay is applied to convert the radar freeboard into the sea-ice freeboard, but the delay correction is based on a conventional assumption that has been revised (Mallett et al. 2019) since the

generation of the SICCI data." Can you give more details about this correction and its revision?

*Response:* Accepted and corrected. We have added more information about this correction: "Moreover, the distance between sea ice surface elevations and the sea surface height is computed with vacuum light speed, which is defined as radar freeboard (RFB). A geometric correction used to correct the slower wave propagation speed in the snow layer is applied to convert the radar freeboard into the sea ice freeboard (FB):

$$FB = RFB + 0.22 \times SD \qquad (7)$$

But the delay correction is based on a conventional assumption that has been assessed by Mallett et al. (2019), which pointed out that it introduced systematic underestimation of up to 15 cm into SIT estimates. While this systematic bias is small compared to other sources, uncertainties of snow depths and incomplete radar wave penetration would cause larger biases in this way." *(please see P13-14 line 390-397 in the revised manuscript)*

Figure S1 to S3: Please change the captions as they are not clear:
-S1: "The origins (30-days ago) of the sea ice (blue dots) that passing through the three ULS sites", could be rephrased to "30-days backtracking of sea ice at the ULS sites".
*Response:* Accepted and corrected. We have changed the caption to: "Fig. S1 30-days backtracking of sea ice at the ULS sites (red dots) in July 2011 by using backward tracking method based on the NSIDC v4 sea ice motion data. The grey vectors represent the monthly mean sea ice drift derived from NSIDC v4."

-S2: "Changes in the differences between Envisat and ICESat SIT for each comparison period and each region under the experiment of the snow depth climatology impacts.", is not clear. It would be good to specify (Envisat minus ICESat SIT) and stating precisely what changes you make to the snow depth.
*Response:* Accepted and corrected. We have modified the caption as: "Changes to the differences that Envisat minus ICESat SIT for each comparison period and each region by replacing snow depth climatology with SICCI AMSR-E snow depth during Envisat SIT retrieval. (Unit: m)"

-S2 and S3: I suggest that you remove the colorscale and turn these figures into tables instead because it is confusing and having a colormap does not provide additional information.
*Response:* Accepted and corrected. We have turned the heatmaps into tables following your comment.

RTable. 1 Changes to the differences that Envisat minus ICESat SIT for each comparison period and each region by replacing snow depth climatology with SICCI AMSR-E snow depth during Envisat SIT retrieval. (Unit: m)

|  | ABS | WW | EW | EA | RS |
|---|---|---|---|---|---|
| FM04 | 0.25 | 0.06 | -0.07 | -0.03 | 0.08 |
| FM05 | -0.14 | 0.13 | -0.05 | -0.24 | -0.04 |
| FM06 | 0.11 | -0.34 | -0.20 | 0.20 | -0.09 |

| | | | | | |
|---|---|---|---|---|---|
| FM07 | 0.03 | -0.21 | 0.09 | 0.03 | 0.04 |
| FM08 | NAN | -0.08 | 0.05 | -0.04 | 0.06 |
| MJ04 | 0.11 | 0.27 | 0.03 | -0.01 | 0.07 |
| MJ05 | 0.24 | 0.18 | -0.07 | -0.03 | 0.11 |
| MJ06 | 0.06 | -0.09 | -0.04 | -0.03 | -0.01 |
| ON04 | 0.05 | -0.01 | 0.06 | -0.06 | -0.01 |
| ON05 | 0.22 | -0.02 | -0.06 | -0.07 | -0.01 |
| ON06 | 0.16 | -0.13 | -0.01 | 0.06 | 0.02 |
| ON07 | -0.05 | -0.00 | -0.01 | -0.02 | 0.11 |

RTable. 2 Changes to the differences that Envisat minus ICESat SIT for each comparison period and each region by subtracting snow depth climatology (used in Envisat retrieval) from ICESat SIT. (Unit: m)

| | ABS | WW | EW | EA | RS |
|---|---|---|---|---|---|
| FM04 | 0.31 | 0.42 | 0.35 | 0.28 | 0.24 |
| FM05 | 0.32 | 0.44 | 0.38 | 0.27 | 0.23 |
| FM06 | 0.31 | 0.44 | 0.35 | 0.24 | 0.22 |
| FM07 | 0.23 | 0.40 | 0.28 | 0.18 | 0.13 |
| FM08 | NAN | 0.45 | 0.34 | 0.26 | 0.22 |
| MJ04 | 0.17 | 0.35 | 0.20 | 0.12 | 0.15 |
| MJ05 | 0.17 | 0.35 | 0.21 | 0.12 | 0.15 |
| MJ06 | 0.18 | 0.36 | 0.20 | 0.12 | 0.15 |
| ON04 | 0.20 | 0.23 | 0.17 | 0.12 | 0.17 |
| ON05 | 0.20 | 0.22 | 0.17 | 0.12 | 0.16 |
| ON06 | 0.19 | 0.21 | 0.16 | 0.12 | 0.16 |
| ON07 | 0.19 | 0.24 | 0.17 | 0.12 | 0.18 |

**References:**

Lebedev, V. V.: The dependence between growth of ice in Arctic rivers and seas and negative air temperature (in Russian). Probl. Arkt. 5-6, 9-25, 1938.

Willatt, R. C., Giles, K. A., Laxon, S. W., Stone-Drake, L., and Worby, A. P.: Field Investigations of Ku-Band Radar Penetration into Snow Cover on Antarctic Sea Ice, IEEE Trans. Geosci. Remote Sens., 48, 365–372, https://doi.org/10.1109/TGRS.2009.2028237, 2010.